# Fast Zeroth-Order Convex Optimization
# with Quantum Gradient Methods

**Junhyung Lyle Kim**[*]     **Brandon Augustino**[*]     **Dylan Herman**[*]     **Enrico Fontana**[*]

**Jacob Watkins**          **Marco Pistoia**          **Shouvanik Chakrabarti**[†]

**Global Technology Applied Research, JPMorganChase**
New York, NY 10001 USA

## Abstract

We study quantum algorithms based on quantum (sub)gradient estimation using noisy function evaluation oracles, and demonstrate the first dimension-independent query complexities (up to poly-logarithmic factors) for zeroth-order convex optimization in both smooth and nonsmooth settings. Interestingly, only using noisy function evaluation oracles, we match the first-order query complexities of classical gradient descent, thereby exhibiting exponential separation between quantum and classical zeroth-order optimization. We then generalize these algorithms to work in non-Euclidean settings by using quantum (sub)gradient estimation to instantiate mirror descent and its variants, including dual averaging and mirror prox. By leveraging a connection between semidefinite programming and eigenvalue optimization, we use our quantum mirror descent method to give a new quantum algorithm for solving semidefinite programs, linear programs, and zero-sum games. We identify a parameter regime in which our zero-sum games algorithm is faster than any existing classical or quantum approach.

## 1 Introduction

Convex optimization has long been a central topic of study in computer science, mathematics, operations research, statistics, and engineering due to its large number of scientific and industrial applications. These problems are at the core of many machine learning pipelines, and generalize well studied settings such as linear programming, second-order conic programming, and semidefinite programming, to name a few. A convex optimization problem is of the form

$$\text{Find } \tilde{x} \in \mathcal{X} \text{ such that } f(\tilde{x}) - \min_{x \in \mathcal{X}} f(x) \leq \varepsilon, \tag{OPT}$$

where $\mathcal{X} \subseteq \mathbb{R}^d$ is a closed convex set with nonempty interior, $f : \mathbb{R}^d \to \mathbb{R}$ is convex and $G$-Lipschitz on $\mathcal{X}$ with respect to a given norm $\| \cdot \|$, and $\varepsilon > 0$ is an error parameter. We make the standard assumption that (OPT) is *solvable*, i.e., there exists an optimal point $x^\star \in \arg\min_{x \in \mathcal{X}} f(x)$, and that there exists $R \geq 1$ such that $\sup_{x \in \mathcal{X}} \text{dist}(x, x^\star) \leq R$ for a suitable distance metric $\text{dist}(\cdot, \cdot)$.

In practical applications, problems of the form (OPT) are solved in very high dimension $d$, to very high precision $\varepsilon$, or both. Developing algorithms with better scaling in $d$ and $1/\varepsilon$ is a primary goal of optimization theory. Due to the ubiquity and practical importance of convex optimization, there has been a significant effort to develop quantum algorithms providing speedups for these problems.

---

[*]These authors contributed equally.

[†]shouvanik.chakrabarti@jpmchase.com

39th Conference on Neural Information Processing Systems (NeurIPS 2025).

Research on convex optimization focuses on two different settings. In the *black-box* setting, the constraints defining the feasible region and the objective function are accessed via black-box oracles. The focus then is on developing algorithms that repeatedly query these oracles, with the aim of minimizing the number of queries necessary to solve the problem, which is called the *query complexity*. In the *white-box* setting, the objective function and constraints are known explicitly in some form, and the focus is on minimizing the time required to perform the optimization. As an example, linear programs are specified by the vector defining the (linear) objective function, while the constraints are specified by a matrix and right-hand-side vector. Aside from linear programs, white-box convex optimization generalizes common settings such as quadratic, second-order conic, and semidefinite programming. Black-box optimization algorithms can be directly applied to white-box optimization, but it is often possible to obtain more efficient methods by leveraging the specific problem structure, which has led to the development of many classical algorithms that are tailored to particular settings.

In this paper, we present improved algorithms for convex optimization in both the black-box and white-box settings. We begin with an investigation of black-box optimization via (sub)gradient methods, and show that quantum algorithms with access to only the noisy function value can match the query complexity of classical (sub)gradient methods that require the (sub)gradient of the objective function. Since the classical computation of a gradient in the oracle setting requires a number of queries that is linear in the dimensionality $d$, this represents an exponential speedup in terms of $d$.

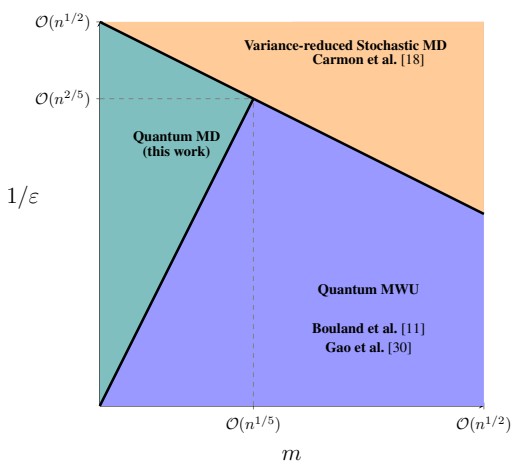

Figure 1: *Regimes for speedup in solving zero-sum games in the $(m, 1/\varepsilon)$-plane.* See also Corollary 3.1. Algorithms: Mirror Descent (MD), Matrix Multiplicative Weights (MWU).

Notably, our results in the black-box setting apply for optimization in both Euclidean and more general $\ell_p$-normed spaces ($p \geq 1$). This allows us to derive algorithms that offer improvements for white-box optimization. To this end, we leverage a connection between semidefinite programming and eigenvalue optimization that has been previously studied to motivate the *spectral bundle method*. Specifically, semidefinite programming is reduced to a nonsmooth optimization problem that is Lipschitz continuous in $\ell_1$-space. Leveraging our black-box results for this setting, we obtain algorithms for semidefinite programming, linear programming, and zero-sum games that improve upon the previously best known quantum and classical algorithms in some regimes; see Figure 1 for an illustration.

In short, we demonstrate several algorithmic speedups using quantum (sub)gradient methods in both the black-box and white-box settings. Specifically, our contributions can be summarized as follows:

- In Section 2, we show that, using only a *noisy* function evaluation oracle (Definition 1), the zeroth-order quantum projected subgradient method ($\ell_2$-space) and quantum mirror descent ($\ell_p$-space) match the first-order query complexities (up to polylogarithmic factors in $d$) achieved by their respective classical counterparts for solving (OPT). See, Theorem 2.1. We thus exhibit an *exponential* separation between quantum and classical zeroth-order optimization of convex and Lipschitz functions. These results are summarized in Table 1.

- In Section 3, we leverage the connection between semidefinite programming and eigenvalue optimization, and demonstrate how the quantum mirror descent framework can be applied to solving white-box settings, including semidefinite programming (Theorem 3.1), linear programming (Theorem 3.2), and zero-sum games (Corollary 3.1). In some regimes, we attain a better complexity (Table 2 and Figure 1) than the current state-of-the-art approaches (both quantum and classical).

- In Section 4, we characterize the effort required by zeroth-order quantum gradient methods to solve (OPT) when the objective function has Lipschitz continuous gradients, often referred to as the *L-smooth* setting in the optimization literature. In this setting, we study the zeroth-order quantum gradient descent ($\ell_2$-space) and mirror prox ($\ell_p$-space), again matching first-order query

Table 1: *Comparison of oracle complexities for **zero-order quantum (sub)gradient methods (this work)** and the first-order classical (sub)gradient methods on minimizing G-Lipschitz functions with additional structures (2nd column).* Algorithms: Gradient Descent (GD), Projected Subgradient Method (PSG), Mirror Descent (MD), Dual Averaging (DA), and Mirror Prox (MP).

| Domain | Assumptions on $f$ | Algorithm | Quantum (0th order) | Result | Classical (1st order) |
|---|---|---|---|---|---|
| $\ell_2$-space | Convex | PSG | $\widetilde{\mathcal{O}}\left(G^2 R^2/\varepsilon^2\right)$ | Thm 2.1 | $\mathcal{O}\left(G^2 R^2/\varepsilon^2\right)$ |
| (Euclidean) | Convex, $L$-smooth | GD | $\widetilde{\mathcal{O}}\left(LR^2/\varepsilon\right)$ | Thm 4.2 | $\mathcal{O}\left(LR^2/\varepsilon\right)$ |
| | $\mu$-PL, $L$-smooth | GD | $\widetilde{\mathcal{O}}\left(L/\mu\right)$ | Thm 4.3 | $\widetilde{\mathcal{O}}\left(L/\mu\right)$ |
| $\ell_p$-space | Convex | MD, DA | $\widetilde{\mathcal{O}}\left(G^2 R^2/\varepsilon^2\right)$ | Thm 2.1 | $\mathcal{O}(G^2 R^2/\varepsilon^2)$ |
| (Non-Euclidean) | Convex, $L$-smooth | MP | $\widetilde{\mathcal{O}}\left(LR^2/\varepsilon\right)$ | Thm 4.2 | $\mathcal{O}(LR^2/\varepsilon)$ |

complexities of the corresponding classical algorithms only using noisy function evaluation oracle. See, Theorem 4.2. Additionally assuming $f$ satisfies the $\mu$-PŁ condition, we show a $\widetilde{\mathcal{O}}(L/\mu)$ query complexity for zeroth-order quantum gradient descent (Theorem 4.3).

**Notation.** We let $\|\cdot\|_p$ denote the $\ell_p$-norm on $\mathbb{R}^d$ for $p \in [1,\infty]$: $\|v\|_p := \left(\sum_{i=1}^d |v_i|^p\right)^{1/p}$. For a finite-dimensional space $\mathcal{E}$, we denote its dual space by $\mathcal{E}^*$. If $\mathcal{E} \subset \mathbb{R}^d$ is equipped with an arbitrary inner product $\langle\cdot,\cdot\rangle$ and a norm $\|\cdot\|_p$, the *dual norm* $\|\cdot\|_*$ is $\|g\|_* := \sup_{\{x \in \mathbb{R}^d : \|x\| \leq 1\}} \langle g, x\rangle$. We define the (closed) $\ell_p$-ball of radius $r$ centered at $c \in \mathbb{R}^d$ as $\mathcal{B}_p^d(c,r) := \{x \in \mathbb{R}^d : \|x - c\|_p \leq r\}$. We define the Bregman divergence associated to $\Phi$ as $D_\Phi(x,y) = \Phi(x) - \Phi(y) - \langle\nabla\Phi(y), x - y\rangle$. We also use the standard definitions of (strongly) convex functions; more detailed notations are summarized in the supplementary material.

## 2 Black-box setting: zeroth-order quantum (sub)gradient methods

In the black-box setting, (sub)gradient methods access the problem through a *qth-order oracle* $O_f^{(q)}$ for the objective function $f$. On input $x \in \mathbb{R}^d$, $O_f^{(q)}$ returns the function value and its derivatives up to the $q$th-order:

$$O_f^{(q)} : x \mapsto \{f(x), \nabla f(x), \ldots, \nabla^q f(x)\}.$$

Let $G^{(q)}$ denote the Lipschitz constant of the $q$th-order derivative, i.e., for any $(x, \bar{x}) \in \mathcal{X} \times \mathcal{X}$,

$$\|\nabla^q f(x) - \nabla^q f(\bar{x})\|_* \leq G^{(q)}\|x - \bar{x}\|_p,$$

where $\|\cdot\|_p$ is an appropriate $\ell_p$-norm and $\|\cdot\|_*$ is its dual norm. For fixed $q$ and any $\varepsilon > 0$, the complexity of an algorithm for solving (OPT) can be expressed as a function of $d$ and $G^{(q)}R/\varepsilon$ (since one can always re-scale the input and output spaces, and adjust the precision accordingly). We focus on the high-dimensional setting where $d$ is potentially much larger than the other problem parameters, i.e., $d \gg \text{poly}(G^{(q)}R\varepsilon^{-1})$. For any $q \geq 1$, the number of queries to $O_f^{(q)}$ needed to solve (OPT) to precision $\varepsilon > 0$ using classical (sub)gradient methods depends polynomially on $G^{(q)}R/\varepsilon$ and thus is "independent" of the ambient dimension $d$ [61].

First-order methods play a fundamental role in large-scale optimization due to their *near* dimension independence. While the convergence rates of first-order methods are far slower than the (local) quadratic convergence enjoyed by Newton's method, first-order methods exhibit an advantage with respect to simplicity, robustness, and the fact that they provably converge to the global optimum of a convex optimization problem irrespective of the starting point. From a practical standpoint, algorithms that rely on computing second-order information at every iterate, such as *interior point methods* (IPMs), are often outperformed by first-order methods for large-scale problems.

Recent studies [31, 32] have demonstrated that no quantum speedup over classical (accelerated) gradient descent rates can be achieved when $q \geq 1$. Consequently, it is natural to consider instances of (OPT) where derivatives are unavailable (or accessible only at a prohibitive cost). In optimization theory, this is modeled by so called *zeroth-order* oracles that only specify the objective function value and not its gradients (i.e., $q = 0$), and hence is also called *derivative-free* optimization.

In the derivative-free setting, there is reason to be optimistic about the potential for quantum speedup due to the aforementioned algorithms for quantum gradient [45, 34] and subgradient [20, 75] estimation, which provide an exponential speedup in $d$ for estimating a gradient from function evaluations in the black-box model. However, previous applications of these algorithms to convex optimization [20, 75] have focused on arbitrarily constrained problems in the high-precision regime, rather than the low-precision regime where gradient methods typically outperform more complex techniques such as cutting plane or center of gravity methods. Consequently, the speedups in terms of dimension are only quadratic. It is natural to wonder whether the dimension-independent rates of gradient methods can be achieved using only function evaluations. Our first set of results answers this question in the affirmative, as summarized in Table 1, under the approximate binary oracle model below:

**Definition 1** ($\theta$-approx. binary oracle). *A unitary $U_f^{(\theta)}$ is a $\theta$-approximate binary oracle for $f$ if*

$$U_f^{(\theta)} : |x\rangle|y\rangle \mapsto |x\rangle|y \oplus \widetilde{f}(x)\rangle \quad \textit{such that} \quad \|\widetilde{f} - f\|_\infty < \theta.$$

Approximate binary oracles are the standard function oracle considered for quantum optimization algorithms [10, 20, 75], since any classical arithmetic circuit for a function can be converted into a binary oracle by implementing each arithmetic operation with reversible quantum arithmetic.

Before discussing the black-box results, we briefly review quantum (sub)gradient estimation. (Sub)gradient-based methods are a popular class of iterative black-box methods which generate a sequence of candidate solutions based on the (sub)gradients of the objective function. Gradient methods lend themselves to quantum speedups due the existence of a quantum algorithm proposed by Jordan [45] and refined by Gilyén et al. [34], to estimate the gradient of a function in $\widetilde{\mathcal{O}}(1)$ function evaluation queries.[3] These algorithms were generalized to compute subgradients of convex functions in [20, 75], resulting in the first query complexity speedups for constrained convex optimization. We state the subgradient estimation lemma based on [75], which we slightly modified for our purposes.

**Lemma 2.1** (Subgradient estimation ([75, Lemma 18])). *Suppose $f : \mathbb{R}^d \to \mathbb{R}$ is a convex function that is $G$-Lipschitz on $\mathcal{B}_\infty(0, 2r_1)$ with a given norm $\|\cdot\|_p$ with $p \geq 1$, and we have quantum query access to $\tilde{f}$, which is a $\theta$-approximate version of $f$, as in Definition 1. Let $r_1, G > 0$, $\rho \in (0, 1/3]$, and suppose $\theta \in (0, r_1 dG/\rho]$. Then, we can compute a subgradient $\tilde{g} \in \mathbb{R}^d$ using $\mathcal{O}(\log(d/\rho))$ queries to $\theta$-approximate binary oracle, such that with probability $\geq 1 - \rho$, we have*

$$f(y) \geq f(x) + \langle \tilde{g}, y - x \rangle - (23d)^2 \sqrt{\frac{\theta G}{\rho r_1}} \|y - x\|_p - 2G\sqrt{d}r_1, \quad \sup_{g \in \partial f(x)} \|\tilde{g} - g\|_\infty \leq \sqrt{\frac{\theta d^3 G}{\rho r_1}}.$$

That is, one can obtain an approximate subgradient $\tilde{g}$ using an approximate binary oracle in Definition 1, with some bias that has to be appropriately controlled. In Section 4, we also provide an improved gradient estimation based on [72] where one can control both the bias and the variance (Theorem 4.1), which we apply to solve convex optimization problems in the $L$-smooth setting.

Equipped with Lemma 2.1, we study the quantum projected subgradient method, which iterates as:

$$x_{t+1} = \Pi_{\mathcal{X}} \left( x_t - \eta \tilde{g}_{x_t} \right), \quad \tilde{g}_{x_t} \overset{\text{estimate}}{\sim} \text{Lemma 2.1} \qquad \text{(QPSM)}$$

While the projected subgradient method is optimal for convex and Lipschitz functions [61], the convergence guarantee depends on problem-dependent parameters being well-behaved in the Euclidean norm. For instance, if $f$ is $G$-Lipschitz with respect to $\ell_\infty$-norm, subgradient method may not retain its dimension-free rate [16]. A powerful extension of the subgradient method to remedy this issue is *mirror descent* [61, 9], which iterates as:

$$\nabla \Phi(y_{t+1}) = \nabla \Phi(x_t) - \eta \tilde{g}_{x_t}, \quad \tilde{g}_{x_t} \overset{\text{estimate}}{\sim} \text{Lemma 2.1} \quad \text{and} \quad x_{t+1} \in \underset{x \in \mathcal{X} \cap \mathcal{P}}{\arg\min} \, D_\Phi(x, y_{t+1}) \quad \text{(QMD)}$$

where $\Phi$ is called the mirror map (see Definition 4), whose gradient defines a bijection between the primal and the dual ambient spaces. Due to space limitations, we review mirror descent in more detail in Appendix B.

A technical challenge in analyzing the above algorithms is that we must handle the additional technicality that these algorithms actually input an erroneous subgradient at a point that is not the

---

[3]Here and throughout, the $\widetilde{\mathcal{O}}(\cdot)$ notation suppresses polylogarithmic factors in the usual $\mathcal{O}(\cdot)$.

Table 2: *State-of-the-art quantum and classical algorithms for SDP and LP.* For zero-sum games, set $r_p r_d = 1$ in LP complexities. Algorithms: Interior Point Method (IPM), Cutting Plane Method (CPM), Matrix Multiplicative Weights (MWU), Mirror Descent (MD), Primal-Dual Hybrid Gradient (PDHG), and Multiplicative Weight Update (MWU).

| SDP solvers | | LP solvers | |
|---|---|---|---|
| **Classical** | **Time complexity** | **Classical** | **Time complexity** |
| IPM [43] | $\widetilde{\mathcal{O}}\left(\sqrt{n}(mns + m^\omega + n^\omega)\right)$ | IPM [23, 77] | $\widetilde{\mathcal{O}}\left((m+n)^\omega\right)$ |
| CPM [44] | $\widetilde{\mathcal{O}}\left(m(mns + m^\omega + n^\omega)\right)$ | PDHG [5] | $\widetilde{\mathcal{O}}(\mathsf{Hoffman}(A,b,c) \cdot mn)$ |
| MWU [6, 76] | $\widetilde{\mathcal{O}}\left(mns\left(\frac{r_p r_d}{\varepsilon}\right)^4 + ns\left(\frac{r_p r_d}{\varepsilon}\right)^7\right)$ | Stochastic MD [37] | $\widetilde{\mathcal{O}}\left((m+n)\left(\frac{r_p r_d}{\varepsilon}\right)^2\right)$ |
| | | Variance-reduced SMD [18] | $\widetilde{\mathcal{O}}\left(mn + \sqrt{mn(m+n)}\left(\frac{r_p r_d}{\varepsilon}\right)\right)$ |
| **Quantum** | **Gate complexity** | **Quantum** | **Gate complexity** |
| QMWU [73] | $\widetilde{\mathcal{O}}\left(\sqrt{m}s\left(\frac{r_p r_d}{\varepsilon}\right)^4 + \sqrt{n}s\left(\frac{r_p r_d}{\varepsilon}\right)^5\right)$ | QMWU [11, 30] | $\widetilde{\mathcal{O}}\left(\sqrt{m+n}\left(\frac{r_p r_d}{\varepsilon}\right)^{2.5} + \left(\frac{r_p r_d}{\varepsilon}\right)^3\right)$ |
| **QMD (Thm 3.1)** | $\widetilde{\mathcal{O}}\left((mns + n^\omega)\left(\frac{r_p r_d}{\varepsilon}\right)^2\right)$ | QIPM [4] | $\widetilde{\mathcal{O}}\left(\sqrt{m}n^{9.5}\right)$ |
| | | **QMD (Thm 3.2)** | $\widetilde{\mathcal{O}}\left(m\sqrt{n}\left(\frac{r_p r_d}{\varepsilon}\right)^2\right)$ |

point queried, but instead a randomly chosen nearby point. This is due to a randomized smoothing procedure employed in [20, 75]. We show how to handle such errors in the robust analysis by leveraging the fact that the algorithms using subgradients average over their iterates. Below, we establish that both QPSM and QMD achieve a query complexity of $\widetilde{\mathcal{O}}((GR/\varepsilon)^2)$, matching the lower bound of classical first-order method [82, 64].

**Theorem 2.1** (Zeroth-order nonsmooth optimization, informal). *Let $\mathcal{X} \subseteq \mathbb{R}^d$ be a closed convex set with nonempty interior. Suppose $f : \mathbb{R}^d \to \mathbb{R}$ is convex and $G$-Lipschitz on $\mathcal{X}$ with respect to a given $p$-norm $\|\cdot\|_p$ and set $\varepsilon \in (0,1)$. There is a quantum algorithm $\mathcal{A}$ that solves* (OPT) *using $\widetilde{\mathcal{O}}((GR/\varepsilon)^2)$ queries to a noisy zeroth-order binary oracle $U_f^{(\theta)}$ (Definition 1) such that:*

- *Euclidean ($p = 2$): $\mathcal{A}$ is QPSM (Theorem B.3); $R = \|x_1 - x^\star\|_2$; $\theta = \mathcal{O}_{1/\varepsilon, d}\left(\varepsilon^5/d^{4.5}\right)$;*

- *non-Euclidean ($p \neq 2$): $\mathcal{A}$ is QMD (Theorem C.1); $R = \sqrt{D_\Phi(x^\star, x_1)}$; $\theta = \mathcal{O}_{1/\varepsilon, d}\left(\varepsilon^5/d^5\right)$.*

**Remark 1.** *Quantum dual averaging (Theorem C.2), a variant of mirror descent, also performs similarly in the non-Euclidean setting ($p \neq 2$). We provide a formal theorem characterizing the complexity of quantum dual averaging in the supplementary material.*

**Related work.** For black-box convex optimization under a noisy evaluation oracle, the most closely related work to ours is by Gong et al. [36] which analyzes quantum gradient methods in the context of nonconvex optimization, under a similar oracle model. Their analysis yields an algorithm for finding stationary points of an $L$-gradient Lipschitz (i.e., $L$-smooth), $\rho$-Hessian Lipschitz function using $\widetilde{\mathcal{O}}\left(1/\varepsilon^{1.75}\right)$ queries, each of which must be $\widetilde{\mathcal{O}}(\varepsilon^6/d^4)$ accurate. This algorithm can be applied to convex optimization as stationary points are global optima in this setting. Our results improve upon this simple application of [36] in three ways: *i)* our result above do not require Hessian or gradient Lipschitzness; *ii)* for functions with $L$-Lipschitz gradients, which we analyze in Section 4, we obtain a faster $\widetilde{\mathcal{O}}(1/\varepsilon)$ rate; and *iii)* a milder accuracy requirement of $\widetilde{\mathcal{O}}(\varepsilon^4/d^3)$ (Theorem 4.2).

There are also other works [36, 69, 54] that investigate convex optimization in the zeroth-order setting that focus on faster estimation of gradient estimators using quantum mean estimation. The focus in these papers is to improve the $\varepsilon$-dependence of classical algorithms and they do not achieve exponential speedups in terms of the dimension $d$. Finally, under a related but different setting of online convex optimization, [41] investigated a similar algorithm to the zeroth-order quantum subgradient method we analyze in Theorem B.3 for the Euclidean nonsmooth case, yet without taking the noisy evaluation oracle into account.

## 3 White-box setting: application to SDPs, LPs, and zero-sum games

**Semidefinite programming.** Define $r_p \geq 1$. Let $b \in \mathbb{R}^m$ be a vector with $\|b\|_\infty \leq r_p$, and $A_1, \ldots, A_m, C \in \mathbb{R}^{n \times n}$ be symmetric matrices with at most $s$ non-zero elements in any of their rows. We write the *primal* semidefinite program (SDP) as:

$$\sup_{X \in \mathbb{R}^{n \times n}} \{\operatorname{tr}(CX) : \operatorname{tr}(X) = r_p, \ \operatorname{tr}(A_i X) \leq b_i \text{ for all } i \in [m], \ X \succeq 0\}. \tag{P}$$

Here $\operatorname{tr}(\cdot)$ denotes the trace and $P \succeq 0$ indicates that $P$ is positive semidefinite. We make the standard assumption that the matrices $\{A_i\}_{i \in [m]}$ are linearly independent. Defining $A_0 := I$ and $b_0 := r_p$, the *dual* problem associated with (P) is given by

$$\inf_{(y_0, y) \in \mathbb{R} \times \mathbb{R}^m_{\geq 0}} \left\{ b_0 y_0 + b^\top y : y_0 I + \sum_{i \in [m]} y_i A_i - C \succeq 0 \right\}, \tag{D}$$

where $\mathbb{R}^m_{\geq 0}$ denotes the nonnegative orthant. We assume that the input matrices are normalized with respect to the operator norm $\|A_1\|_{\mathrm{op}}, \ldots, \|A_m\|_{\mathrm{op}}, \|C\|_{\mathrm{op}} \leq 1$, and that any primal-feasible solution satisfies $\operatorname{tr}(X) = r_p$. These assumptions are mild (and standard in the literature), since any nontrivial SDP with bounded feasible region can be brought into this form through projection, scaling and (possibly) the introduction of linear slack variables.

SDPs constitute a fundamental class of convex optimization problems due to their expressive power, capturing fundamental applications in control [12], information theory [68], machine learning [50, 80], finance [25, 81], and quantum information science [1, 29, 40, 79]. Both Linear Programming (LP) and Second-Order Conic Programming can be cast as special instances of SDP. Famously, SDP can be used to obtain approximate solutions to NP-Hard problems in polynomial time [55, 35].

SDPs have been known to be polynomial time solvable (to finite precision) since the pioneering work of Nesterov and Nemirovskii [65, 66] and Grötschel, Lovász and Schrijver [38]. The best performing algorithms for general SDPs (in theory and practice) are *interior point methods* (IPMs) [3, 59, 71, 43]. There are also algorithms for SDP based on first-order methods [6, 42, 17, 60].

Currently there are two classes of quantum SDP solvers with provable guarantees. One is comprised of *quantum* IPMs [8, 48], which replace the classical solution of the Newton linear system in interior point methods with a quantum linear systems algorithm (QLSA) [39, 22, 21]. The other class is comprised of *quantum matrix multiplicative weights update methods* [13, 15, 14, 76, 73], which recast trace-normalized positive semidefinite matrices as mixed quantum states that can be efficiently prepared using Gibbs sampling. Currently the state-of-the-art running time is due to van Apeldoorn and Gilyén [73], whose algorithm runs in time $\widetilde{\mathcal{O}}\left(\sqrt{m}s(r_p r_d/\varepsilon)^4 + \sqrt{n}s(r_p r_d/\varepsilon)^5\right)$, where $r_d \geq 1$ is an $\ell_1$-norm bound on dual optimal solutions $(y_0^\star, y^\star) \in \mathbb{R} \times \mathbb{R}^m_{\geq 0}$. Our main result on solving SDPs is the following theorem. Let $\omega \in [2, 2.38)$ be the exponent of matrix multiplication [23, 77].

We now describe an application of the algorithmic frameworks underlying our results on black-box nonsmooth convex optimization to white-box problems including SDPs, LPs, and zero-sum games. It can be shown that, upon performing the normalization $\frac{1}{r_p}(b_0, b) = (1, \tilde{b})$, the dual SDP in (D) can be equivalently reformulated as a nonsmooth convex optimization problem (see, e.g., [42, 70, 81]):

$$\min_{y \in \mathbb{R}^m_{\geq 0}} f(y) := \lambda_{\max}\left(C - \sum_{i \in [m]} y_i A_i\right) + \tilde{b}^\top y. \tag{Eigenvalue SDP}$$

This problem concerns the minimization of the largest eigenvalue of a symmetric matrix, and can be readily solved using the quantum mirror descent method (QMD) we developed in the previous section, as we detail below.

**Theorem 3.1** (SDP solver, informal version of Theorem D.2). *There is a quantum algorithm which solves* (P)-(D) *to precision* $\varepsilon \in (0, 1)$ *in time* $\widetilde{\mathcal{O}}((mns + n^\omega)(r_p r_d/\varepsilon)^2)$. *The output is an $\varepsilon$-optimal solution to the dual problem* (D). *That is, a vector* $(y_0^\star, y^\star) \in \mathbb{R} \times \mathbb{R}^m_{\geq 0}$ *satisfying*

$$y_0^\star I + \sum_{i \in [m]} y_i^\star A_i - C \succeq 0, \quad \text{and} \quad b_0 y_0^\star + b^\top y^\star \leq \mathsf{OPT} + \varepsilon,$$

*where* $\mathsf{OPT}$ *is the optimal objective value of the primal and dual SDPs in* (P)-(D).

The objective in (Eigenvalue SDP) is 2-Lipschitz with respect to the $\ell_1$-norm of $y$ (Lemma D.1). Therefore, we can apply the (zeroth-order) QMD to find an $\varepsilon$-precise approximation of the optimal objective value using $\widetilde{\mathcal{O}}((r_p r_d/\varepsilon)^2)$ queries to an evaluation oracle for $f(y)$. The rub is that our algorithms are sensitive to noise in the evaluation oracle, and so we are relegated to high-precision oracles, i.e., those with polylogarithmic dependence on the inverse accuracy to which we evaluate $f$. A straightforward implementation is to "classically" compute the inner product term and perform an eigendecomposition on $C - \sum_{i \in [m]} y_i A_i$ to determine its largest eigenvalue. Evaluating $f$ in this manner has cost $\mathcal{O}((mns + n^\omega) \log(1/\varepsilon))$, yielding a quantum SDP solver with gate complexity $\widetilde{\mathcal{O}}((mns + n^\omega)(r_p r_d/\varepsilon)^2)$.

**Remark 2.** *We compare our running times to state of the art quantum and classical algorithms for SDPs and LPs in Table 2. For dense SDPs with $m \geq n$ constraints, our SDP algorithm provides a quadratic speedup in $r_p r_d/\varepsilon$ over the classical matrix-multiplicative weights method of Arora and Kale [6]. We observe a similar enhancement over the quantum SDP solvers in [73], but their dependence on $m$ and $n$ is superior to ours.*

**Linear programming.**  Linear Programs (LPs) correspond to SDPs in which each of the input matrices is a diagonal matrix. The primal and dual LPs can be written as

$$\max_{\{x \in \mathbb{R}^n_{\geq 0} : \mathbf{1}_n^\top x = r_p, Ax \leq b\}} c^\top x \quad \text{and} \quad \min_{\{(y_0, y) \in \mathbb{R} \times \mathbb{R}^m_{\geq 0} : y_0 \mathbf{1}_n + A^\top y \geq c\}} r_p y_0 + b^\top y, \qquad \text{(LP)}$$

where $A \in \mathbb{R}^{m \times n}$, $c \in \mathbb{R}^n$ and $\mathbf{1}_n \in \mathbb{R}^n$ is the all-ones vector. LPs have been known to be polynomial-time solvable since the work of Khachiyan [49], and the first polynomial-time IPM for LP is due to Karmarkar [47].

Naturally, the formulation in (Eigenvalue SDP) becomes simpler for LPs, as the $\lambda_{\max}(\cdot)$ term simplifies to $\max_{j \in [n]} \{c_j - \langle A_j, y \rangle\}$, where $A_j$ is the $j$th column of $A$. Thus, one can evaluate $f$ in time $\widetilde{\mathcal{O}}(m\sqrt{n})$ using generalized quantum maximum finding [27, 76, 33]. This yields a quantum LP solver with gate complexity $\widetilde{\mathcal{O}}(m\sqrt{n}(r_p r_d/\varepsilon)^2)$ (and $\widetilde{\mathcal{O}}(m\sqrt{n}\varepsilon^{-2})$ complexity for zero-sum games.)

Due to their simpler structure, LPs admit more efficient algorithms than SDPs. Classical IPMs can solve LPs in matrix-multiplication time $\widetilde{\mathcal{O}}((m + n)^\omega)$ [24, 77]. Like the case of SDP, there are quantum IPMs that are based on QLSAs [48, 56–58]. More recent quantum algorithms accelerate the IPM framework without using QLSAs: Apers and Gribling [4] give a speedup for tall LPs with $m \gg n$, and Augustino et al. [7] solve LPs through quantum simulation of the central path. Our application of (QMD) to solving LPs results in the following.

**Theorem 3.2** (LP solver, informal version of Theorem D.3). *There is a quantum algorithm which solves (LP) to precision $\varepsilon \in (0, 1)$ in time $\widetilde{\mathcal{O}}(m\sqrt{n}(r_p r_d/\varepsilon)^2)$.*

**Remark 3.** *In the case of LP solving, the simplified structure of the objective function enables it to be evaluated quadratically faster by quantum amplitude amplification. We thus obtain an algorithm whose dependence on $m$ and $n$ compares favorably to other quantum and classical algorithms, but we cannot make decisive conclusions regarding an end-to-end speedup without making further assumptions on the behavior of the $r_p r_d/\varepsilon$ term.*

**Zero-sum games.**  There are also fast LP algorithms based on a reduction to *zero-sum games*, which are matrix games in which each player has a finite number of pure strategies. These problems are fundamental to computer science, economics and machine learning. A standard setup concerns two players Alice and Bob, whose action spaces are $[m]$ and $[n]$ respectively. Payoffs from the game are encoded in the entries of a matrix $A \in [-1, 1]^{m \times n}$. If Alice plays action $i \in [m]$ and Bob plays $j \in [n]$, then Alice obtains the payoff $A_{ij}$, while Bob receives a payoff of $-A_{ij}$. Each player aims to maximize their expected payoff through randomized strategies over the probability simplex $(x, y) \in \Delta^n \times \Delta^m$, giving rise to the minimax optimization problem:

$$\min_{x \in \Delta^n} \max_{y \in \Delta^m} y^\top A x. \qquad \text{(ZSG)}$$

Saddle points of (ZSG) are called *mixed Nash equilibria*, which always exist for zero-sum games due to von Neumann's minimax theorem [67].

Zero-sum games can be reformulated as LPs, and vice versa (they are LPs with $r_p r_d = \mathcal{O}(1)$). This relationship has led to the development of fast classical and quantum algorithms that obtain

large speedups in $m$ and $n$ over second-order methods like IPMs, at the cost of polynomial scaling in the inverse precision. A classical algorithm due to Grigoriadis and Khachiyan [37] can solve zero-sum games in time $\widetilde{\mathcal{O}}((m+n)\varepsilon^{-2})$, which implies an $\widetilde{\mathcal{O}}((m+n)(r_p r_d/\varepsilon)^2)$ algorithm for LP. Quantumly, the dependence on $m$ and $n$ can be improved quadratically, and a series of works [53, 74, 11, 30] have reduced the overall complexity of solving zero-sum games from $\widetilde{\mathcal{O}}(\sqrt{m+n}\varepsilon^{-4})$ to $\widetilde{\mathcal{O}}(\sqrt{m+n}\varepsilon^{-2.5} + \varepsilon^{-3})$. Our quantum algorithm for (ZSG) is as follows.

**Corollary 3.1** (Zero-sum games, informal version of Corollary D.1). *There is a quantum algorithm which solves* (ZSG) *to precision* $\varepsilon \in (0,1)$ *in time* $\widetilde{\mathcal{O}}(m\sqrt{n}(1/\varepsilon)^2)$.

Figure 1 illustrates the parameter regimes for which our algorithm provides outperforms the state of the art approaches for solving zero-sum games.

**Remark 4.** *It is interesting to note that previous quantum algorithms for zero-sum games [53, 74, 11, 30] improved the dependence on the leading term from $\sqrt{m+n}\varepsilon^{-4}$ to $\sqrt{m+n}\varepsilon^{-2.5}$ through the use of better Gibbs sampling techniques. In contrast, our algorithm makes $\widetilde{\mathcal{O}}(\varepsilon^{-2})$ queries, and does not incur any additional polynomial dependence on $1/\varepsilon$ because our per-iteration complexity boils down to the cost of evaluating the objective function, which we do with $\mathcal{O}(m\sqrt{n}\log(1/\varepsilon))$ cost.*

## 4 Zeroth-order smooth optimization

In Section 2, we studied zeroth-order convex optimization in a general form, where the only structure of the objective function $f$ we assumed is the Lipschitz continuity of $f$, measured both in $\ell_2$-space (quantum projected subgradient method) and in $\ell_p$-space (quantum mirror descent). We showed that the zeroth-order quantum algorithms match the first-order query complexities of the corresponding classical methods, and then applied the quantum mirror descent framework to white-box settings in Section 3, yielding the state-of-the-art results in some regimes.

A natural next question to ask is whether the zeroth-order quantum gradient methods can achieve faster convergence rates when the objective function $f$ exhibits additional structures. Motivated by this question, we now analyze quantum gradient methods in the case where the gradient $\nabla f$ exists and is also $L$-Lipschitz continuous. This condition is often referred to as $L$-smoothness in optimization literature, and the classical gradient descent enjoys an improved convergence rate.

We first need a quantum algorithm to estimate the gradient. The work of van Apeldoorn et al. introduced a "suppressed-bias" version of Jordan's quantum gradient estimation, wherein the expectation of the gradient to be controlled is independent of the accuracy to which the phase estimation step is carried out [72, Theorem 31]. We modify this gradient estimation protocol so that the variance is also controlled with noisy evaluation oracles (Corollary A.1), and hence the gradients of smooth functions can also be estimated efficiently (Theorem A.2). We state the informal version below.

**Theorem 4.1** (Gradient estimation, informal version of Theorem A.2). *Let $f : \mathbb{R}^d \to \mathbb{R}$ be $G$-Lipschitz with $L$-Lipschitz gradients, and let $\theta, \sigma$ be real positive parameters. Further, suppose we can query a $\theta$-approximate binary oracle with $\theta = \widetilde{\mathcal{O}}(\sigma^4/(Ld^2G^2))$. Then, there exists a procedure that outputs $k$ that satisfies $\mathbb{E}[\|k - \nabla f(y)\|_\infty^2] \leq \sigma^2$ and $\|\mathbb{E}[k] - \nabla f(y)\|_\infty \leq \sigma$ using $\widetilde{\mathcal{O}}(1)$ calls to the unitary $U_f^{(\theta)}$ (see Definition 1).*

Equipped with the gradient estimation above, the quantum gradient descent algorithm iterates as:

$$x_{t+1} = x_t - \eta g_{x_t}, \quad g_{x_t} \overset{\text{estimate}}{\sim} \text{Theorem } 4.1. \tag{QGD}$$

We also quantize mirror prox, a variant of mirror descent, to achieve similar rate for general $\ell_p$-spaces, similarly to the nonsmooth cases in Section 2. Quantum mirror prox iterates as:

$$
\begin{aligned}
\nabla\Phi(\bar{z}_{t+1}) &= \nabla\Phi(x_t) - \eta g_{x_t} & \nabla\Phi(\bar{x}_{t+1}) &= \nabla\Phi(x_t) - \eta g_{z_{t+1}} \\
z_{t+1} &\in \underset{x \in \mathcal{X} \cap \mathcal{P}}{\arg\min}\, D_\Phi(x, \bar{z}_{t+1}) \quad \text{and} & x_{t+1} &\in \underset{x \in \mathcal{X} \cap \mathcal{P}}{\arg\min}\, D_\Phi(x, \bar{x}_{t+1}),
\end{aligned} \tag{QMP}
$$

again with $g_{x_t}$ and $g_{z_t}$ estimated via Theorem 4.1. We now show that the zeroth-order quantum gradient descent and quantum mirror prox match the first-order rates of corresponding classical algorithms, $\widetilde{\mathcal{O}}(LR^2/\varepsilon)$, as follows.

**Theorem 4.2** (Zeroth-order smooth optimization, informal). *Let $\mathcal{X} \subseteq \mathbb{R}^d$ be a closed convex set with nonempty interior. Suppose $f : \mathbb{R}^d \to \mathbb{R}$ is convex, G-Lipschitz, and L-smooth on $\mathcal{X}$ with respect to a given p-norm $\|\cdot\|_p$ and set $\varepsilon \in (0, 1)$. There is a quantum algorithm $\mathcal{A}$ that solves* (OPT) *using $\widetilde{\mathcal{O}}(LR^2/\varepsilon)$ queries to a noisy zeroth-order binary oracle $U_f^{(\theta)}$ (Definition 1) such that:*

- *Euclidean ($p = 2$): $\mathcal{A}$ is QGD (Theorem B.2); $R = \max_{f(x) \leq f(x_0)} \|x - x^\star\|_2$; $\theta = \mathcal{O}_{1/\varepsilon, d}\left(\varepsilon^4/d^3\right)$*

- *non-Euclidean ($p \neq 2$): $\mathcal{A}$ is QMP (Theorem C.3) and $R = \sqrt{D_\Phi(x^\star, x_1)}$; $\theta = \mathcal{O}_{1/\varepsilon, d}\left(\varepsilon^4/d^4\right)$.*

The primary technical challenge in deriving the above result is that quantum gradient estimation algorithms are inexact and incur errors that depend on algorithmic parameters, which are themselves connected to the query complexity and bounds on the input precision. Our task therefore is to prove robust convergence theorems for all the settings considered here that accommodate errors of the form incurred by quantum gradient estimation. The convergence of gradient methods that use erroneous/inexact gradients has been well studied in optimization theory [26, 28] but we were unable to find preexisting robustness results that suffice directly for our purposes.

**Remark 5.** *As an illustration of the challenge, a recent paper [19] considers the optimization of a smooth and strongly convex function with condition number $\kappa$, and obtain an oracle complexity of $(1/\varepsilon)^\kappa$ due to a multiplicative accumulation of errors, which is much slower than classical gradient descent with exact gradients. In order to recover convergence rates that match classical gradient descent, we use proofs based on the classical analysis of stochastic gradient descent with controlled bias (c.f., Lemma B.1). As a result, our quantum gradient descent do not incur such multiplicative-error overheads and do in fact match the classical first-order rates.*

**Departing from convexity.** Finally, we also analyze the quantum gradient descent for $L$-smooth functions that, instead of convexity, satisfy the *Polyak-Łojasiewicz inequality*:

$$\|\nabla f(x)\|_2^2 \geq 2\mu \left(f(x) - f(x^\star)\right), \tag{PŁ}$$

which we refer to as the $\mu$-PŁ condition, following [46]. Note that the $\mu$-PŁ condition is implied by, and hence weaker than, $\mu$-strong convexity [46]. In fact, there exist nonconvex functions that satisfy the $\mu$-PŁ condition [78]. Therefore, the result below can be applied to strongly convex functions and attain the same rate up to constant factors, showing that the zeroth-order quantum gradient descent is efficient for minimizing (structured) nonconvex functions as well as strongly convex functions.

**Theorem 4.3** (Informal version of Theorem B.1). *Let $f : \mathbb{R}^d \to \mathbb{R}$ be G-Lipschitz, $\mu$-PŁ, and L-smooth. The zeroth-order quantum gradient descent* (QGD) *minimizes $f$ to accuracy $\varepsilon$ with high probability with $\widetilde{\Theta}(\kappa)$ queries to a $\theta$-approximate binary oracle of $f$ with $\theta = \mathcal{O}_{1/\varepsilon, d}\left(\varepsilon^2/d^3\right)$.*

## 5 Conclusion and discussions

In this work, we studied the zeroth-order convex optimization with quantum gradient methods. We demonstrated that, using only a noisy evaluation oracle, zeroth-order quantum gradient methods can match the first-order query complexities of the corresponding classical methods. These results exhibit exponential separations between quantum and classical zeroth-order convex optimization. In particular, we analyzed our framework both in the $\ell_2$-space (quantum (sub)gradient methods) as well as the $\ell_p$-space (quantum mirror descent methods). We then applied the quantum mirror descent framework to develop quantum algorithms for solving SDPs, LPs, and zero-sum games, which achieve the state-of-the-art complexity in some regimes. In short, this work provides improved quantum algorithms for zeroth-order convex optimization, both in black- and white-box settings.

*Limitations and future work.* The main limitation is that practical performances of the analyzed algorithms cannot be experimentally studied due to the lack of sufficiently powerful quantum hardware at the moment. Theoretically, our results generally match the first-order query complexities of classical gradient descent. For convex optimization, however, there exists faster (and in fact optimal) gradient methods due to Nesterov [62, 64], especially in the smooth setting. An interesting future work is whether the zeroth-order quantum gradient methods can match the accelerated convergence rates of the classical first-order methods. Moreover, the precision requirements are stringent to achieve the presented query complexities. As such, in the white-box setting, other quantum LP/SDP solvers can have better overall complexity in some regimes. Therefore, alleviating the precision requirement, as

well as developing more structure-aware zeroth-order quantum algorithms can be interesting future directions.

## Disclaimer

## Acknowledgements

The authors thank Rob Otter and Shaohan Hu for their support, and all their colleagues at Global Technology Applied Research center of JPMorganChase for helpful discussions and feedback. We are grateful to Sander Gribling for pointing out a bug in the evaluation oracle used for the LP solver in the first arXiv version of this paper, which has been fixed in all subsequent versions.

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

# Technical Appendices and Supplementary Material for "Fast Zeroth-Order Convex Optimization with Quantum Gradient Methods"

In this appendix, we provide more thorough preliminaries and background information that supplement the main text. We also state the formal version of the informal theorems from the main text, as well as their proofs. This appendix is organized as follows:

- In Section A, we establish notation and the class of functions we study in this work. We also briefly review the quantum protocols for (sub)gradient estimation.

- In Section B, we analyze the efficiency of zeroth-order quantum (sub)gradient methods instantiated with quantum (sub)gradient applied to black-box convex optimization. We consider three classes of functions $f$ that are well-behaved with respect to the Euclidean norm: *i)* convex and $G$-Lipschitz; *ii)* convex and $L$-smooth (i.e., gradients are $L$-Lipschitz); and *iii)* $L$-smooth and $\mu$-PL, which also implies $\mu$-strong convexity.

- In Section C, we study the zeroth-order quantum mirror descent and its variants, including quantum dual averaging and quantum mirror prox. This extends the quantum (sub)gradient methods from the previous section to non-Euclidean settings ($\ell_p$-space).

- In Section D, we leverage a connection between eigenvalue optimization and semidefinite programming to apply the quantum mirror descent framework to convex optimization in white-box settings. We analyze the complexity of this approach applied to solving semidefinite programs (SDPs), linear programs (LPs), and zero-sum games, and identify parameter regimes for which our algorithm outperforms the best-known algorithms for solving these problems.

## A  Preliminaries

### A.1  Notation

We let $[d] = \{1, \ldots, d\}$. For a finite-dimensional space $\mathcal{E}$, we denote its dual space by $\mathcal{E}^*$. If $\mathcal{E} \subset \mathbb{R}^d$ is equipped with an arbitrary inner product $\langle \cdot, \cdot \rangle$ and norm $\| \cdot \|$, the *dual norm* $\| \cdot \|_*$ is

$$\|g\|_* := \sup_{\{x \in \mathbb{R}^d : \|x\| \leq 1\}} \langle g, x \rangle.$$

From Hölder's inequality, one has

$$\langle x, y \rangle \leq \|x\| \|y\|_* \quad \forall (x, y) \in \mathbb{R}^d \times \mathbb{R}^d.$$

We let $\|\cdot\|_p$ denote the usual $\ell_p$-norm on $\mathbb{R}^d$ for $p \in [1, \infty]$:

$$\|v\|_p := \left( \sum_{i=1}^{d} |v_i|^p \right)^{1/p}.$$

At times, we also work with Schatten norms, which correspond to the $\ell_p$-norms of the singular values of Hermitian operators in a given Hilbert space. For example, the operator norm $\|\cdot\|_{\mathrm{op}}$ is the Schatten-$\infty$ norm, and the trace norm $\| \cdot \|_{\mathrm{tr}}$ is the Schatten-1 norm. Note that for $\ell_p$-norms, the dual norm is $\| \cdot \|_q$ where

$$1/p + 1/q = 1$$

Hence, the $\ell_2$-norm is self-dual, i.e., setting $\| \cdot \| = \| \cdot \|_2$, one has $\| \cdot \|_* = \| \cdot \|_2$. We define the (closed) $\ell_p$-ball of radius $R$ centered at $c \in \mathbb{R}^d$ as $\mathcal{B}_p^d(c, R) := \{x \in \mathbb{R}^n : \|x - c\|_p \leq R\}$.

## A.2 Convexity and smoothness

This work is concerned with convex functions $f : \mathbb{R}^n \to [-\infty, \infty]$ (taking extended-real values) which are *proper*, meaning $f(x) > -\infty$ for all $x$ and $f(y) < \infty$ for some $y$. For such functions a *subgradient* may be defined.

**Definition 2** (Subgradient). *Let $f : \mathbb{R}^n \to (-\infty, \infty]$ be a proper convex function and let $x \in \mathrm{dom}(f)$. We say that $g \in \mathcal{X}^*$ is a subgradient of $f$ at $x$ if*

$$f(\bar{x}) \geq f(x) + \langle g, \bar{x} - x \rangle \quad \forall \bar{x} \in \mathcal{X}. \tag{1}$$

Equation (1) defines the *subgradient inequality*, which asserts that the first-order approximation of a convex function $f$ serves as a global lower bound. Subgradients extend the concept of differentiability to functions that are nondifferentiable: when $f$ is differentiable, the subgradient is simply the gradient. The set of all subgradients is called the *subdifferential* of $f$ at $x$, which we denote by $\partial f(x)$.

A function $f : \mathcal{X} \to \mathbb{R}$ is *G-Lipschitz* with respect to $\| \cdot \|$ on $\mathcal{X}$ if

$$\|g\|_* \leq G, \quad \forall x \in \mathcal{X}, g \in \partial f(x).$$

We say that $f$ is *L-smooth* with respect to $\| \cdot \|$ if the gradients $\nabla f$ are *L*-Lipschitz continuous:

$$\|\nabla f(x) - \nabla f(\bar{x})\|_* \leq L \|x - \bar{x}\| \quad \forall (x, \bar{x}) \in \mathcal{X} \times \mathcal{X}.$$

Note that any twice-continuously differentiable function is *L*-smooth with respect to $\| \cdot \|$ on $\mathbb{R}^d$ if and only if

$$\langle \nabla^2 f(x) z, z \rangle \leq L \|z\|^2 \quad \text{for all } x, z \in \mathbb{R}^d.$$

Smoothness assumptions provide a global quadratic upper bound on the function around a point $x$:

$$f(\bar{x}) \leq f(x) + \langle \nabla f(x), \bar{x} - x \rangle + \frac{L}{2} \|\bar{x} - x\|^2.$$

A function $f$ is *$\mu$-strongly convex* on $\mathcal{X}$ if there exists a $\mu > 0$ such that

$$f(\bar{x}) \geq f(x) + \langle \nabla f(x), \bar{x} - x \rangle + \frac{\mu}{2} \|\bar{x} - x\|^2.$$

Collecting these facts, when $\| \cdot \| = \| \cdot \|_2$ is the Euclidean norm, smoothness implies

$$\nabla^2 f(x) \preceq LI$$

and strong convexity implies

$$\nabla^2 f(x) \succeq \mu I$$

for all $x \in \mathcal{X}$. The ratio $\kappa = \frac{L}{\mu}$ is called the *condition number* of $f$.

The *Fenchel conjugate* $f^* : \mathbb{R}^d \to \mathrm{cl}(\mathbb{R})$ of a function $f$ is defined by

$$f^*(y) := \sup_{x \in \mathbb{R}^d} \{ \langle x, y \rangle - f(x) \} \quad y \in \mathbb{R}^d.$$

The conjugate function is defined as the point-wise supremum of affine functions of $y$, and is therefore convex by definition. When $f$ is convex and differentiable, $f^*$ is the *Legendre transform* of $f$.

The conjugate function admits a useful identity: the inverse operator for the gradient of $f$ is the gradient of the conjugate:

$$\nabla f^* = (\nabla f)^{-1}.$$

## A.3 Quantum gradient estimation

**Gradients with suppressed bias.** We will use the "low-bias" version of Jordan's protocol for quantum gradient estimation restated below, which were introduced by [72] in the context of state tomography. Such protocols allow for the expectation value of the gradient estimate to be made arbitrarily close to the true value, independent of the protocol accuracy itself.

**Theorem A.1** (Suppressed-bias gradient estimation ([72, Theorem 31])). *Let $\sigma', \delta \in \left(0, \frac{1}{6}\right]$ and $g \in \mathbb{R}^d$ such that $\|g\|_\infty \leq \frac{1}{3}$. Let $b := \lceil \log_2\left(\frac{2}{\sigma'}\right) \rceil$ and $B := 2^b$. If*

$$\left\| |\psi\rangle - \frac{1}{\sqrt{B^d}} \sum_{x \in \mathcal{G}_b^d} \exp\left(2\pi i \langle g, x \rangle\right) |x\rangle \right\|_2 \leq \frac{\delta}{24\lceil \ln(6d/\delta) \rceil + 3},$$

*given access to $8\lceil \ln(6d/\delta) \rceil + 1$ copies of $|\psi\rangle$, we can compute $k \in \left[-\frac{1}{2}, \frac{1}{2}\right]^d$ satisfying*

$$\Pr[\|k - g\|_\infty > \sigma'] \leq \delta \quad \text{and} \quad \|\mathbb{E}[k] - g\|_\infty \leq \delta.$$

*The procedure has a gate complexity of $\mathcal{O}\left(d \log(\frac{d}{\delta}) \log(\frac{1}{\sigma'}) \log(\frac{d}{\delta} \log(\frac{1}{\sigma'}))\right)$ and requires a corresponding circuit depth of $\mathcal{O}\left(\log(\frac{1}{\sigma'}) \log(\frac{d}{\delta} \log(\frac{1}{\sigma'}))\right)$.*

We utilize the suppressed-bias gradient estimation subroutine above, which we modify such that the variance is also controlled with noisy evaluation oracles. We start with the following corollary:

**Corollary A.1** (Suppressed-bias, low-variance gradient estimation). *Let $\sigma \in \left(0, \frac{1}{3}\right]$ and $g \in \mathbb{R}^d$ such that $\|g\|_\infty \leq \frac{1}{3}$. Let $b := \lceil \log_2\left(4/\sigma\right) \rceil$ and $B := 2^b$. If*

$$\left\| |\psi\rangle - \frac{1}{\sqrt{B^d}} \sum_{x \in \mathcal{G}_b^d} \exp\left(2\pi i \langle g, x \rangle\right) |x\rangle \right\|_2 \leq \frac{\sigma^2}{32\lceil \ln(8d/\sigma^2) \rceil + 4},$$

*given access to $8\lceil \ln(8d/\sigma^2) \rceil + 1$ copies of $|\psi\rangle$, we can compute $k \in \left[-\frac{1}{2}, \frac{1}{2}\right]^d$ satisfying*

$$\mathbb{E}[\|k - g\|_\infty^2] \leq \sigma^2 \quad \text{and} \quad \|\mathbb{E}[k] - g\|_\infty \leq \sigma.$$

*The procedure has a gate complexity $\mathcal{O}\left(d \log^2(\sqrt{d}/\sigma) \log(1/\sigma)\right)$ and requires a corresponding circuit depth of $\mathcal{O}\left(\log(\sqrt{d}/\sigma) \log(1/\sigma)\right)$.*

*Proof.* Define $\sigma \in \left(0, \frac{1}{3}\right]$. We use the procedure in Theorem A.1 with parameters $\sigma' = \sigma/2$, $\delta = 3\sigma^2/4$. The number of copies, gate complexity, and circuit depth follow by direct substitution with minor simplifications. It remains to calculate the variance of $k - g$. Noting that by assumption and the guarantee on $k$, it holds that $\|k - g\|_\infty < 1$, we observe

$$\mathbb{E}[\|k - g\|_\infty^2] \leq \frac{\sigma^2}{4} + \Pr[\|k - g\|_\infty > \sigma/2] \leq \sigma^2,$$

where the probability is bounded due to the guarantees of Theorem A.1. This completes the proof. □

Using the above corollary, we can estimate the gradient of smooth functions with noisy oracles. The quantum gradient descent algorithm we analyze in Theorems B.1 and B.2 will therefore invoke the following result at each iterate. In the Euclidean case in Section B, note that $\vartheta = \vartheta_* = \sqrt{d}$ in the below statement.

**Theorem A.2** (Suppressed-Bias Gradient Estimation of Smooth Functions with Noisy Oracles). *Let $f : \mathbb{R}^d \to \mathbb{R}$ be a $G$-Lipschitz function with $L$-Lipschitz gradients in a $p \geq 1$ norm $\|\cdot\|$, and let $\theta, \sigma$ be real positive parameters, $\sigma/G \leq 1$. Let $1 \leq \vartheta, \vartheta_* \leq d$ be such that*

$$\|x\| \leq \vartheta \|x\|_\infty, \forall x \in \mathbb{R}^d$$
$$\|x\|_* \leq \vartheta_* \|x\|_\infty, \forall x \in \mathbb{R}^d,$$

*satisfying $\vartheta \vartheta_* \leq d$. Define $r := \frac{\sqrt{2\theta}}{\sqrt{L}\vartheta}$, and let $\tilde{f}$ be an $\theta$-approximation for $f$ in $\mathcal{B}_\infty^d(y, r)$ for some $y \in \mathbb{R}^d$. Then, if*

$$\theta = \mathcal{O}\left(\frac{\sigma^4}{L\vartheta^2 G^2 \log^2\left(dG^2/\sigma^2\right)}\right),$$

*there exists a procedure that outputs $k$ that satisfies*

$$\mathbb{E}[\|k - \nabla f(y)\|_*^2] \leq (\vartheta_* \sigma)^2 \quad \text{and} \quad \|\mathbb{E}[k] - \nabla f(y)\|_* \leq \vartheta_* \sigma,$$

*using $8\lceil \ln(72dG^2/\sigma^2)\rceil + 1$ calls to the unitary $U_{\tilde{f}} = \sum_{x \in \mathcal{G}_b^d} \exp\left(\frac{2\pi i \tilde{f}(y+rx)}{3Gr}\right)|x\rangle\langle x|$. The procedure has a gate complexity $\mathcal{O}\left(d\log^2(\sqrt{d}G/\sigma)\log(G/\sigma)\right)$ and circuit depth of $\mathcal{O}\left(\log(\sqrt{d}G/\sigma)\log(G/\sigma)\right)$. The unitary $U_{\tilde{f}}$ can be implemented using $\mathcal{O}(1)$ queries to a binary oracle for $\tilde{f}$, or $\mathcal{O}\left(R_0\sqrt{dL}/\sqrt{\theta}\right)$ to a $(GR_0)$-phase oracle where $R_0$ is a length scale chosen to ensure that for all $x \in \mathcal{B}_\infty^d(y,r), f(x) \leq GR_0$.*

*Proof.* First, note that by assumption for arbitrary $p$-norm $\|\cdot\|$ with $p \geq 1$, we have

$$|f(x) - f(y)| \leq G\|x - y\| \leq G\vartheta\|x - y\|_\infty,$$

and

$$|f(x) - f(y) - \langle \nabla f(x), y - x\rangle| \leq \frac{L}{2}\|x - y\|^2 \leq \frac{\vartheta^2 L}{2}\|x - y\|_\infty^2.$$

It suffices to show how to compute $\nabla f(0)$ by a simple shift of co-ordinates, and by a similar argument we assume without loss of generality that $f(0) = 0$. First, note that since $\tilde{f}$ is the $\theta$-approximate version of $f$ and the gradient $\nabla f$ is $L$-Lipschitzness, we have for all $x \in \mathcal{G}_b^d$:

$$\left|\frac{\tilde{f}(rx)}{3Gr} - \frac{f(rx)}{3Gr} + \frac{f(rx)}{3Gr} - \left\langle \frac{\nabla f(0)}{3G}, x\right\rangle\right| \leq \left|\frac{\tilde{f}(rx)}{3Gr} - \frac{f(rx)}{3Gr}\right| + \left|\frac{f(rx)}{3Gr} - \left\langle \frac{\nabla f(0)}{3G}, x\right\rangle\right|$$

$$\leq \frac{\theta}{3Gr} + \frac{Lr\vartheta^2}{6G}$$

$$\leq \frac{\sqrt{2\theta L}\vartheta}{3G}. \tag{2}$$

Now, let $|\psi\rangle := \frac{1}{\sqrt{B^d}}\sum_{x \in \mathcal{G}_b^d}\exp\left(\frac{2\pi i \tilde{f}(rx)}{3Gr}\right)|x\rangle$. Using the above estimates and the inequality $|e^{ia} - e^{ib}| \leq |a - b|$, it follows that

$$\left\|\,|\psi\rangle - \frac{1}{\sqrt{B^d}}\sum_{x \in \mathcal{G}_b^d}\exp\left(2\pi i\left\langle\frac{\nabla f(0)}{3G}, x\right\rangle\right)|x\rangle\right\|_2 \leq 2\pi \max_{x \in \mathcal{G}_b^d}\left|\frac{\tilde{f}(rx)}{3Gr} - \left\langle\frac{\nabla f(0)}{3G}, x\right\rangle\right|$$

$$\overset{Eq.(2)}{\leq} 2\pi\frac{\sqrt{2\theta L}\vartheta}{3G}$$

$$\leq \frac{(\sigma/3G)^2}{32\lceil\ln(8d/(\sigma/3G)^2)\rceil + 4},$$

where the last line follows from our choice of $\theta$ in the statement. Since $f$ is $G$-Lipschitz (in $\ell_p$-norm, $p \geq 1$), we have $\|\nabla f(0)/3G\|_\infty \leq 1/3$, and we can now apply Corollary A.1 with $\sigma \to \sigma/3G$ to obtain an estimate $k'$ satisfying,

$$\mathbb{E}\left[\left\|k' - \frac{\nabla f(0)}{3G}\right\|_*^2\right] \leq (\vartheta_*\sigma/3G)^2,$$

$$\text{and} \quad \left\|\mathbb{E}[k'] - \frac{\nabla f(0)}{3G}\right\|_* \leq \vartheta_*\sigma/3G.$$

Clearly, $k = 3Gk'$ is an estimate satisfying our desired properties. The number of copies of $|\psi\rangle$ required, the gate complexity, and the circuit depth can be determined by direct substitution. It is evident that each copy of $|\psi\rangle$ can be prepared using a single call to $U_f$, resulting in the desired oracle complexities. $\square$

**Quantum subgradients.** In the case of minimizing convex and $G$-Lipschitz functions, which can be nonsmooth, the gradients might not be available. In that case, we use the subgradient estimation Lemma 2.1, which is based on [75]; we slightly modify for our purposes, which we prove below. Note that one can perform a similar analysis with [20, Lemma 2.4].

**Lemma A.1** (Subgradient Estimation in arbitrary $\ell_p$-norms). *Let $r_1 > 0$, $G > 0$, $\rho \in (0, 1/3]$, and suppose $\theta \in (0, r_1 dG/\rho]$. Suppose $f : \mathbb{R}^d \to \mathbb{R}$ is a convex function that is $G$-Lipschitz with respect to a given p-norm $\|\cdot\|$ with $p \geq 1$, and we have quantum query access to $\tilde{f}$, which is a $\theta$-approximate version of $f$, via a unitary $U$ over a (fine-enough) hypergrid of $\mathcal{B}_\infty(x, 2r_1)$. Then we can compute an approximate subgradient $\tilde{g} \in \mathbb{R}^d$ at $x$ using $\mathcal{O}(\log(d/\rho))$ queries to $U$ and $U^\dagger$, such that with probability $\geq 1 - \rho$, we have*

$$f(q) \geq f(x) + \langle \tilde{g}, q - x \rangle - 23^2 \vartheta_* \sqrt{\frac{\theta d^3 G}{\rho r_1}} \|q - x\| - 2G\vartheta r_1, \tag{3}$$

*where $1 \leq \vartheta, \vartheta^* \leq d$, and $\vartheta\vartheta_* \leq d$.*

*Proof.* Let $r_2 := \sqrt{\frac{\theta r_1 \rho}{dG}}$ and note that $r_2 \leq r_1$. The quantum algorithm chooses a uniformly random $z \in B_\infty(0, r_1)$ and applies Jordan's quantum algorithm to compute an approximate gradient at $z$ by approximately evaluating $f$ in superposition over a discrete hypergrid in $B_\infty(z, r_2/d)$.

Since $B_\infty(z, r_2/d) \subseteq B_1(z, r_2)$, [75, Lemma 17] implies

$$\sup_{y \in B_\infty(0, r_2/d)} \left| f(z + y) - f(z) - \left\langle y, \nabla^{(r_2)} f(z) \right\rangle \right| \leq \frac{r_2^2 \Delta^{(r_2)} f(z)}{2}. \tag{4}$$

Also as shown by [75, Lemma 11] and Markov's inequality we have

$$\Delta^{(r_2)} f(z) \leq \frac{2dG}{\rho r_1} \tag{5}$$

with probability $\geq 1 - \rho/2$ over the choice of $z$. If $z$ is such that Equation (5) holds, then we get

$$\sup_{y \in B_\infty(0, r_2/d)} \left| f(z + y) - f(z) - \left\langle y \nabla^{(r_2)} f(z) \right\rangle \right| \leq \frac{dG r_2^2}{\rho r_1} = \theta.$$

Also we have by Definition 1

$$\|f - \tilde{f}\|_\infty \leq \theta.$$

Now apply the quantum algorithm of [75, Corollary 15] with $r = 2r_2/d$, $c = f(z)$, $g = \nabla^{(r_2)} f(z)$, and $B = Gr$. This uses $\mathcal{O}\left(\log(d/\rho)\right)$ queries to $U$ and $U^\dagger$, and with probability $\geq 1 - \rho/2$ computes an approximate gradient $\tilde{g}$ such that

$$\left\| \nabla^{(r_2)} f(z) - \tilde{g} \right\|_\infty \leq \frac{8 \cdot 42\pi d}{2r_2} \cdot \theta = 4 \cdot 42 \cdot \pi \sqrt{\frac{\theta d^3 G}{\rho r_1}}. \tag{6}$$

Also, if $z$ is such that Equation (5) holds, then by [75, Lemma 10] we get that

$$\sup_{g \in \partial f(z)} \left\| \nabla^{(r_2)} f(z) - g \right\|_1 \leq \frac{r_2 \Delta^{(r_2)} f(z)}{2} \leq \frac{dG r_2}{\rho r_1} = \sqrt{\frac{\theta dG}{\rho r_1}},$$

and therefore by the triangle inequality and Equation (6) we get that

$$\begin{aligned}
\sup_{g \in \partial f(z)} \|g - \tilde{g}\|_\infty &\leq \sup_{g \in \partial f(z)} \left\| g - \nabla^{(r_2)} f(z) \right\|_\infty + \left\| \nabla^{(r_2)} f(z) - \tilde{g} \right\|_\infty \\
&\leq \sup_{g \in \partial f(z)} \left\| g - \nabla^{(r_2)} f(z) \right\|_1 + \left\| \nabla^{(r_2)} f(z) - \tilde{g} \right\|_\infty \\
&\leq \sqrt{\frac{\theta dG}{\rho r_1}} + 4 \cdot 42 \cdot \pi \sqrt{\frac{\theta d^3 G}{\rho r_1}} \\
&< 23^2 \sqrt{\frac{\theta d^3 G}{\rho r_1}}.
\end{aligned} \tag{7}$$

Hence,

$$\sup_{g \in \partial f(z)} \|g - \tilde{g}\|_* < 23^2 \vartheta_* \sqrt{\frac{\theta d^3 G}{\rho r_1}}. \tag{8}$$

Thus with probability at least $1 - \rho$, for all $y \in \operatorname{dom} f$ and for all $g \in \partial f(z)$ we have that

$$
\begin{aligned}
f(y) &\geq f(z) + \langle g, y - z \rangle \\
&= f(0) + \langle \tilde{g}, y \rangle + \langle g - \tilde{g}, y \rangle + (f(z) - f(0)) + \langle g, -z \rangle \\
&\geq f(0) + \langle \tilde{g}, y \rangle - |\langle g - \tilde{g}, y \rangle| - G\|z\| - \|g\|_* \|z\| \\
&\geq f(0) + \langle \tilde{g}, y \rangle - \|g - \tilde{g}\|_* \|y\| - 2G\|z\| \\
&\geq f(0) + \langle \tilde{g}, y \rangle - 23^2 \vartheta_* \sqrt{\frac{\theta d^3 G}{\rho r_1}} \|y\| - 2G\vartheta r_1.
\end{aligned}
$$

$\square$

# B  Convex optimization with quantum (sub)gradient estimation

## B.1  Zeroth-order quantum gradient descent

In order to analyze quantum gradient descent, we need a quantum algorithm for gradient estimation. As mentioned in the introduction, a core challenge is that quantum gradient estimation algorithms incur errors that depend on algorithmic parameters, which have to be handled carefully to achieve compelling convergence rates and query complexities.

### B.1.1  Strongly convex functions with Lipschitz gradients

We first analyze the quantum gradient descent for functions with $L$-Lipschitz gradients (also known as $L$-smooth functions in optimization literature). We further assume that $f$ satisfies the *Polyak-Łojasiewicz inequality*:

$$\|\nabla f(x)\|_2^2 \geq 2\mu\left(f(x) - f(x^\star)\right), \tag{PŁ}$$

which hereafter we refer to as the $\mu$-PŁ condition, following [46]. Note that the $\mu$-PŁ condition is implied by, and hence weaker than, $\mu$-strong convexity [46]. [4] Therefore, the results we present below can be applied to strongly convex functions, and attain the same rate up to constant factors.

Before analyzing the quantum gradient descent equipped with Theorem A.2, we first state a descent lemma for gradient descent with bias and stochasticity, which are the main ingredients for proving Theorems B.1 and B.2. The proof largely follows [2], modified appropriately for our setting.

**Lemma B.1** (Descent lemma of stochastic gradient descent with bias for smooth functions). *Suppose $f \colon \mathbb{R}^d \to \mathbb{R}$ has $L$-Lipschitz gradients. Consider the biased stochastic gradient descent algorithm $x_{t+1} = x_t - \eta g_t$, where $g_t = \nabla f(x_t) + b_t + n_t$ such that*

$$\mathbb{E}_t[g_t] = \nabla f(x_t) + b_t \quad (\Leftrightarrow \mathbb{E}_t[n_t] = 0) \quad and \quad \mathbb{E}_t[\|g_t - \mathbb{E}_t[g_t]\|_2^2] := \sigma_v^2, \quad \forall t,$$

*where $\mathbb{E}_t[\cdot] := \mathbb{E}\left[\cdot \,|\, x_t\right]$ denotes expectation conditional on $x_t$. Then, for any step size $\eta \leq \frac{1}{L}$, the following is satisfied:*

$$\mathbb{E}_t\left[f(x_{t+1}) - f(x_t)\right] \leq -\frac{\eta}{2}\|\nabla f(x_t)\|_2^2 + \frac{\eta}{2}\|b_t\|_2^2 + \frac{\eta^2 L}{2}\sigma_v^2.$$

*Proof.* We start with the $L$-smooth inequality [64, Equation 2.1.6]:

$$f(y) \leq f(x) + \langle \nabla f(x), y - x \rangle + \frac{L}{2}\|y - x\|_2^2. \tag{9}$$

---

[4]In fact, there exist nonconvex functions that satisfy the $\mu$-PŁ condition [78].

Invoking the above with $y \leftarrow x_{t+1}$ and $x \leftarrow x_t$, we have

$$f(x_{t+1}) \leq f(x_t) + \langle \nabla f(x_t), x_{t+1} - x_t \rangle + \frac{L}{2}\|x_{t+1} - x_t\|_2^2$$

$$= f(x_t) - \eta \langle \nabla f(x_t), g_t \rangle + \frac{L\eta^2}{2}\|g_t\|_2^2$$

$$= f(x_t) - \eta \langle \nabla f(x_t), g_t \rangle + \frac{L\eta^2}{2}\|g_t - \mathbb{E}_t[g_t] + \mathbb{E}_t[g_t]\|_2^2$$

$$= f(x_t) - \eta \langle \nabla f(x_t), g_t \rangle + \frac{L\eta^2}{2} \left( \|g_t - \mathbb{E}_t[g_t]\|_2^2 + \|\mathbb{E}_t[g_t]\|_2^2 + 2 \langle g_t - \mathbb{E}_t[g_t], \mathbb{E}_t[g_t] \rangle \right).$$

Taking expectations conditional on $x_t$, we have

$$\mathbb{E}_t[f(x_{t+1})] \leq f(x_t) - \eta \langle \nabla f(x_t), \mathbb{E}_t[g_t] \rangle + \frac{L\eta^2}{2}\mathbb{E}_t[\|g_t - \mathbb{E}_t[g_t]\|_2^2] + \mathbb{E}_t[\|\mathbb{E}_t[g_t]\|_2^2]$$

$$= f(x_t) - \eta \langle \nabla f(x_t), \nabla f(x_t) + b_t \rangle + \frac{L\eta^2}{2} \left( \mathbb{E}_t[\|g_t - \mathbb{E}_t[g_t]\|_2^2] + \mathbb{E}_t[\|\nabla f(x_t) + b_t\|_2^2] \right)$$

$$\leq f(x_t) - \eta \langle \nabla f(x_t), \nabla f(x_t) + b_t \rangle + \frac{L\eta^2}{2}\sigma_v^2 + \frac{L\eta^2}{2}\|\nabla f(x_t) + b_t\|_2^2.$$

Choose $\eta \leq \frac{1}{L}$, then we have

$$\mathbb{E}_t\left[f(x_{t+1})\right] \leq f(x_t) - \eta \langle \nabla f(x_t), \nabla f(x_t) + b_t \rangle + \frac{L\eta^2}{2}\sigma_v^2 + \frac{\eta}{2}\|\nabla f(x_t) + b_t\|_2^2$$

$$= f(x_t) + \frac{\eta}{2} \left( -2 \langle \nabla f(x_t), \nabla f(x_t) + b_t \rangle + \|\nabla f(x_t) + b_t\|_2^2 \right) + \frac{L\eta^2}{2}\sigma_v^2$$

$$= f(x_t) + \frac{\eta}{2} \left( -\|\nabla f(x_t)\|_2^2 + \|b_t\|_2^2 \right) + \frac{L\eta^2}{2}\sigma_v^2,$$

where in the last equality we used the following identity:

$$-2 \langle a, a + b \rangle + \|a + b\|_2^2 = -\|a\|_2^2 + \|b\|_2^2.$$

$\square$

**Lemma B.2** (Stochastic gradient descent with bias for $\mu$-PŁ & smooth functions)**.** *Consider the biased stochastic gradient estimator described in Lemma B.1. Assume $f \colon \mathbb{R}^d \to \mathbb{R}$ is $\mu$-PŁ as in* (PŁ)*, which implies $\mu$-strong convexity. Then, after running $T$ iterations of stochastic gradient descent, the following is satisfied:*

$$\mathbb{E}[f(x_T)] - f(x^\star) \leq (1 - \eta\mu)^T (f(x_0) - f(x^\star)) + \frac{\|b\|_2^2}{2\mu} + \frac{\eta L}{2\mu}\sigma_v^2.$$

*Proof.* First observe that by Lemma B.1 and (PŁ), we have

$$\mathbb{E}_t[f(x_{t+1})] - f(x^\star) + f(x^\star) - f(x_t) \leq -\frac{\eta}{2}\|\nabla f(x_t)\|_2^2 + \frac{\eta}{2}\|b_t\|_2^2 + \frac{\eta^2 L}{2}\sigma_v^2$$

$$\leq -\eta\mu(f(x_t) - f(x^\star)) + \frac{\eta}{2}\|b_t\|_2^2 + \frac{\eta^2 L}{2}\sigma_v^2$$

$$\implies \mathbb{E}_t[f(x_{t+1})] - f(x^\star) \leq (1 - \eta\mu)(f(x_t) - f(x^\star)) + \frac{\eta}{2}\|b_t\|_2^2 + \frac{\eta^2 L}{2}\sigma_v^2.$$

Assuming $\|b_t\|_2^2 = \|b\|_2^2$ for all $t$ (due to quanutm gradient estimation, we can ensure all $b_t$ are below some bound which we denote $\|b\|$), and unfolding for $T$ iterations, and using the tower law, we have

$$\mathbb{E}[f(x_T)] - f(x^\star) \leq (1 - \eta\mu)^T (f(x_0) - f(x^\star)) + \sum_{k=0}^{T-1} (1 - \eta\mu)^k \left( \frac{\eta}{2}\|b\|_2^2 + \frac{\eta^2 L}{2}\sigma_v^2 \right)$$

$$\leq (1 - \eta\mu)^T (f(x_0) - f(x^\star)) + \frac{\|b\|_2^2}{2\mu} + \frac{\eta L}{2\mu}\sigma_v^2,$$

where in the last step we used

$$\sum_{k=0}^{T-1}(1-c)^k \le \sum_{k=0}^{\infty}(1-c)^k = \frac{1}{c} \quad \text{for} \quad c \in (0,1).$$

$\square$

We are now ready to analyze the convergence of quantum gradient descent for $\mu$-PŁ functions with $L$-Lipschitz gradients.

**Theorem B.1** (Quantum gradient descent for $\mu$-PŁ & smooth functions). *Suppose $f\colon \mathbb{R}^d \to \mathbb{R}$ is $G$-Lipschitz, $L$-smooth, and $\mu$-PŁ, which also implies $\mu$-strong convexity. Consider the (zeroth-order) quantum gradient descent $x_{t+1} = x_t - \eta g_t$ with step size $\eta = 1/L$ that outputs the last iterate $x_T$ after running $T = \mathcal{O}(\kappa \log 1/\varepsilon)$ iterations, where the (biased stochastic) gradient $g_t$ is estimated via the quantum gradient estimation in Theorem A.2 on each iteration. Let $\varepsilon_0 := \varepsilon\mu/G^2$ and $\kappa := L/\mu$. Assume that $d/\varepsilon_0 \ge 1/5$, and one has access to $\theta$-evaluation oracle satisfying*

$$\theta \le \frac{\varepsilon^2 \mu^2}{450 d^3 G^2 L \left(32\left\lceil \ln\left(360 d^2 G^2/(\varepsilon\mu)\right)\right\rceil + 4\right)^2} = \Theta\left(\frac{\varepsilon \cdot \varepsilon_0}{d^3 \kappa \log(d^2/\varepsilon_0)^2}\right).$$

*Then, with probability at least $2/3$, one can obtain an $\varepsilon$-approximate solution to* (OPT) *with*

$$\widetilde{\Theta}\left(\kappa \log\left(d^2/\varepsilon_0\right)\right)$$

*queries to an $\theta$-approximate binary oracle of $f$. The procedure has a gate complexity $\mathcal{O}\left(d \log^2(d^2/\varepsilon_0) \log(d/\varepsilon_0)\right)$, and corresponding circuit depth of $\mathcal{O}\left(\log(d^2/\varepsilon_0) \log(d/\varepsilon_0)\right)$.*

*With $(GR_0)$-phase oracle access instead, the above complexities must be multiplied by a factor of $\mathcal{O}\left(R_0\sqrt{dL}/\sqrt{\theta}\right)$, where $R_0$ is a length scale chosen to ensure that for all $x \in \mathcal{B}_\infty^d(y, r), f(x) \le GR_0$.*

*Proof.* We start from Lemma B.2, where we plug in the quantum gradient estimation from Theorem A.2 (with $\vartheta = \vartheta_* = \sqrt{d}$). First, for the bias term involving $\|b\|^2$, we have

$$\|b_t\|_2^2 = \|\mathbb{E}_t[g_t] - \nabla f(x_t)\|_2^2 \le d\|\mathbb{E}_t[g_t] - \nabla f(x_t)\|_\infty^2 \le d\sigma^2, \quad \forall t.$$

Next, for the variance term involving $\sigma_v^2$, we have

$$\begin{aligned}
\sigma_v^2 := \mathbb{E}_t[\|g_t - \mathbb{E}_t[g_t]\|_2^2] &\le d\mathbb{E}_t[\|g_t - \mathbb{E}_t[g_t]\|_\infty^2] \\
&\le 2d\left(\mathbb{E}_t[\|g_t - \nabla f(x_t)\|_\infty^2] + \mathbb{E}_t[\|\nabla f(x_t) - \mathbb{E}_t[g_t]\|_\infty^2]\right) \\
&\le 4d\sigma^2.
\end{aligned}$$

Therefore, we arrive at

$$\mathbb{E}[f(x_T)] - f(x^\star) \le (1 - \eta\mu)^T (f(x_0) - f(x^\star)) + (1 + 4\eta L)\frac{d\sigma^2}{2\mu}.$$

The first term exhibits linear convergence, which we bound by $\varepsilon/2$ as follows. Observe that

$$T\log\left(\frac{1}{1-\mu\eta}\right) = T(-\log(1-\mu r)) \ge -\log\left(\frac{\varepsilon}{2(f(x_0) - f(x^\star))}\right) = \log\left(\frac{2(f(x_0) - f(x^\star))}{\varepsilon}\right).$$

Using $\log(1/\xi) \ge 1 - \xi$ for $\xi \in (0,1)$, we have

$$\log\left(\frac{1}{1-\mu\eta}\right) \le \mu\eta \implies \frac{1}{\mu\eta} \ge \frac{1}{\log\left(\frac{1}{1-\mu\eta}\right)}.$$

Thus,

$$T \ge \frac{1}{\mu\eta}\log\left(\frac{2(f(x_0) - f(x^\star))}{\varepsilon}\right) \ge \frac{1}{\log\left(\frac{1}{1-\mu\eta}\right)}\log\left(\frac{2(f(x_0) - f(x^\star))}{\varepsilon}\right)$$

suffices to have the first term bounded by $\varepsilon/2$; with $\eta = 1/L$, we have $T \geq \kappa \log\left(\frac{2(f(x_0)-f(x^\star))}{\varepsilon}\right)$.

Now, for the combined bias and variance term, we want

$$(1 + 4\eta L)\frac{d\sigma^2}{2\mu} \leq \frac{\varepsilon}{2} \implies \sigma \leq \sqrt{\frac{\varepsilon\mu}{d(1 + 4\eta L)}} \overset{\eta = 1/L}{=} \sqrt{\frac{\varepsilon\mu}{5d}}. \tag{10}$$

By assumption $\frac{\varepsilon\mu}{dG^2} \leq 5 \implies \frac{\sigma}{G} \leq 1$ as required. Plugging (10) back into the precision requirement for the evaluation oracle in Theorem A.2, we get:

$$\theta \leq \frac{\sigma^4}{2Ld(3G)^2\left(32\lceil\ln(72dG^2/\sigma^2)\rceil + 4\right)^2} = \frac{\left(\frac{\varepsilon\mu}{5d}\right)^2}{2Ld(3G)^2\left(32\left\lceil\ln\left(72dG^2 \cdot \frac{5d}{\varepsilon\mu}\right)\right\rceil + 4\right)^2}.$$

Simplifying the above bound gives the desired result on the precision requirement to the evaluation oracle. Using our choice of $T$ and applying Markov's inequality, we get

$$\frac{1}{3} \geq \Pr\left[f(x_t) - f(x^\star) \geq 3\mathbb{E}\left[f(x_t) - f(x^\star)\right]\right]$$
$$= \Pr\left[f(x_t) - f(x^\star) \geq \varepsilon\right].$$

To complete the proof, for each gradient estimation, we make $8\lceil\ln(72dG^2/\sigma^2)\rceil + 1 = \Theta(\log(d^2/\varepsilon_0))$ calls to the unitary $U_{\tilde{f}}$ per Theorem A.2 and using the bound in (10). This gives the total number of calls in the statement combined with $T \geq \kappa \log\left(\frac{2(f(x_0)-f(x^\star))}{\varepsilon}\right)$ we obtained previously. Finally, for the gate complexity, circuit depth, and phase oracle cost, we again use the bound (10) to the statement in Theorem A.2. □

### B.1.2 Convex functions with Lipschitz gradients

We now perform a similar analysis for convex functions with Lipschitz continuous gradients, but not necessarily strongly convex.

**Theorem B.2** (Quantum Gradient Descent for Convex & Smooth Functions). *Suppose $f\colon \mathbb{R}^d \to \mathbb{R}$ is a convex function with $L$-Lipschitz gradients. Consider the (zeroth-order) quantum gradient descent $x_{t+1} = x_t - \eta g_t$ with step size $\eta = 1/L$ that outputs the last iterate $x_T$ after running $T = \mathcal{O}(LR^2/\varepsilon)$ iterations, where the (biased stochastic) gradient $g_t$ is estimated via the quantum gradient estimation in Theorem A.2 on each iteration, and $R := \max_{f(x) \leq f(x_0)}\|x - x^\star\|_2$. Additionally, assume we are given access to the value of $f(x^\star)$.[5] Let $\varepsilon_0 := \varepsilon/(GR)$. Assume one has access to an $\theta$-precise evaluation oracle satisfying*

$$\theta \leq \frac{\varepsilon^4}{4608R^4d^3G^2L\left(32\left\lceil\ln\left(1152d^2G^2R^2/\varepsilon^2\right)\right\rceil + 4\right)^2} = \Theta\left(\frac{\varepsilon_0^4G^2}{d^3L\log(d/\varepsilon_0)^2}\right).$$

*Then, with probability at least $2/3$, one can obtain an $\varepsilon$-approximate solution with*

$$\Theta\left(\frac{LR^2}{\varepsilon}\log\left(d/\varepsilon_0\right)\right)$$

*queries to $\theta$-approximate binary oracle of $f$. The procedure has gate complexity $\mathcal{O}\left(d\log^2(d/\varepsilon_0)\log(\sqrt{d}/\varepsilon_0))\right)$ and corresponding circuit depth of $\mathcal{O}\left(\log(d/\varepsilon_0)\log(\sqrt{d}/\varepsilon_0)\right)$.*

*With $(GR_0)$-phase oracle access instead the above complexities must be multiplied by a factor of $\mathcal{O}\left(R_0\sqrt{dL}/\sqrt{\theta}\right)$, where $R_0$ is a length scale chosen to ensure that for all $x \in \mathcal{B}_\infty^d(x_0, R), f(x) \leq GR_0$.*

---

[5]This can typically be assumed without loss of generality with only polylogarithmic overhead in the inverse error. This is because as long as a bounded interval containing $f(x_\star)$ is known, we can guess a value and run our algorithm with that guess. The success of the algorithm can be verified by computing a single gradient at the final point. This allows the true value of the optimum to be calculated by binary search.

*Proof.* We follow the standard convergence proof for gradient descent on smooth and convex objectives (e.g., [51, Theorem 2.9]), with modifications to handle with the bias of the gradient. Define $\delta_t := f(x_{t+1}) - f(x^\star)$. By convexity and Cauchy-Schwarz we have

$$\delta_t = f(x_t) - f(x^\star) \leq \langle \nabla f(x_t), x_t - x^\star \rangle \leq \|\nabla f(x_t)\|_2 \|x_t - x^\star\|_2 \leq \|\nabla f(x_t)\|_2 R,$$

from which we have the bound $\delta_t/R \leq \|\nabla f(x_t)\|_2$. From Lemma B.1, we have:

$$\mathbb{E}_t [\delta_{t+1}] \leq \delta_t - \frac{\eta}{2} \|\nabla f(x_t)\|_2^2 + \frac{\eta}{2} \|b_t\|_2^2 + \frac{\eta^2 L}{2} \sigma_v^2,$$

$$\leq \delta_t - \frac{\eta}{2} \cdot \frac{\delta_t^2}{R^2} + \frac{\eta}{2} \|b_t\|_2^2 + \frac{\eta^2 L}{2} \sigma_v^2$$

$$= \delta_t - \frac{1}{2L} \cdot \frac{\delta_t^2}{R^2} + \frac{1}{2L} \|b_t\|_2^2 + \frac{1}{2L} \sigma_v^2$$

where we denoted $\mathbb{E}_t := \mathbb{E} [\cdot | x_t]$. The key insight is that, using Theorem A.2, we can suppress both the bias and variance enough on each iteration to ensure descent. Hence, we demand

$$\|b_t\|_2 \leq \frac{\delta_t}{2R} \quad \text{and} \quad \sigma_v \leq \frac{\delta_t}{2R}, \tag{11}$$

which we will justify later. Given that we assumed access to $f(x^\star)$, at each iteration we can compute the value $\delta_t$ to input to the suppressed-bias quantum gradient algorithm. Then, we have

$$\mathbb{E}_t [\delta_{t+1}] \leq \delta_t - \frac{\delta_t^2}{2LR^2} + \frac{1}{2L} \|b_t\|_2^2 + \frac{1}{2L} \sigma_v^2$$

$$\leq \delta_t - \frac{\delta_t^2}{2LR^2} + \frac{\delta_t^2}{8LR^2} + \frac{\delta_t^2}{8LR^2}$$

$$= \delta_t - \frac{1}{4L} \left( \frac{\delta_t}{R} \right)^2.$$

Taking expectation again and using the law of total expectation, we have

$$\mathbb{E} [\delta_{t+1}] \leq \mathbb{E} [\delta_t] - \frac{1}{4L} \mathbb{E} \left[ \left( \frac{\delta_t}{R} \right)^2 \right] \leq \mathbb{E} [\delta_t] - \frac{1}{4L} \left( \frac{\mathbb{E} [\delta_t]}{R} \right)^2,$$

where the last step follows from an application of Jensen's inequality. Rearranging, we have

$$\frac{1}{4LR^2} \leq \frac{\mathbb{E} [\delta_{t+1}] - \mathbb{E} [\delta_{t+1}]}{\left( \mathbb{E} [\delta_t] \right)^2} \leq \frac{\mathbb{E} [\delta_{t+1}] - \mathbb{E} [\delta_{t+1}]}{\mathbb{E} [\delta_t] \mathbb{E} [\delta_{t+1}]} = \frac{1}{\mathbb{E} [\delta_{t+1}]} - \frac{1}{\mathbb{E} [\delta_t]}.$$

Rearranging and unfolding, we have

$$\frac{1}{\mathbb{E} [\delta_t]} \geq \frac{1}{\mathbb{E} [\delta_{t-1}]} + \frac{1}{4LR^2} \geq \cdots \geq \frac{1}{\delta_0} + \frac{t}{4LR^2},$$

where we used $\mathbb{E}[\delta_t] \geq \mathbb{E}[\delta_{t+1}]$ coming from the assurance of descent. By $L$-smoothness in (9), we have

$$\delta_0 = f(x_0) - f(x^\star) \leq \frac{L}{2} \|x_0 - x^\star\|_2^2 \leq \frac{LR^2}{2}.$$

Thus,

$$\frac{1}{\mathbb{E} [\delta_t]} \geq \frac{1}{\delta_0} + \frac{t}{4LR^2} \geq \frac{2}{LR^2} + \frac{t}{4LR^2} \geq \frac{t+8}{4LR^2} \implies \mathbb{E} [\delta_t] \leq \frac{4LR^2}{t+8}.$$

This gives the desired scaling in $T$. It remains to justify the choices in (11). To that end, we again use Theorem A.2 (with $\vartheta = \vartheta_* = \sqrt{d}$). For the bias term $\|b_t\|_2$, we have for all $t$,

$$\|b_t\|_2 = \|\mathbb{E}[g_t] - \nabla f(x_t)\|_2 \leq \sqrt{d}\sigma.$$

For the variance term involving $\sigma_v^2$,

$$\sigma_v^2 := \mathbb{E}[\|g_t - \mathbb{E}[g_t]\|_2^2] \leq 2 \left( \mathbb{E}[\|g_t - \nabla f(x_t)\|_2^2] + \mathbb{E}[\|\nabla f(x_t) - \mathbb{E}[g_t]\|_2^2] \right)$$

$$\leq 4d\sigma^2.$$

Therefore, it suffices to choose $4d\sigma^2 \geq \sigma_v^2, \|b_t\|_2^2$ to compute the precision requirement on the evaluation oracle from Theorem A.2. We will assume that once we observe that $\delta_{t+1} \leq \varepsilon$, we terminate the algorithm, and so $\delta_t$ in the bias and variance requirements in (11) is lower bounded by $\varepsilon$. Thus, we only have to suppress the bias and variance enough such that

$$\sigma^2 \leq \frac{\varepsilon^2}{16R^2 d}. \tag{12}$$

We now plug in our condition on $\sigma$ in (12) to obtain the precision requirement on the evaluation oracle.

$$\theta \leq \frac{\sigma^4}{2Ld(3G)^2 \left(32\lceil \ln(72dG^2/\sigma^2)\rceil + 4\right)^2} = \frac{\left(\frac{\varepsilon^2}{16R^2 d}\right)^2}{2Ld(3G)^2 \left(32\left\lceil \ln\left(72dG^2 \cdot \frac{16R^2 d}{\varepsilon^2}\right)\right\rceil + 4\right)^2}.$$

The statement in the theorem can be obtained by simplifying the above.

To complete the proof, for each gradient estimation, we make $8\lceil \ln(72dG^2/\sigma^2)\rceil + 1 = \Theta(\ln(dG^2/\sigma^2))$ calls to the unitary $U_{\tilde{f}}$ per Theorem A.2. Using the bound in (12), we have

$$\Theta\left(\ln\left(\frac{dG^2}{\sigma^2}\right)\right) = \Theta\left(\ln\left(\frac{d^2 G^2 R^2}{\varepsilon^2}\right)\right),$$

giving the total number of calls in the statement combined with $T = \mathcal{O}\left(\frac{LR^2}{\varepsilon}\right)$ we obtained previously. Finally, for the gate complexity and circuit depth, we again use the bound (10) to the statement in Theorem A.2.

Using our choice of $T$ and applying Markov's inequality, we get

$$\frac{1}{3} \geq \Pr\left[f(x_t) - f(x^\star) \geq 3\mathbb{E}\left[f(x_t) - f(x^\star)\right]\right]$$
$$= \Pr\left[f(x_t) - f(x^\star) \geq \varepsilon\right].$$

$\square$

## B.2  Zeroth-order quantum projected subgradient method

Suppose that $f : \mathbb{R}^d \to \mathbb{R}$ is $G$-Lipschitz on $\mathcal{X}$, where $\mathcal{X}$ is closed, bounded, and nonempty. Define $K := \text{diam}(\mathcal{X}) = \sup_{x,y \in \mathcal{X}} \|x - y\|_2$. Define $\Pi_{\mathcal{X}}$ to be the Euclidean projection onto $\mathcal{X}$. The subgradient method is defined by the following equations:

$$x_{t+1} = \Pi_{\mathcal{X}}\left(x_t - \eta g_t\right), \quad g_t \in \partial f(x_t). \tag{PSM}$$

We now analyze the precision requirement on the evaluation oracle to implement the zeroth-order quantum projected subgradient method for convex and $G$-Lipschtiz functions, which can be nonsmooth. The algorithm we analyze is similar to the (classical) subgradient method but equipped with the quantum subgradient estimation from [75, Lemma 18], which we reproduced in Lemma A.1. Note that one can perform a similar analysis with [20, Theorem 2.2].

**Theorem B.3** (Quantum projected subgradient method). *Suppose $f : \mathbb{R}^d \to \mathbb{R}$ is a convex function that is $G$-Lipschitz. Consider the zeroth-order quantum projected subgradient method $x_{t+1} = \Pi_{\mathcal{X}}\left(x_t - \eta \tilde{g}_t\right)$, with $\eta = \frac{R}{G\sqrt{T}}$, that outputs the average $\frac{1}{T}\sum_{t=1}^{T} x_t$ after running $T$ iterations, where the $\tilde{g}_t$ is an approximate subgradient at $x_t$ computed via quantum subgradient estimation in Lemma A.1 on each iteration. Assume one has access to an $\theta$-precise evaluation oracle for $f$, with $\theta = \mathcal{O}\left(\frac{\varepsilon^5}{G^4 R^4 d^{4.5}}\right)$. Then, using the quantum projected subgradient method, with probability at least $2/3$, one can obtain an $\varepsilon$-approximate solution to (OPT) with $\widetilde{\mathcal{O}}(G^2 R^2/\varepsilon^2)$ queries to an $\theta$-approximate binary oracle of $f$ and $\widetilde{\mathcal{O}}((d+\mathcal{T}_\Pi)(GR/\varepsilon)^2)$ gates, where $\mathcal{T}_\Pi$ is the cost of performing the projection onto $\mathcal{X}$.*

*Proof.* Recall that the projected subgradient algorithm iterates as $x_{t+1} = \Pi_{\mathcal{X}}\left(x_t - \eta \tilde{g}_t\right)$, where $\tilde{g}_s$ is a subgradient at $x_t$, and $\eta > 0$ is a step size, which we choose later. For analysis purposes, we

denote the intermediate variable before the projection step as $z_t$. That is, we can write the projected subgradient method as

$$z_{t+1} = x_t - \eta g_t,$$
$$x_{t+1} = \Pi_{\mathcal{X}}(z_{t+1}).$$

We analyze the case where the projected subgradient method is performed for $T$ iterations, and outputs the average: $\frac{1}{T}\sum_{t=1}^{T} x_t$. Following standard analysis of projected subgradient methods (e.g. [16, Theorem 3.2]), and utilizing Lemma A.1 (where $\vartheta = \vartheta^* = \sqrt{d}$ when we specifically use the Euclidean norm $\|\cdot\|_2$, i.e., $p = 2$), we have:

$$f(x_t) - f(x^\star) \leq \langle \tilde{g}_t, x_t - x^\star \rangle + (23d)^2 \sqrt{\frac{\theta G}{\rho r_1}} \|x_t - x^\star\|_2 + 2G\sqrt{d}r_1$$

$$= \frac{1}{\eta} \langle x_t - z_{t+1}, x_t - x^\star \rangle + (23d)^2 \sqrt{\frac{\theta G}{\rho r_1}} \|x_t - x^\star\|_2 + 2G\sqrt{d}r_1$$

$$= \frac{1}{2\eta} \left( \|x_t - x^\star\|_2^2 - \|z_{t+1} - x^\star\|_2^2 \right) + \frac{\eta}{2}\|\tilde{g}_t\|_2^2 + (23d)^2 \sqrt{\frac{\theta G}{\rho r_1}} \|x_t - x^\star\|_2 + 2G\sqrt{d}r_1$$

$$\leq \frac{1}{2\eta} \left( \|x_t - x^\star\|_2^2 - \|z_{t+1} - x^\star\|_2^2 \right) + \frac{\eta}{2}\|\tilde{g}_t\|_2^2 + (23d)^2 \sqrt{\frac{\theta G}{\rho r_1}} K + 2G\sqrt{d}r_1$$

$$\leq \frac{1}{2\eta} \left( \|x_t - x^\star\|_2^2 - \|x_{t+1} - x^\star\|_2^2 \right) + \frac{\eta}{2}\|\tilde{g}_t\|_2^2 + (23d)^2 \sqrt{\frac{\theta G}{\rho r_1}} K + 2G\sqrt{d}r_1$$

$$\leq \frac{1}{2\eta} \left( \|x_t - x^\star\|_2^2 - \|x_{t+1} - x^\star\|_2^2 \right) + \frac{\eta}{2}\|g - g + \tilde{g}_t\|_2^2 + (23d)^2 \sqrt{\frac{\theta G}{\rho r_1}} K + 2G\sqrt{d}r_1$$

$$\leq \frac{1}{2\eta} \left( \|x_t - x^\star\|_2^2 - \|x_{t+1} - x^\star\|_2^2 \right) + G^2\eta + 23^4\eta \frac{\theta d^4 G}{\rho r_1} + (23d)^2 \sqrt{\frac{\theta G}{\rho r_1}} K + 2G\sqrt{d}r_1$$

where in the second equality we used $\|a\|_2^2 + \|b\|_2^2 - \|a-b\|_2^2 = 2\langle a, b \rangle$, and the second inequality is since $x_{t+1} = \Pi_{\mathcal{X}}(z_{t+1})$ and hence $\|z_{t+1} - x^\star\|_2 \geq \|x_{t+1} - x^\star\|_2$. In the last step, we also used from (8) with $\vartheta = \vartheta_* = \sqrt{d}$ such that

$$\sup_{g \in \partial f(z)} \|g - \tilde{g}\|_2 < 23^2 \sqrt{\frac{\theta d^4 G}{\rho r_1}}.$$

for some $z$ that is internal to the quantum subgradient estimation algorithm.

Summing over $t \in [T]$, and using $\|x_1 - x^\star\|_2^2 \leq R^2$, we get

$$\sum_{t=1}^{T} (f(x_t) - f(x^\star)) \leq \frac{R^2}{2\eta} + \frac{\eta G^2}{2}T + 23^4 T\eta \frac{\theta d^4 G}{\rho r_1} + (23d)^2 \sqrt{\frac{\theta G}{\rho r_1}} KT + 2G\sqrt{d}r_1 T.$$

Dividing both sides by $T$, choosing $\eta = \frac{R}{G\sqrt{T}}$, and using $f\left(\frac{1}{T}\sum_{t=1}^{T} x_t\right) \leq \frac{1}{T}\sum_{t=1}^{T} f(x_t)$ by convexity, we have

$$f\left(\frac{1}{T}\sum_{t=1}^{T} x_t\right) - f(x^\star) \leq \frac{RG}{\sqrt{T}} + 23^4 \frac{\theta d^4 R}{\sqrt{T}\rho r_1} + (23d)^2 \sqrt{\frac{\theta G}{\rho r_1}} K + 2G\sqrt{d}r_1.$$

We want the above expression to be bounded by $\varepsilon$. First, choose $T = \left(\frac{3RG}{\varepsilon}\right)^2$ and $r_1 = \frac{\varepsilon}{6G\sqrt{d}}$. Then, the first and the fourth terms are bounded by $\varepsilon/3$. We want to second term to be bounded by $\varepsilon/3$ as well; before doing so, we first choose the failure probability $\rho$ such that the all $T$ calls of quantum

subgradient estimation in Lemma A.1 succeed with (overall) probability $\geq 2/3$. Observe that, if we choose $\rho = 1/(3T)$, then the probability that none of the $T$ subroutines fail is:

$$\left(1 - \frac{1}{3T}\right)^T \geq 1 - \frac{T}{3T} = \frac{2}{3}.$$

Note that the above bound holds even when each call of the quantum subgradient estimation is dependent; indeed, by union bound, at least one call of the quantum subgradient estimation fails is at most $T \cdot 1/(3T) \leq 1/3$, or equivalently, the probability that none of the $T$ subroutines fails is $\geq 2/3$. With our choice $T = \left(\frac{3RG}{\varepsilon}\right)^2$, we have $\rho = \frac{\varepsilon^2}{27G^2R^2}$. Given our choice the second term becomes

$$23^4 \frac{\theta d^4 R}{\sqrt{T}\rho r_1} \leq (23^4 \cdot 9 \cdot 6) \frac{d^{4.5}\theta G^2 R^2}{\varepsilon^2},$$

and the precision requirement on the evaluation oracle $\theta$ is

$$(23^4 \cdot 9 \cdot 6) \frac{d^{4.5}\theta G^2 R^2}{\varepsilon^2} + (23d)^2 \cdot K \cdot \left(\theta G \cdot \frac{27G^2 R^2}{\varepsilon^2} \cdot \frac{6G\sqrt{d}}{\varepsilon}\right)^{1/2} \leq \frac{\varepsilon}{3}.$$

Solving for $\theta$, we get $\theta = \mathcal{O}\left(\frac{\varepsilon^5}{G^4 R^4 d^{4.5}}\right)$, completing the proof.

Finally, from Lemma 18 in [75] the discrete hypergrid used in the subgradient estimation algorithm has radius $r = \frac{2r_2}{d}$, with $r_2 := \sqrt{\frac{\theta r_1 \rho}{dG}}$, giving the phase oracle cost. $\qquad\square$

## C   Generalizing to non-Euclidean geometries

We have seen that one can obtain dimension-free oracle complexities when the objective function $f$ is Lipschitz in the Euclidean norm. However, if $f$ is well-behaved in some other norm, the schemes discussed in the previous section may fail to maintain their dimension-free rates. Moreover, many problems in optimization are defined over non-Euclidean spaces like the probability simplex and the cone of positive semidefinite matrices.

In [61] Nemirovskii and Yudin observed that one can first define a mapping from the primal space to the dual space, perform the subgradient update in the dual space, and subsequently map back to the primal. This approach allows one to better leverage the problem geometry, and is well-suited to high-dimensional constrained optimization problems. In order to keep the paper self-contained, we briefly review some concepts that are essential to the mirror descent framework. Then, we prove that one can instantiate mirror descent with quantum (sub)gradient estimation, and provide specialized implementations, such as Nesterov's dual averaging [63] and mirror prox [60].

**Note that within this section, we use $\| \cdot \|$ to denote an arbitrary $p$-norm ($p \geq 1$), and its dual norm as $\| \cdot \|_*$.**

### C.1   Bregman divergence and Bregman projection

The methods we discuss in this section will require distance measures beyond the Euclidean metric. We work with the *Bregman divergence*, defined below.

**Definition 3** (Bregman divergence)**.** *Let $\mathcal{P} \subset \mathbb{R}^d$ be a convex open set. Given a strictly convex differentiable function $\Phi : \mathcal{P} \to \mathbb{R}$, we define the associated Bregman divergence $D_\Phi : \mathcal{P} \times \mathcal{P} \to \mathbb{R}_{\geq 0}$ by*

$$D_\Phi(x, \bar{x}) := \Phi(x) - \Phi(\bar{x}) - \langle \nabla\Phi(\bar{x}), x - \bar{x} \rangle.$$

Note that $D_\Phi(x, \bar{x})$ is nonnegative for all $(x, \bar{x}) \in \mathcal{P} \times \mathcal{P}$, since $\Phi$ is convex. The Bregman divergence can be interpreted as the level of error present in the first-order Taylor series approximation of $\Phi$ at $\bar{x}$ to $\Phi(x)$. Unlike the Euclidean distance, the Bregman divergence is not symmetric. This is because the Bregman divergence is computed according to the local geometry with respect to $\nabla\Phi$ at $\bar{x}$.

The following identity will be useful in our analysis later:

$$\langle \nabla\Phi(x) - \nabla\Phi(\bar{x}), x - w \rangle = D_\Phi(x, \bar{x}) + D_\Phi(w, x) - D_\Phi(w, \bar{x}). \tag{13}$$

For a closed convex set $\mathcal{X} \subseteq \mathcal{P}$, the *Bregman projection* $\Pi_\mathcal{X}^\Phi : \mathcal{P} \to \mathcal{X}$ of a point $x \in \mathbb{R}^d$ onto $\mathcal{X}$ is defined as $\Pi_\mathcal{X}^\Phi(\bar{x}) := \arg\min_{x \in \mathcal{X}} D_\Phi(x, \bar{x})$. Note that the Bregman projection is uniquely defined since $\Phi$ is strictly convex.

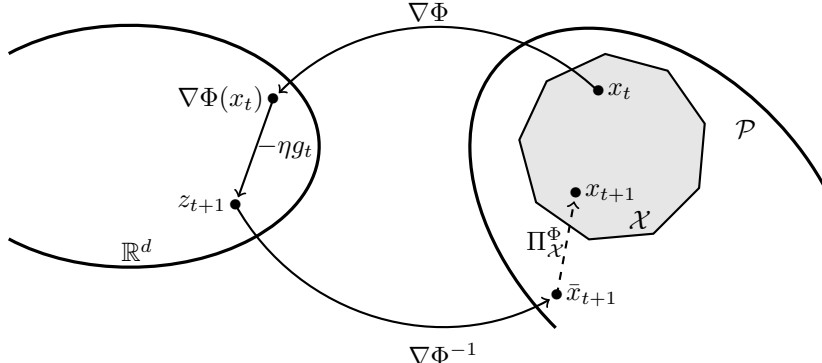

Figure 2: One iteration of mirror descent. At iteration $t$, the point $x_t \in \mathcal{X}$ in the primal space $\mathcal{P}$ is mapped into the dual space $\mathbb{R}^d$, where a subgradient step can be taken. Upon mapping back to $\mathcal{P}$, the Bregman projection $\Pi_{\mathcal{X}}^{\Phi}$ is applied to obtain the point $x_{t+1}$ in $\mathcal{X}$.

## C.2 Methods of mirror descent

Mirror decent fundamentally applies a primal-dual approach to convex optimization, relying on the ability to map points from a primal ambient space $\mathcal{P}$ to its dual space $\mathbb{R}^d$ (and vise-versa). This mapping is defined as the gradient of a *mirror map*, which we formally define next. For more details on mirror maps, we refer the reader to [16, Section 4.1].

**Definition 4** (Mirror map). *Let $\mathcal{P} \subset \mathbb{R}^d$ be a convex open set that contains $\mathcal{X}$ in its closure, i.e., $\mathcal{X} \subset \mathrm{cl}(\mathcal{P})$. Further assume that the intersection of $\mathcal{X}$ and $\mathcal{P}$ is nonempty. A mirror map $\Phi : \mathcal{P} \to \mathbb{R}$ is a strictly convex, differentiable function, whose gradient $\nabla\Phi$ satisfies the following:*

1. *The gradient of $\Phi$ takes all possible values in $\mathbb{R}^d$, i.e., $\nabla\Phi(\mathcal{P}) = \mathbb{R}^d$.*

2. *The gradient of $\Phi$ diverges on the boundary of $\mathcal{P}$, i.e., $\lim_{x \to \partial\mathcal{P}} \|\nabla\Phi(x)\| = \infty$.*

At each iterate $t$ of a mirror descent algorithm, a primal point $x_t \in \mathcal{X} \cap \mathcal{P}$ is mapped to a point $\nabla\Phi(x_t) \in \mathbb{R}^d$ in the dual space. From here, a subgradient step (with step length $\eta$) is taken in an improving direction with respect to the dual. One can compactly write the mirror descent iterate as

$$x_{t+1} = \Pi_{\mathcal{X}}^{\Phi}\left(\nabla\Phi^{-1}\left(\nabla\Phi(x_t) - \eta g_t\right)\right), \quad g_t \in \partial f(x_t). \tag{MD}$$

The following lemma from [16] illustrates how the Bregman divergence functions similar to the Euclidean norm squared in terms of projections.

**Lemma C.1** (Lemma 4.1 in [16]). *Let $x \in \mathcal{X} \cap \mathcal{P}$ and $\bar{x} \in \mathcal{P}$. Then,*

$$\left\langle \nabla\Phi\left(\Pi_{\mathcal{X}}^{\Phi}(\bar{x})\right) - \nabla\Phi(\bar{x}), \Pi_{\mathcal{X}}^{\Phi}(\bar{x}) - x \right\rangle \leq 0 \implies D_{\Phi}\left(x, \Pi_{\mathcal{X}}^{\Phi}(\bar{x})\right) + D_{\Phi}\left(\Pi_{\mathcal{X}}^{\Phi}(\bar{x}), \bar{x}\right) \leq D_{\Phi}(x, \bar{x}).$$

Adapting a standard convergence proof for mirror descent (see, e.g., [16, Theorem 4.2]) to account for quantum subgradient estimation suffices to prove the following result for $\ell_p$ and Schatten norms.

**Theorem C.1** (Mirror Descent Using Quantum Subgradient Estimation). *Let $\mathcal{X} \subseteq \mathbb{R}^d$ be a closed and bounded convex set equipped with a given $p$-norm $\|\cdot\|$ with $p \geq 1$. Denote $K := \mathrm{diam}(\mathcal{X}) = \sup_{x,y \in \mathcal{X}} \|x - y\|$. Suppose $f : \mathbb{R}^d \to \mathbb{R}$ is a convex function that is $G$-Lipschitz with respect to $\|\cdot\|$ and attains its minimum at $x^\star \in \mathcal{X} \cap \mathcal{P}$. Let $\Phi$ be a mirror map that is $\mu$-strongly convex on $\mathcal{X} \cap \mathcal{P}$ with respect to $\|\cdot\|$. Suppose we are given a point $x_1 \in \mathcal{X} \cap \mathcal{P}$ and define $R^2 = D_{\Phi}(x^\star, x_1)$. Assume one has access to an $\theta$-precise binary oracle for $f$, with*

$$\theta = \mathcal{O}\left(\frac{\mu\varepsilon^5}{G^4 R^2 K^2 \vartheta_*^2 \vartheta d^3}\right).$$

*Consider the zeroth-order quantum mirror descent method*

$$x_{t+1} = \Pi_{\mathcal{X}}^{\Phi}\left(\nabla\Phi^{-1}\left(\nabla\Phi(x_t) - \eta\tilde{g}_t\right)\right), \tag{MD}$$

with $\eta = \frac{R}{G}\sqrt{\frac{\mu}{T}}$, that outputs the average $\frac{1}{T}\sum_{t=1}^{T} x_t$ after running $T$ iterations, where the $\tilde{g}_t$ is an approximate element of $\partial f(x_t)$ estimated via the quantum subgradient estimation in Lemma A.1 at each iterate. Then, with probability at least $2/3$, one can obtain an $\varepsilon$-approximate solution with $\widetilde{\mathcal{O}}\left(\left(\frac{GR}{\sqrt{\mu}\varepsilon}\right)^2\right)$ queries to $\theta$-approximate evaluation oracle of $f$ and $\widetilde{\mathcal{O}}\left((d + \mathcal{T}_{next})\left(\frac{GR}{\sqrt{\mu}\varepsilon}\right)^2\right)$ gates, where $\mathcal{T}_{next}$ is the time to carry out the operations necessary to progress to the next iterate.

*Proof.* From Lemma A.1, the output $\tilde{g}$ satisfies the following with at least $1 - \rho$ probability:

$$f(q) \geq f(x) + \langle \tilde{g}, q - x \rangle - 23^2 \vartheta_* \sqrt{\frac{\theta d^3 G}{\rho r_1}} \|q - x\| - 2G\vartheta r_1,$$

where $1 \leq \vartheta, \vartheta^* \leq d$ and $\vartheta\vartheta_* \leq d$. With $q \leftarrow x^\star$ and rearranging, we have the following:

$$\Xi_1 := 23^2 \vartheta_* \sqrt{\frac{\theta d^3 G}{\rho r_1}} K + 2G\vartheta r_1 \tag{14}$$

$$\geq 23^2 \vartheta_* \sqrt{\frac{\theta d^3 G}{\rho r_1}} \|x^\star - x\| + 2G\vartheta r_1$$

$$\geq f(x) - f(x^\star) + \langle \tilde{g}, x^\star - x \rangle,$$

where we also define the quantity $\Xi_1$.

Continuing, we have

$$f(x_t) - f(x^\star) \leq \langle \tilde{g}_t, x_t - x^\star \rangle + \Xi_1$$

$$\stackrel{\text{Eq. (MD)}}{=} \frac{1}{\eta} \langle \nabla \Phi(x_t) - \nabla \Phi(\bar{x}_{t+1}), x_t - x^\star \rangle + \Xi_1$$

$$\stackrel{\text{Eq. (13)}}{=} \frac{1}{\eta} \left( D_\Phi\left(x^\star, x_t\right) + D_\Phi\left(x_t, \bar{x}_{t+1}\right) - D_\Phi\left(x^\star, \bar{x}_{t+1}\right) \right) + \Xi_1$$

$$\stackrel{\text{Lem. C.1}}{\leq} \frac{1}{\eta} \left( D_\Phi\left(x^\star, x_t\right) + D_\Phi\left(x_t, \bar{x}_{t+1}\right) - D_\Phi\left(x^\star, x_{t+1}\right) - D_\Phi\left(x_{t+1}, \bar{x}_{t+1}\right) \right) + \Xi_1.$$

The term $D_\Phi\left(x^\star, x_t\right) - D_\Phi\left(x^\star, x_{t+1}\right)$ leads to a telescopic sum when summing over $t \in [T]$. What remains is to bound the other term using $\mu$-strong convexity of the mirror map combined with the identity

$$az - bz^2 \leq \frac{a^2}{4b}, \quad \forall z \in \mathbb{R}.$$

Indeed,

$$D_\Phi\left(x_t, \bar{x}_{t+1}\right) - D_\Phi\left(x_{t+1}, \bar{x}_{t+1}\right) = \Phi(x_t) - \Phi(x_{t+1}) - \langle \nabla\Phi(\bar{x}_{t+1}), x_t - x_{t+1} \rangle$$

$$\leq \langle \nabla\Phi(x_t) - \nabla\Phi(\bar{x}_{t+1}), x_t - x_{t+1} \rangle - \frac{\mu}{2} \|x_t - x_{t+1}\|^2$$

$$= \eta \langle \tilde{g}_t, x_t - x_{t+1} \rangle - \frac{\mu}{2} \|x_t - x_{t+1}\|^2$$

$$\leq \eta \|\tilde{g}_t\|_* \|x_t - x_{t+1}\| - \frac{\mu}{2} \|x_t - x_{t+1}\|^2$$

$$\leq \eta \|\tilde{g}_t - g + g\|_* \|x_t - x_{t+1}\| - \frac{\mu}{2} \|x_t - x_{t+1}\|^2$$

$$\leq \eta \left(\left(\|\tilde{g}_t - g\|_* + G\right) \|x_t - x_{t+1}\|\right) - \frac{\mu}{2} \|x_t - x_{t+1}\|^2$$

$$\leq \frac{(\eta(G + \Xi_2))^2}{2\mu}$$

$$\leq \frac{\eta^2 G^2}{\mu} + \frac{\eta^2 \Xi_2^2}{\mu}, \tag{15}$$

where in the second to last step, we used from (8):

$$\sup_{g \in \partial f(z)} \|g - \tilde{g}\|_* < 23^2 \vartheta_* \sqrt{\frac{\theta d^3 G}{\rho r_1}} := \Xi_2, \tag{16}$$

for some $z$ that is internal to the quantum subgradient estimation algorithm.

Therefore,

$$\sum_{s=1}^{T} (f(x_t) - f(x^\star)) \leq \frac{D_\Phi(x^\star, x_1)}{\eta} + \eta \frac{G^2}{\mu} T + T\Xi_1 + T\Xi_2^2 \frac{\eta}{\mu}$$

$$\leq \frac{R^2}{\eta} + \eta \frac{G^2}{\mu} T + \left[ 23^2 \vartheta_* \sqrt{\frac{\theta d^3 G}{\rho r_1}} K + 2G\vartheta r_1 + (23)^4 \frac{\eta}{\mu} \vartheta_*^2 \frac{\theta d^3 G}{\rho r_1} \right] T$$

Dividing both sides by $T$, taking $\eta = \frac{R}{G}\sqrt{\frac{\mu}{T}}$ and using convexity, i.e.,

$$f\left( \frac{1}{T} \sum_{s=1}^{T} x_t \right) \leq \frac{1}{T} \sum_{s=1}^{T} f(x_t)$$

one obtains

$$f\left( \frac{1}{T} \sum_{s=1}^{T} x_t \right) - f(x^\star) \leq \frac{2GR}{\sqrt{\mu T}} + \mathcal{O}\left( \vartheta_* \sqrt{\frac{\theta d^3 G}{\rho r_1}} K + G\vartheta r_1 + \frac{R}{\sqrt{\mu T}} \vartheta_*^2 \frac{\theta d^3}{\rho r_1} \right). \qquad (17)$$

We want the above expression to be bounded by $\varepsilon$. First, choose $T = \left( \frac{6GR}{\sqrt{\mu}\varepsilon} \right)^2$ and $r_1 = \mathcal{O}\left( \frac{\varepsilon}{G\vartheta} \right)$.
Then, the first and the fourth terms are bounded by $\varepsilon/3$. We want the middle terms to be bounded by $\varepsilon/3$ as well; before doing so, we first choose the failure probability $\rho$ such that the all $T$ calls of quantum subgradient estimation in Lemma A.1 succeed with (overall) probability $\geq 2/3$. Observe that, if we choose $\rho = 1/(3T)$, then the probability that none of the $T$ subroutines fail is:

$$\left( 1 - \frac{1}{3T} \right)^T \geq 1 - \frac{T}{3T} = \frac{2}{3}.$$

Note that the above bound holds even when each call of the quantum subgradient estimation is dependent; indeed, by union bound, at least one call of the quantum subgradient estimation fails is at most $T \cdot 1/(3T) \leq 1/3$, or equivalently, the probability that none of the $T$ subroutines fails is $\geq 2/3$.
With our choice $T = \left( \frac{6GR}{\sqrt{\mu}\varepsilon} \right)^2$, we have $\rho = \mathcal{O}\left( \frac{\mu\varepsilon^2}{G^2 R^2} \right)$.

Given these choices, the precision requirement on the evaluation oracle $\theta$ is determined by the middle two terms in (17), giving

$$\theta = \mathcal{O}\left( \frac{\mu\varepsilon^5}{G^4 R^2 K^2 \vartheta_*^2 \vartheta d^3} \right).$$

$\square$

We now review some common setups for the mirror descent framework that are relevant to the sequel, following a discussion from [16, Chapter 4.3]. When one takes $\Phi(x) = \frac{1}{2} \|x\|_2^2$ on $\mathcal{P} = \mathbb{R}^d$, the mirror map is strongly convex with respect to $\|\cdot\|_2$ and the associated Bregman divergence is $D_\Phi(x, \bar{x}) = \frac{1}{2} \|x - \bar{x}\|_2^2$. In other words, mirror descent reduces to the projected subgradient method described in Theorem B.3.

For the simplex $\Delta^d = \{x \in \mathbb{R}_{\geq 0}^d : \sum_{i \in [d]} x_i = 1\}$, one can take the mirror map to be the negative entropy

$$\Phi(x) = \sum_{i \in [d]} x_i \log x_i$$

on $\mathcal{P} = \mathbb{R}_+^d$, which is 1-strongly convex with respect to $\|\cdot\|_1$. This setting is relevant for linear programming and zero-sum games. When $\mathcal{X} = \Delta^d$, one has $x_1 = (1/d, \ldots, 1/d)$, $R^2 = \log(d)$ and projection is simply defined as renormalization by the $\ell_1$-norm, i.e., for any $\bar{x} \in \mathcal{P}$

$$\Pi_{\Delta^d}^\Phi : \bar{x} \mapsto \frac{\bar{x}}{\|\bar{x}\|_1}.$$

As a consequence, the iterates also admit a simple closed form expression:

$$x_{t+1,i} = \frac{x_{t,i}e^{-\eta g_{t,i}}}{\sum_{i \in [d]} x_{t,i}e^{-\eta g_{t,i}}}, \quad g_t \in \partial f(x_t), \ i \in [d], \tag{18}$$

and the associated Bregman divergence is the *Kullback-Leibler* (KL) divergence

$$D_\Phi(x, \bar{x}) = \sum_{i \in [d]} x_i \log(x_i/\bar{x}_i).$$

Mirror descent can also be applied to solve semidefinite programs by casting the problem into the form of minimizing a convex function $f$ over the spectraplex

$$\mathbb{S}^d := \{X \in \mathbb{R}^{d \times d} : \mathrm{tr}(X) = 1, X \succeq 0\}.$$

Here, the relevant mirror map is the negative von Neumann entropy

$$\Phi(X) := \mathrm{tr}\left(X \log(X)\right),$$

which is $\frac{1}{2}$-strongly convex with respect to the trace norm $\|\cdot\|_{\mathrm{tr}}$ (i.e., the $\ell_1$-norm of the singular values). The associated Bregman divergence is the *quantum relative entropy*

$$D_\Phi(X, \overline{X}) = S(X\|\overline{X}) := \mathrm{tr}\left(X\left(\log(X) - \log(\overline{X})\right)\right),$$

and the Bregman projection is simply renormalization with respect to trace:

$$\Pi_{\mathbb{S}^d}^\Phi : \overline{X} \mapsto \frac{\overline{X}}{\mathrm{tr}(\overline{X})}.$$

In fine, the iterates take the form

$$X_{t+1} = \frac{\exp(\log(X_t) - \eta g_t)}{\mathrm{tr}\left(\exp(\log(X_t) - \eta g_t)\right)}. \tag{19}$$

For $\mathcal{X} = \mathbb{S}^d$, one can choose the starting point $X = \frac{1}{d}I$, which gives $R^2 = \log(d)$.

Observe that for functions $f$ that are 1-Lipschitz with respect to $\|\cdot\|_1$, mirror descent with the negative entropy mirror map enjoys a $\mathcal{O}\left(\sqrt{\frac{\log(d)}{T}}\right)$ rate of convergence, whereas the projected subgradient method in Theorem B.3 only achieves a rate of $\mathcal{O}\left(\sqrt{\frac{d}{T}}\right)$. Moreover, the rate of convergence for the spectraplex is the same as that for the simplex.

As noted in [16, Chapter 4.2], one can re-write mirror descent iterates as

$$\begin{aligned}
x_{s+1} &= \underset{x \in \mathcal{X} \cap \mathcal{P}}{\arg\min} D_\Phi(x, \bar{x}_{s+1}) \\
&= \underset{x \in \mathcal{X} \cap \mathcal{P}}{\arg\min} \Phi(x) - \langle \nabla\Phi(\bar{x}_{s+1}), x \rangle \\
&= \underset{x \in \mathcal{X} \cap \mathcal{P}}{\arg\min} \Phi(x) - \langle \nabla\Phi(x_s) - \eta g_s, x \rangle \\
&= \underset{x \in \mathcal{X} \cap \mathcal{P}}{\arg\min} \eta \langle g_s, x \rangle + D_\Phi(x, x_s).
\end{aligned} \tag{20, 21}$$

Next, we will see how (20) is used to motivate Nesterov's *dual averaging* [63], while (21) yields the so-called *mirror prox* perspective [9, 60]. The former can be more efficient than the prototypical mirror descent scheme in some settings, whereas mirror prox will allow us to work with functions whose gradients are Lipschitz in arbitrary norms.

## C.3  Dual averaging

Now we discuss an alternative instantiation of the mirror descent algorithm, commonly referred to as Nesterov's dual averaging or *lazy mirror descent*. The basic idea is to replace the mirror descent update (MD) with

$$\nabla\Phi(y_{t+1}) = \nabla\Phi(x_t) - \eta g_t, \tag{DA}$$

with $y_{t+1} \in \mathcal{P}$, and $x_1$ is chosen such that $\nabla\Phi(x_1) = 0$. If asked to return a primal point, we project the current dual point using

$$x_t \in \underset{x \in \mathcal{X} \cap \mathcal{P}}{\arg\min} \left\{ \eta \sum_{s=1}^{t-1} \langle \tilde{g}_s, x \rangle + \Phi(x) \right\}. \qquad \text{(DA-Project)}$$

**Theorem C.2** (Quantum Dual Averaging). *Let $\mathcal{X} \subseteq \mathbb{R}^d$ be a closed and bounded, convex set equipped with a given p-norm $\|\cdot\|$ with $p \geq 1$. Denote $K := \operatorname{diam}(\mathcal{X}) = \sup_{x,y \in \mathcal{X}} \|x - y\|$. Suppose $f : \mathbb{R}^d \to \mathbb{R}$ is a convex function that is G-Lipschitz with respect to $\|\cdot\|$ and attains its minimum at $x^\star \in \mathcal{X} \cap \mathcal{P}$. Let $\Phi$ be a mirror map that is $\mu$-strongly convex on $\mathcal{X} \cap \mathcal{P}$ with respect to $\|\cdot\|$. Assume one has access to an $\theta$-precise evaluation oracle for $f$, with*

$$\theta = \mathcal{O}\left( \frac{\mu\varepsilon^5}{G^4 R^2 K^2 \vartheta_*^2 \vartheta d^3} \right),$$

*and access to a starting point $x_1 \in \mathcal{X} \cap \mathcal{P}$ such that $\nabla\Phi(x_1) = 0$, then define $R^2 = D_\Phi(x^\star, x_1)$. Consider the zeroth-order quantum lazy mirror descent method (DA) with $\eta = \frac{R}{G}\sqrt{\frac{\mu}{T}}$, that outputs the average $\frac{1}{T}\sum_{t=1}^{T} x_t$ (using Equation (DA-Project)) after running $T$ iterations, where the $\tilde{g}_t$ is an approximate element of $\partial f(x_t)$ estimated via the quantum subgradient estimation in Lemma A.1 at each iterate. Then, with probability at least $2/3$, one can obtain an $\varepsilon$-approximate solution with $\widetilde{\mathcal{O}}\left( \left(\frac{GR}{\sqrt{\mu}\varepsilon}\right)^2 \right)$ queries to $\theta$-approximate binary oracle of $f$ and $\widetilde{\mathcal{O}}\left( d\left(\frac{GR}{\sqrt{\mu}\varepsilon}\right)^2 \right)$ gates.*

*Proof.* As usual, let $\tilde{g}_t$ be the output of the quantum subgradient algorithm for $x_t$. Let $s \in \{1, \dots, T\}$ and define

$$\psi_s(x) := \eta \sum_{t=1}^{s} \langle \tilde{g}_t, x \rangle + \Phi(x)$$

Also define

$$x_s \in \underset{x \in \mathcal{X} \cap \mathcal{P}}{\arg\min} \left\{ \eta \sum_{t=1}^{s-1} \langle \tilde{g}_t, x \rangle + \Phi(x) \right\} = \underset{x \in \mathcal{X} \cap \mathcal{P}}{\arg\min} \left\{ \psi_{s-1}(x) \right\}. \qquad (22)$$

Since $\Phi$ is $\mu$-strongly convex, one has that $\psi_s$ is also $\mu$-strongly convex for all $s$. Therefore,

$$\psi_s(x_{s+1}) - \psi_s(x_s) \leq \langle \nabla\psi_s(x_{s+1}), x_{s+1} - x_s \rangle - \frac{\mu}{2}\|x_{s+1} - x_s\|^2$$

$$\leq -\frac{\mu}{2}\|x_{s+1} - x_s\|^2, \qquad (23)$$

where the second inequality follows from the first-order optimality condition. We also have

$$\psi_s(x_{s+1}) - \psi_s(x_s) = \psi_{s-1}(x_{s+1}) - \psi_{s-1}(x_s) + \eta\langle \tilde{g}_s, x_{s+1} - x_s \rangle$$

$$\geq \eta\langle \tilde{g}_s, x_{s+1} - x_s \rangle. \qquad (24)$$

Combining (23) and (24), one obtains

$$\frac{\mu}{2}\|x_{s+1} - x_s\|^2 \leq \eta\langle \tilde{g}_s, x_s - x_{s+1} \rangle \leq \eta\|\tilde{g}_s\|_*\|x_{s+1} - x_s\|$$

$$\leq \eta\left((\|\tilde{g}_t - g\|_* + G)\|x_{s+1} - x_s\|\right),$$

so following the derivation of (15), we have

$$\eta\langle \tilde{g}_s, x_s - x_{s+1} \rangle \leq \eta\|\tilde{g}_s\|_*\|x_{s+1} - x_s\|$$

$$\leq \eta\left((\|\tilde{g}_s - g\|_* + G)\|x_{s+1} - x_s\|\right) - \frac{\mu}{2}\|x_{s+1} - x_s\|^2$$

$$\leq \frac{\eta^2 G^2}{\mu} + \frac{\eta^2 \Xi_2^2}{\mu}, \qquad (25)$$

with $\Xi_2$ as in Equation (16). Next, we claim that for any $x \in \mathcal{X} \cap \mathcal{P}$ and any $r \geq 0$.

$$\sum_{s=1}^{r} \langle \tilde{g}_s, x_s - x \rangle \leq \sum_{s=1}^{r} \langle \tilde{g}_s, x_s - x_{s+1} \rangle + \frac{D_\Phi(x, x_1)}{\eta}, \qquad (26)$$

which using the definition of Bregman divergence and the fact that $\nabla \Phi(x_1) = 0$, this amounts to

$$\sum_{s=1}^{r} \langle \tilde{g}_s, x_s - x \rangle \leq \sum_{s=1}^{r} \langle \tilde{g}_s, x_s - x_{s+1} \rangle + \frac{\Phi(x) - \Phi(x_1)}{\eta}. \tag{27}$$

By rearranging, Equation (27) can be equivalently expressed as

$$\sum_{s=1}^{r} \langle \tilde{g}_s, x_{s+1} \rangle + \frac{\Phi(x_1)}{\eta} \leq \sum_{s=1}^{r} \langle \tilde{g}_s, x \rangle + \frac{\Phi(x)}{\eta}. \tag{28}$$

We prove (28) by induction on $r$, and hence prove Equation (26). Note that the result trivially holds at $r = 0$ since $x_1 \in \arg\min_{x \in \mathcal{X} \cap \mathcal{P}} \Phi(x)$. Assuming by induction the statement holds for $r - 1$, it follows that at $r$, we have

$$\sum_{s=1}^{r} \langle \tilde{g}_s, x_{s+1} \rangle + \frac{\Phi(x_1)}{\eta} \leq \langle \tilde{g}_r, x_{r+1} \rangle + \sum_{s=1}^{r-1} \langle \tilde{g}_s, x_{s+1} \rangle + \frac{\Phi(x_1)}{\eta}$$

$$\leq \langle \tilde{g}_r, x_{r+1} \rangle + \sum_{s=1}^{r-1} \langle \tilde{g}_s, x_{r+1} \rangle + \frac{\Phi(x_{r+1})}{\eta}$$

$$\leq \sum_{s=1}^{r} \langle \tilde{g}_s, x \rangle + \frac{\Phi(x)}{\eta},$$

Thus, by induction the statement holds for all $r$. The second-to-last inequality follows from the induction hypothesis, and the last inequality holds by the definition of $x_{r+1}$ (Equation (22)). Hence combining Equation (26) with Equation (25), we have for any optimizer $x^\star$ and $r = T$

$$\sum_{s=1}^{T} \langle \tilde{g}_s, x_s - x^\star \rangle \leq \frac{\eta G^2 T}{\mu} + \frac{R^2}{\eta} + \frac{\eta \Xi_2^2 T}{\mu},$$

where by definition $R^2 = D_\Phi(x^\star, x_1)$. Continuing like in the proof of Theorem C.1 with $\Xi_1$ as in Equation (14)

$$f\left(\frac{1}{T} \sum_{s=1}^{T} x_s\right) - f(x^\star) \leq \frac{1}{T} \sum_{s \in [T]} (f(x_s) - f(x^\star))$$

$$\leq \frac{1}{T} \sum_{s \in [T]} \langle \tilde{g}_s, x_s - x^\star \rangle + \Xi_1$$

$$\leq \frac{R^2}{T\eta} + \frac{\eta G^2}{\mu} + \frac{\eta \Xi_2^2}{\mu} + \Xi_1.$$

One will note that the last line is exactly Equation (15) from the proof of Theorem C.1 but divided by $T$. Hence, the same $T$, $r_1$, $\rho$, $\eta$ and $\theta$ from the proof Theorem C.1 suffice to ensure the sub-optimality is $\varepsilon$.

$\square$

## C.4  Mirror prox

Mirror prox is a variant of mirror descent that enables one to achieve a rate $1/T$ for smooth functions, and is described by the following set of equations:

$$\begin{aligned} \nabla \Phi(\bar{z}_{t+1}) = \nabla \Phi(x_t) - \eta \nabla f(x_t) && \nabla \Phi(\bar{x}_{t+1}) = \nabla \Phi(x_t) - \eta \nabla f(z_{t+1}) \\ z_{t+1} \in \arg\min_{x \in \mathcal{X} \cap \mathcal{P}} D_\Phi(x, \bar{z}_{t+1}) && \text{and} && x_{t+1} \in \arg\min_{x \in \mathcal{X} \cap \mathcal{P}} D_\Phi(x, \bar{x}_{t+1}). \end{aligned} \tag{MP}$$

**Theorem C.3** (Quantum Mirror Prox). *Let $\mathcal{X} \subseteq \mathbb{R}^d$ be a closed and bounded, convex set equipped with a given $p$-norm $\|\cdot\|$ with $p \geq 1$. Denote $K := \operatorname{diam}(\mathcal{X}) = \sup_{x,y \in \mathcal{X}} \|x - y\|$. Suppose $f : \mathbb{R}^d \to \mathbb{R}$ is a convex function that is $G$-Lipschitz and $L$-smooth with respect to $\|\cdot\|$ and attains*

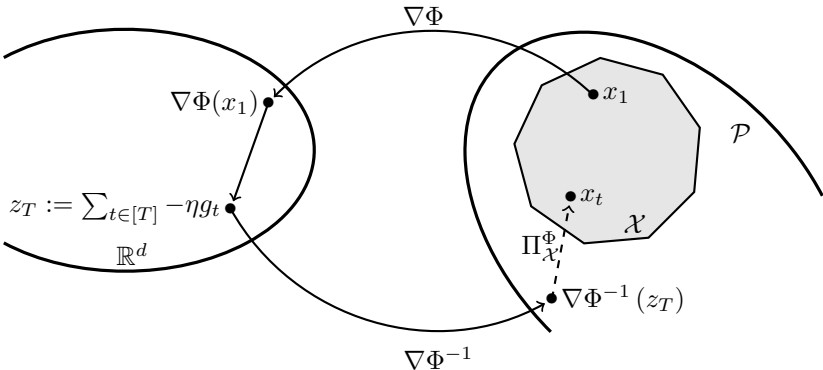

Figure 3: Dual averaging.

*its minimum at $x^\star \in \mathcal{X} \cap \mathcal{P}$. Let $\Phi$ be a mirror map that is $\mu$-strongly convex on $\mathcal{X} \cap \mathcal{P}$ with respect to $\|\cdot\|$. Suppose we are given a point $x_1 \in \mathcal{X} \cap \mathcal{P}$ and define $R^2 = D_\Phi(x^\star, x_1)$. Assume one has access to an $\theta$-precise binary evaluation oracle for $f$, with*

$$\theta = \mathcal{O}\left(\frac{\mu^4 \varepsilon^4}{L^5 \vartheta^2 \vartheta_*^4 G^2 K^4 \log\left(\sqrt{d}\vartheta_* LKG/\mu\varepsilon\right)^2}\right),$$

*Consider the zeroth-order quantum mirror prox (MP), with $\eta = \frac{\mu}{L}$, that outputs the running average $\frac{1}{T}\sum_{s=0}^{T-1} z_{s+1}$ after $T$ iterations. Then, with probability at least $2/3$, one can obtain an $\varepsilon$-approximate solution with $\widetilde{\mathcal{O}}\left(\frac{LR^2}{\mu\varepsilon}\right)$ queries to $\theta$-approximate binary oracle of $f$ and $\widetilde{\mathcal{O}}\left((d + \mathcal{T}_{next})\left(\frac{GR}{\sqrt{\mu}\varepsilon}\right)^2\right)$ gates, where $\mathcal{T}_{next}$ is the time to carry out the operations necessary to progress to the next iterate.*

*Proof.* Let $g_y$ be the estimate of the gradient $\nabla f(y)$ returned by quantum gradient estimation. Let $x^\star \in \mathcal{X} \cap \mathcal{P}$ be a minimizer. Then,

$$f(z_{t+1}) - f(x^\star)$$
$$\leq \langle \nabla f(z_{t+1}), z_{t+1} - x^\star \rangle$$
$$= \langle g_{z_{t+1}}, z_{t+1} - x^\star \rangle + \langle \nabla f(z_{t+1}) - g_{z_{t+1}}, z_{t+1} - x^\star \rangle$$
$$= \langle g_{z_{t+1}}, x_{t+1} - x^\star \rangle + \langle g_{x_t}, z_{t+1} - x_{t+1} \rangle + \langle g_{z_{t+1}} - g_{x_t}, z_{t+1} - x_{t+1} \rangle$$
$$\quad + \langle \nabla f(z_{t+1}) - g_{z_{t+1}}, z_{t+1} - x^\star \rangle.$$

We start by bounding the first three terms in the above expression.

For the first term, we have

$$\eta \langle g_{z_{t+1}}, x_{t+1} - x^\star \rangle \overset{\text{Eq. (MP)}}{=} \langle \nabla\Phi(x_t) - \nabla\Phi(\bar{x}_{t+1}), x_{t+1} - x^\star \rangle$$
$$\overset{\text{Lem. C.1}}{\leq} \langle \nabla\Phi(x_t) - \nabla\Phi(x_{t+1}), x_{t+1} - x^\star \rangle$$
$$\overset{\text{Eq. (13)}}{=} D_\Phi(x^\star, x_t) - D_\Phi(x^\star, x_{t+1}) - D_\Phi(x_{t+1}, x_t).$$

The second term is bounded using the same steps as the first, and accounting for strong convexity of the mirror map:

$$\eta \langle g_{x_t}, z_{t+1} - x_{t+1} \rangle \overset{\text{Eq. (MP)}}{=} \langle \nabla\Phi(x_t) - \nabla\Phi(\bar{z}_{t+1}), z_{t+1} - x_{t+1} \rangle$$
$$\overset{\text{Lem. C.1}}{\leq} \langle \nabla\Phi(x_t) - \nabla\Phi(z_{t+1}), z_{t+1} - x_{t+1} \rangle$$
$$\overset{\text{Eq. (13)}}{=} D_\Phi(x_{t+1}, x_t) - D_\Phi(x_{t+1}, z_{t+1}) - D_\Phi(z_{t+1}, x_t)$$
$$\leq D_\Phi(x_{t+1}, x_t) - \frac{\mu}{2}\|x_{t+1} - z_{t+1}\|^2 - \frac{\mu}{2}\|z_{t+1} - x_t\|^2.$$

For the third term, we have

$$\langle g_{z_{t+1}} - g_{x_t}, z_{t+1} - x_{t+1} \rangle$$

$$\leq \langle g_{z_{t+1}} - \nabla f(z_{t+1}), z_{t+1} - x_{t+1} \rangle + \langle g_{x_t} - \nabla f(x_t), z_{t+1} - x_{t+1} \rangle$$

$$\qquad + \langle \nabla f(z_{t+1}) - \nabla f(x_t), z_{t+1} - x_{t+1} \rangle$$

$$\leq \|\nabla f(z_{t+1}) - \nabla f(x_t)\|_* \|z_{t+1} - x_{t+1}\| + \|g_{x_t} - \nabla f(x_t)\|_* K + \|g_{z_{t+1}} - \nabla f(z_{t+1})\|_* K$$

$$\leq L\|z_{t+1} - x_t\|\|z_{t+1} - x_{t+1}\| + \|g_{x_t} - \nabla f(x_t)\|_* K + \|g_{z_{t+1}} - \nabla f(z_{t+1})\|_* K$$

$$\leq \frac{L}{2}\|z_{t+1} - x_t\|^2 + \frac{L}{2}\|z_{t+1} - x_{t+1}\|^2 + \|g_{x_t} - \nabla f(x_t)\|_* K + \|g_{z_{t+1}} - \nabla f(z_{t+1})\|_* K,$$

where we used that $K$ is a bound on the $\|\cdot\|$ diameter of $\mathcal{X} \cap \mathcal{P}$ and $z_{t+1}, x_{t+1}, x^\star \in \mathcal{X} \cap \mathcal{P}$.

Summing the above terms and setting $\eta = \frac{\mu}{L}$ gives

$$f(z_{t+1}) - f(x^\star) \leq \frac{L}{\mu} \left[ D(x^\star, x_t) - D(x^\star, x_{t+1}) \right] + \frac{LK}{\mu}\|g_{x_t} - \nabla f(x_t)\|_*$$

$$+ 2\frac{LK}{\mu}\|g_{z_{t+1}} - \nabla f(z_{t+1})\|_*.$$

Next we take the expectation of the future iterates conditioned on $z_{t+1}$:

$$f(z_{t+1}) - f(x) \leq \frac{L}{\mu} \left[ D(x^\star, x_t) - \mathbb{E}[D(x^\star, x_{t+1})|z_{t+1}] \right] + \frac{LK}{\mu}\|g_{x_t} - \nabla f(x_t)\|_*$$

$$+ 2\frac{LK}{\mu}\mathbb{E}[\|g_{z_{t+1}} - \nabla f(z_{t+1})\|_* |z_{t+1}].$$

Note that given $z_{t+1}$, $g_{x_t}$ is uniquely determined by the strong convexity of $D_\Phi$, and hence $x_t$ by MP, so $\mathbb{E}[h(x_t, g_{x_t})|z_{t+1}] = h(x_t, g_{x_t})$ for any deterministic $h$. Continuing, we have that

$$\mathbb{E}[\|g_{z_{t+1}} - \nabla f(z_{t+1})\|_* |z_{t+1}] \leq \mathbb{E}[\|g_{z_{t+1}} - \mathbb{E}[g_{z_{t+1}}|z_{t+1}]\|_* |z_{t+1}] + \|\nabla f(z_{t+1}) - \mathbb{E}[g_{z_{t+1}}|z_{t+1}]\|_*$$

$$\leq \vartheta_* \mathbb{E}[\|g_{z_{t+1}} - \mathbb{E}[g_{z_{t+1}}|z_{t+1}]\|_\infty |z_{t+1}]$$

$$+ \vartheta_* \|\nabla f(z_{t+1}) - \mathbb{E}[g_{z_{t+1}}|z_{t+1}]\|_\infty$$

$$\leq 2\vartheta_* \sigma.$$

Then we condition on $x_{t+1}$:

$$\mathbb{E}[f(z_{t+1})|x_t] - f(x) \leq \frac{D(x, x_t) - \mathbb{E}[D(x, x_{t+1})|x_t]}{\eta} + \mathbb{E}[\|g_{x_t} - \nabla f(x_t)\|_2 |x_t]K + 4\vartheta_* \frac{LK}{\mu}\sigma.$$

Similarly, we have

$$\mathbb{E}[\|g_{x_t} - \nabla f(x_t)\|_* |x_t] \leq 2\vartheta_* \sigma.$$

Taking expectation over everything and using the tower law, we get:

$$\mathbb{E}[f(z_{t+1})] - f(x^\star) \leq \frac{L}{\mu} \left[ \mathbb{E}[D(x, x_t)] - \mathbb{E}[D(x^\star, x_{t+1})] \right] + 6\vartheta_* \frac{LK}{\mu}\sigma.$$

Averaging over and using convexity, we have

$$\mathbb{E}[f\left(\sum_{t=1}^{T} \frac{z_t}{T}\right) - f(x^\star)] \leq \frac{1}{T}\sum_{t=1}^{T} \mathbb{E}[f(z_t) - f(x^\star)]$$

$$\leq \frac{L\left(D(x^\star, x_1) - \mathbb{E}[D(x^\star, x_T)]\right)}{\mu T} + 6\vartheta_* \frac{LK}{\mu}\sigma$$

$$\leq \frac{L\left(D(x^\star, x_1)\right)}{\mu T} + 6\vartheta_* \frac{LK}{\mu}\sigma$$

$$\leq \frac{LR^2}{\mu T} + 6\vartheta_* \frac{LK}{\mu}\sigma.$$

Now, say we set $T = \mathcal{O}\left(\frac{LR^2}{\mu\varepsilon}\right)$ and $\sigma = \mathcal{O}\left(\frac{\varepsilon\mu}{\vartheta_* LK}\right)$. We can then apply Theorem A.2 to get the required precision

$$\theta = \Theta\left(\frac{\mu^4\varepsilon^4}{L^5\vartheta^2\vartheta_*^4 G^2 K^4 \log\left(\sqrt{d}\vartheta_* LKG/(\mu\varepsilon)\right)^2}\right).$$

If use our choice of $T$ and apply Markov's inequality, then we get

$$\Pr\left[f\left(\sum_{t=0}^{T-1}\frac{z_{t+1}}{T}\right) - f(x^\star) \geq \varepsilon\right] \leq \frac{1}{3}.$$

$\square$

# D    Application to SDPs, LPs and zero-sum games

## D.1    From SDP to eigenvalue optimization

Following a discussion found in [81, Chapter 11.1], we will review how the dual problem (D) can be reformulated as an eigenvalue optimization problem. We are interested in solving the primal and dual SDPs

$$\sup_{X\in\mathbb{R}^{n\times n}}\left\{\mathrm{tr}(CX) : \mathrm{tr}(X) = r_p, \mathrm{tr}\left(A_i X\right) \leq b_i \text{ for all } i \in [m], X \succeq 0\right\}, \tag{P}$$

and

$$\inf_{(y_0,y)\in\mathbb{R}\times\mathbb{R}^m_{\geq 0}}\left\{r_p y_0 + b^\top y : y_0 I + \sum_{i\in[m]} y_i A_i - C \succeq 0\right\}, \tag{D}$$

where $A_0 = I$, $b_0 = r_p$ and $\|A_1\|_{\mathrm{op}}, \ldots, \|A_m\|_{\mathrm{op}}, \|C\|_{\mathrm{op}} \leq 1$, $\|b\|_\infty \leq r_p$.

We say that $(X, y_0, y) \in \mathbb{R}^{n\times n} \times \mathbb{R} \times \mathbb{R}^m_{\geq 0}$ is a *primal-dual feasible* solution if

$$\mathrm{tr}(X) = r_p, \quad \mathrm{tr}\left(A_i X\right) \leq b_i \text{ for all } i \in [m], \quad X \succeq 0, \quad y_0 I + \sum_{i\in[m]} y_i A_i - C \succeq 0.$$

A primal-dual feasible solution $(X, y_0, y)$ is *strictly feasible* when $X$ and $y_0 I + \sum_{i\in[m]} y_i A_i - C$ are positive definite.

The primal problem (P) is strictly feasible by assumption. The dual problem (D) is also strictly feasible since $\|C\|_{\mathrm{op}} \leq 1$ and $A_0 = I$, so *strong duality* holds. Consequently, any *primal-dual optimal* solution $(X^\star, y_0^\star, y^\star)$ to (P)-(D) has zero duality gap:

$$\mathrm{tr}(CX^\star) - r_p y_0^\star - b^\top y^\star = 0,$$

and the optimal objective value $\mathsf{OPT} := \mathrm{tr}(CX^\star) = r_p y_0^\star + b^\top y^\star$ is finite and attained. By Hölder's inequality, one has

$$|\mathsf{OPT}| = |\mathrm{tr}(CX^\star)| \leq \|C\|_{\mathrm{op}}\|X^\star\|_{\mathrm{tr}} \leq r_p,$$

since $\|C\|_{\mathrm{op}} \leq 1$, $\mathrm{tr}(X^\star) = r_p$ and $X^\star \succeq 0$. We may thus assume without loss of generality that there exists a finite value $r_d \geq 1$ such that $\|(y_0, y^\star)\|_1 \leq r_d$ for any dual optimal solution $(y_0^\star, y^\star) \in \mathbb{R} \times \mathbb{R}^m_{\geq 0}$.

Upon normalizing $(b_0, b)^\top \in \mathbb{R}^{m+1}$ with respect to its $\ell_\infty$-norm $(1, \tilde{b})^\top \mapsto 1/r_p(b_0, b)^\top$, it is easy to see that the set of primal-feasible solutions is a subset of the spectraplex

$$\mathbb{S}^n := \{X \in \mathbb{R}^{n\times n} : \mathrm{tr}(X) = 1, X \succeq 0\}.$$

Hence, the primal and dual SDPs (P)-(D) may be written as

$$\max_{X\in\mathbb{S}^n}\left\{\mathrm{tr}(CX) : \mathrm{tr}\left(A_i X\right) \leq \tilde{b}_i \text{ for all } i \in [m]\right\}, \tag{$\tilde{\mathrm{P}}$}$$

and

$$\min_{(y_0, y) \in \mathbb{R} \times \mathbb{R}^m_{\geq 0}} \left\{ y_0 + \tilde{b}^\top y : y_0 I + \sum_{i \in [m]} y_i A_i - C \succeq 0 \right\}. \tag{$\tilde{\mathrm{D}}$}$$

The optimal objective value of the normalized pair $(\tilde{\mathrm{P}})$-$(\tilde{\mathrm{D}})$ is $\mathsf{OPT}/r_p \in [-1, 1]$, and the sets of optimal solutions to these problems are equivalent up to a constant scaling: if $(X^\star, y_0^\star, y^\star)$ is optimal for $(\tilde{\mathrm{P}})$-$(\tilde{\mathrm{D}})$, then $(RX^\star, y_0^\star, y^\star)$ is optimal for (P)-(D).

The Karush-Kuhn-Tucker (KKT) optimality conditions for $(\tilde{\mathrm{P}})$-$(\tilde{\mathrm{D}})$ (and hence, (P)-(D)) stipulate that a primal-dual feasible solution $(X^\star, y_0^\star, y^\star)$ is optimal if and only if

$$X^\star Z^\star = Z^\star X^\star = 0,$$

where $Z^\star := y_0^\star I + \sum_{i \in [m]} y_i^\star A_i - C$. As a consequence, any optimal $X^\star$ and $Z^\star$ are simultaneously diagonalizable. Letting $P \in \mathbb{R}^{n \times n}$ be an orthonormal matrix, we may write

$$X^\star = P \Lambda_{X^\star} P^\top, \quad Z^\star = P \Lambda_{Z^\star} P^\top,$$

where $\Lambda_{X^\star}$ and $\Lambda_{Z^\star}$ are diagonal matrices whose entries are the nonnegative eigenvalues of $X^\star$ and $Z^\star$, respectively. Since $X^\star \neq 0$ (as $\operatorname{tr}(X^\star) \neq 0$) and $\Lambda_{X^\star} \Lambda_{Z^\star} = 0$, it follows

$$0 = \lambda_{\min}(\Lambda_{Z^\star}) = \lambda_{\min}\left( y_0^\star I + \sum_{i \in [m]} y_i^\star A_i - C \right) = y_0^\star + \lambda_{\min}\left( \sum_{i \in [m]} y_i^\star A_i - C \right).$$

Rearranging terms, we obtain

$$y_0^\star = -\lambda_{\min}\left( \sum_{i \in [m]} y_i^\star A_i - C \right) = \lambda_{\max}\left( C - \sum_{i \in [m]} y_i^\star A_i \right), \tag{29}$$

and thus solving $(\tilde{\mathrm{D}})$ is *equivalent* to solving $\min_{y \in \mathbb{R}^m_{\geq 0}} f(y)$, where

$$f(y) := \lambda_{\max}\left( C - \sum_{i \in [m]} y_i A_i \right) + \tilde{b}^\top y. \tag{SDP-eig}$$

Note that this reduction is valid for any nontrivial SDP with a bounded feasible set [42, 81]. The linear term $\tilde{b}^\top y$ may also be incorporated into the $\lambda_{\max}(\cdot)$ term by replacing each $A_i$ with $A_i - \tilde{b}_i I$, see [70, Section 6.3].

Problem (SDP-eig) is a nonsmooth convex optimization problem and is well-studied in the literature, see e.g., [52] and the references therein. Before proceeding further, we recite a few important facts to keep the paper self-contained.

First, note that since $b_0/r_p = 1$, by (29) one has

$$f(y^\star) = \lambda_{\max}\left( C - \sum_{i \in [m]} y_i^\star A_i \right) + \tilde{b}^\top y^\star = \frac{1}{r_p}\left( r_p y_0^\star + b^\top y^\star \right)$$

$$:= \frac{1}{r_p} \min_{\left\{ (y_0, y) \in \mathbb{R} \times \mathbb{R}^m_{\geq 0} : y_0 I + \sum_{i \in [m]} y_i A_i - C \succeq 0 \right\}} r_p y_0 + b^\top y = \mathsf{OPT}/r_p.$$

Since dual optimal solutions $(y_0^\star, y^\star)$ to $(\tilde{\mathrm{D}})$ satisfy $\|(y_0^\star, y^\star)^\top\|_1 \leq r_d$, and $y^\star$ is also an optimal solution to (SDP-eig), it follows that (SDP-eig) has a global minimizer in the dilated $m$-dimensional simplex

$$r_d \Delta^m := \left\{ y \in \mathbb{R}^m_{\geq 0} : \sum_{i \in [m]} y_i = r_d \right\}.$$

This motivates us to solve (SDP-eig) with mirror descent using the simplex setup described earlier in Section C.2.

The complexity of our algorithm will scale with the Lipschitz constant of the objective in (SDP-eig). The upshot of using mirror descent is that we can choose to work with the norm in which $f$ is well-behaved.

**Lemma D.1.** *The objective function $f : \mathbb{R}^m \to \mathbb{R}$ in* (SDP-eig) *is 2-Lipschitz on $\mathbb{R}^m$ with respect to the $\ell_1$-norm.*

*Proof.* The idea is to consider the two terms defining $f$ separately, and sum their Lipschitz constants. By Hölder's inequality, the Lipschitz constant of the linear term $\tilde{b}^\top y$ is simply $\|\tilde{b}\|_\infty \leq 1$.

For the other term, let $M(y) = C - \sum_{i \in [m]} y_i A_i$, which is a Hermitian matrix for all $y \in \mathbb{R}^m$. Thus, for any $(y, \bar{y}) \in \mathbb{R}^m \times \mathbb{R}^m$, one has

$$|\lambda_{\max}(M(y)) - \lambda_{\max}(M(\bar{y}))| \leq \|M(y) - M(\bar{y})\|_{\mathrm{op}} = \sum_{i \in [m]} |y_i - \bar{y}_i| \, \|A_i\|_{\mathrm{op}}$$
$$\leq \max_{i \in [m]} \{\|A_i\|_{\mathrm{op}}\} \|y - \bar{y}\|_1,$$

where the first inequality is a consequence of Weyl's Perturbation Theorem for Hermitiain matrices. Since $\|A_1\|_{\mathrm{op}}, \dots, \|A_m\|_{\mathrm{op}} \leq 1$, we have that $\lambda_{\max}(M(y))$ is 1-Lipschitz with respect to the $\ell_1$-norm.

The proof is complete upon summing the Lipschitz constants of $\lambda_{\max}(M(y))$ and $\tilde{b}^\top y$. $\qquad\square$

We are now in a position to establish the (black-box) complexity of determining an $\varepsilon$-optimal solution to the dual SDP (D) using a quantum mirror descent method.

**Theorem D.1.** *Let $f : \mathbb{R}^m \to \mathbb{R}$ be the objective function in* (SDP-eig)*, and let $r_p, r_d \geq 1$ be such that an optimal solution to the SDP* (P)-(D) *satisfies $\mathrm{tr}(X^\star) = r_p$ and $\|y^\star\|_1 \leq r_d$. Suppose that one has access to an $\theta$-precise binary evaluation oracle to $f$, with $\theta = \widetilde{\mathcal{O}}(\varepsilon^5/r_p^5 r_d^5 m^{4.5})$. Then, with probability at least $2/3$, the quantum mirror descent method described in Theorem C.1 solves* (P)-(D) *to additive error $\varepsilon \in (0, 1)$ using at most $\widetilde{\mathcal{O}}\left(\left(\frac{r_p r_d}{\varepsilon}\right)^2\right)$ queries to a binary evaluation oracle for $f$ and $\widetilde{\mathcal{O}}\left(m\left(\frac{r_p r_d}{\varepsilon}\right)^2\right)$ gates.*

*The output is a vector $\tilde{y}^\star \in \mathbb{R}^m_{\geq 0}$ such that*

$$(y_0^\star, y^\star) = \left(-\lambda_{\min}\left(\sum_{i \in [m]} (r_d \cdot \tilde{y}_i^\star) A_i - C\right), r \tilde{y}^\star\right) \in \mathbb{R} \times \mathbb{R}^m_{\geq 0}$$

*is a feasible solution to* (D) *that satisfies*

$$b_0 y_0^\star + b^\top y^\star \leq \mathsf{OPT} + \varepsilon.$$

*In other words, $(y_0^\star, y^\star) \in \mathbb{R} \times \mathbb{R}^m_{\geq 0}$ is an $\varepsilon$-precise solution to* (D)*.*

*Proof.* Let $(A_1, \dots, A_m, b, C)$ be the input data defining (P)-(D), and denote the optimal objective value by $\mathsf{OPT}$. We first normalize $\tilde{b} \mapsto b/r_p$ and re-scale the input space $\tilde{y} \mapsto y/r_d$. With these modifications, the result in Lemma D.1 certifies that $f$ is 2-Lipschitz over $\mathbb{R}^m$ with respect to the $\ell_1$-norm, and the optimal objective value of the normalized problem is $\mathsf{OPT}/r_p r_d$. Accordingly, if we seek to approximate the optimal objective value of (P)-(D) to additive error $\varepsilon \in (0, 1)$, we must solve

$$\min_{\tilde{y} \in \Delta^m} f(\tilde{y}) := \lambda_{\max}\left(\frac{1}{r_d} C - \sum_{i \in [m]} \tilde{y}_i A_i\right) + \tilde{b}^\top \tilde{y}$$

to precision $\varepsilon' := \varepsilon/r_p r_d$.

The basic idea is to utilize the simplex setup for mirror descent, taking the negative entropy function as the mirror map $\Phi(\tilde{y}) = \sum_{i \in [m]} \tilde{y}_i \log \tilde{y}_i$, and choosing the starting point $\tilde{y} = m^{-1} \mathbf{1}_m \in \Delta^m_{>0}$. Recalling that $\Phi$ is 1-strongly convex on $\Delta^m$, and that our choice of starting point ensures $R = \mathcal{O}(\log(m))$, the result in Theorem C.1 (taking $\vartheta = 1$ since $\|\cdot\|_* = \|\cdot\|_\infty$) establishes that one can determine $\tilde{y}^\star \in \Delta^m$ satisfying

$$f(\tilde{y}^\star) \leq \frac{\mathsf{OPT}}{r_p r_d} + \frac{\varepsilon}{r_p r_d} \implies r_p r_d \cdot f(\tilde{y}^\star) \leq \mathsf{OPT} + \varepsilon, \tag{30}$$

using at most

$$Q_f = \widetilde{\mathcal{O}}\left(\left(\frac{1}{\varepsilon'}\right)^2\right) = \widetilde{\mathcal{O}}\left(\left(\frac{r_p r_d}{\varepsilon}\right)^2\right)$$

queries to the $\theta$-precise binary evaluation oracle for $f$ and $\widetilde{\mathcal{O}}\left((m + \mathcal{T}_{\text{next}})\left(\frac{r_p r_d}{\varepsilon}\right)^2\right)$ gates, where $\mathcal{T}_{\text{next}}$ represents the cost to proceed to the next iterate.

To make the gate complexity explicit, note that $\widetilde{\mathcal{O}}(m)$ work is required to proceed to the next iterate. After estimating $\tilde{g}_t \in \mathbb{R}^m$ at iterate $t$, we compute the entries of $\tilde{y}_{t+1} \in \Delta^m$ according to (18), setting

$$\tilde{y}_{t+1,i} = \frac{\tilde{y}_{t,i} e^{-\eta g_{t,i}}}{\sum_{i \in [m]} \tilde{y}_{t,i} e^{-\eta g_{t,i}}},$$

which clearly requires $\widetilde{\mathcal{O}}(m)$ operations. One can also reduce storage requirements by recursively maintaining the running average as the convex combination:

$$\frac{1}{t+1}\sum_{s \in [t+1]} \tilde{y}_s = \frac{t}{t+1}\tilde{y}_t + \frac{1}{t+1}\tilde{y}_{t+1}.$$

Finally, we show that our output $\tilde{y}^\star$ encodes an $\varepsilon$-precise solution to the dual SDP (D). Define $y^\star := r_d \tilde{y}^\star \in \mathbb{R}^m_{\geq 0}$ and set

$$y_0^\star = -\lambda_{\min}\left(\sum_{i \in [m]} y_i^\star A_i - C\right) \in \mathbb{R}.$$

Then, $y_0^\star I + \sum_{i \in [m]} y_i^\star A_i - C \succeq 0$ (in fact, $\lambda_{\min}(y_0^\star I + \sum_{i \in [m]} y_i^\star A_i - C) = 0$), so $(y_0^\star, y^\star) \in \mathbb{R} \times \mathbb{R}^m_{\geq 0}$ is a feasible solution to (D). Applying (30), one can also observe

$$r_p y_0^\star + b^\top y^\star = r_p\left(y_0^\star + \tilde{b}^\top y^\star\right) = r_p r_d\left[\lambda_{\max}\left(\frac{1}{r_d}C - \sum_{i \in [m]} \tilde{y}_i^\star A_i\right) + \tilde{b}^\top \tilde{y}^\star\right] = r_p r_d \cdot f(\tilde{y}^\star)$$

$$\leq \mathsf{OPT} + \varepsilon.$$

$\square$

Thus far we have established the query complexity of our algorithm applied to solving SDPs. However, these queries are to a black-box evaluation oracle for the objective function $f$. In order to make our results comparable to other SDP solvers found in the literature, we need to open the black-box and establish the cost of each of these evaluations in terms of queries to the input data defining (P)-(D). This is what we do next.

### D.2 Complexity

#### D.2.1 Semidefinite programs

**Theorem D.2** (SDP Solver). *Suppose that the SDP problem data* $(A_1, \ldots, A_m, b, C)$ *is stored in a sparse-access data structure. Choose* $\varepsilon \in (0, 1)$. *Then, with probability* $2/3$, *the quantum mirror descent method described in Theorem C.1 solves* (P)-(D) *to additive error* $\varepsilon$ *using*

$$\mathcal{O}\left((mns + n^\omega)(r_p r_d/\varepsilon)^2 \cdot \text{polylog}(m, n, (r_p r_d/\varepsilon))\right)$$

*queries and gates. The output is an* $\varepsilon$-*precise solution* $(y_0^\star, y^\star) \in \mathbb{R} \times \mathbb{R}^m_{\geq 0}$ *to the dual SDP* (D).

*Proof.* We first discuss the cost of implementing the evaluation oracle.

To compute the inner product $b^\top y$, we need $\mathcal{O}(m \log(1/\theta))$ queries to the elements of $b$. To see this, note that $\mathcal{O}(m \log(1/\theta))$ queries are necessary to implement the unitary

$$U_b \ket{z} \mapsto \ket{z \oplus b}.$$

With access to $U_b$, one can compute the inner product as

$$|0\rangle |y_i\rangle |0\rangle \overset{U_b}{\mapsto} |b\rangle |y_i\rangle |0\rangle \overset{\widetilde{\mathcal{O}}(m)\text{ gates}}{\mapsto} |b\rangle |y_i\rangle |b^\top y\rangle \overset{U_b^\dagger}{\mapsto} |y_i\rangle |b^\top y\rangle .$$

Since $U_b^\dagger = U_b$, the above circuit requires 2 applications of $U_b$, and hence, $\mathcal{O}(m\log(1/\theta))$ queries to the elements of $b$ in total. Note that the total number of gates is also $\mathcal{O}(m\log(1/\theta))$.

Implementing the matrix $M = C - \sum_{i\in[m]} y_i A_i$ from $y$ requires $\mathcal{O}(mns)$ gates, and from here we can perform an eigendecomposition on $M$ to find the top eigenvalue using $\mathcal{O}(n^\omega \log(1/\theta))$ gates. Hence, in the high-precision regime, we can implement an $\theta$-precise evaluation oracle for $f$ using

$$\mathcal{O}((mns + n^\omega)\log(1/\theta))$$

queries and gates.

Setting $\theta = \widetilde{\mathcal{O}}(\varepsilon^5/r_p^5 r_d^5 m^{4.5})$ and applying Theorem D.1, the total number of queries can be bounded as

$$\widetilde{\mathcal{O}}\left((mns + n^\omega)\left(\frac{r_p r_d}{\varepsilon}\right)^2\right) = \mathcal{O}\left((mns + n^\omega)\left(\frac{r_p r_d}{\varepsilon}\right)^2 \cdot \mathrm{polylog}\left(m, n, \frac{r_p r_d}{\varepsilon}\right)\right)$$

and the number of gates is

$$\widetilde{\mathcal{O}}\left((mns + n^\omega + m)\left(\frac{r_p r_d}{\varepsilon}\right)^2\right) = \mathcal{O}\left((mns + n^\omega)\left(\frac{r_p r_d}{\varepsilon}\right)^2 \cdot \mathrm{polylog}\left(m, n, \frac{r_p r_d}{\varepsilon}\right)\right).$$

The theorem statement is obtained upon noting that, given $\tilde{y}^\star$, one can compute

$$(y_0^\star, y^\star) = \left(-\lambda_{\min}\left(\sum_{i\in[m]} (r\tilde{y}_i^\star)A_i - C\right), r\tilde{y}^\star\right) \in \mathbb{R} \times \mathbb{R}_{\geq 0}^m$$

using at most $\widetilde{\mathcal{O}}(mns + n^\omega)$ operations. $\qquad\square$

### D.2.2 Linear programs

Recall that LPs are a special case of SDPs in which each of the input matrices $A_1, \ldots, A_m, C$ is a diagonal matrix. In this case, the dual slack matrix $Z$ is also a diagonal matrix. Thus for LPs, we may write the objective function in (SDP-eig) as

$$f(y) = \lambda_{\max}\left(C - \sum_{i\in[m]} y_i A_i\right) + \tilde{b}^\top y = \max_{j\in[n]}\left\{\left(C - \sum_{i\in[m]} y_i A_i\right)_{jj}\right\} + \tilde{b}^\top y.$$

One can leverage the simplified structure of the objective for LPs to obtain an attractive running time, as we show next.

**Theorem D.3** (LP solver). *Let $(A, b, c)$ be the input data for an LP with $m$ constraints and $n$ variables. Suppose that the columns of $A$ and $c$ are stored in a quantum read-only memory. Then, with probability $2/3$, the quantum mirror descent method described in Theorem C.1 solves the primal-dual LP pair in* (LP) *to additive error $\varepsilon$ using*

$$\mathcal{O}\left(m\sqrt{n}\,(r_p r_d/\varepsilon)^2 \cdot \mathrm{polylog}\,(m, n, r_p r_d/\varepsilon)\right)$$

*queries and gates. The output is an $\varepsilon$-precise solution to the dual LP problem.*

*Proof.* As discussed earlier in the proof of Theorem D.2, we can compute the linear term $\tilde{b}^\top y$ using $\widetilde{\mathcal{O}}(m)$ gates.

For the other term, recall that in LP we need to evaluate

$$\max_{j\in[n]}\left\{\left(C - \sum_{i\in[m]} y_i A_i\right)_{jj}\right\} = \min_{j\in[n]}\left\{\left(\sum_{i\in[m]} y_i A_i - C\right)_{jj}\right\}.$$

To do so, we proceed as follows. Letting $j \in [n]$, we implement $U_M$, which acts as

$$|j\rangle \, |y\rangle \, |0\rangle \, |0\rangle \, |0\rangle \stackrel{\widetilde{\mathcal{O}}(1) \text{ QROM queries}}{\mapsto} |j\rangle \, |y\rangle \, |A_j\rangle \, |c_j\rangle \, |0\rangle \stackrel{\widetilde{\mathcal{O}}(m) \text{ gates}}{\mapsto} |j\rangle \, |y\rangle \, |\langle A_j, y\rangle - c_j\rangle \, |c_j\rangle \, |0\rangle,$$

The above requires $\widetilde{\mathcal{O}}(m)$ gates. With this construction, we can apply generalized minimum finding [76, Theorem 49] to compute to smallest entry with $\mathcal{O}(\sqrt{n})$ applications of $U_M$ and its inverse.

From here, setting $\theta = \widetilde{\mathcal{O}}(\varepsilon^5 / r_p^5 r_d^5 m^{4.5})$ and applying Theorem D.1 as we did in the proof of Theorem D.2 suffices to bound the query and gate complexity of determining $\tilde{y}^\star$ using the mirror descent method. To complete the proof note that we can construct an $\varepsilon$-precise solution $(y_0^\star, y^\star) \in \mathbb{R} \times \mathbb{R}_{\geq 0}^m$ to the dual LP from $\tilde{y}^\star \in \mathbb{R}_{\geq 0}^m$ with cost $\widetilde{\mathcal{O}}(m\sqrt{n})$. Clearly, computing $y^\star = r_d \cdot \tilde{y}^\star$ takes $\mathcal{O}(m)$ operations, and one can compute

$$y_0^\star = -\min_{j \in [n]} \left\{ \left( \sum_{i \in [m]} y_i^\star A_i - C \right)_{jj} \right\}$$

with cost $\widetilde{\mathcal{O}}(m\sqrt{n})$, using the same strategy we employed to evaluate $f$. $\qquad\square$

### D.2.3 Zero-sum games

*Zero-sum games* are matrix games in which each player has a finite number of pure strategies. These problems are well-studied in the game theory literature, and a standard setup concerns two players Alice and Bob whose action spaces are $[m]$ and $[n]$ respectively. Payoffs from the game are encoded in the entries of a matrix $A \in [-1, 1]^{m \times n}$. If Alice plays action $i \in [m]$ and Bob plays $j \in [n]$, then Alice obtains the payoff $A_{ij}$, while Bob receives a payoff of $-A_{ij}$. Each player aims to maximize their expected payoff through randomized strategies $(x, y) \in \Delta^n \times \Delta^m$, giving rise to the minimax optimization problem:

$$\min_{x \in \Delta^n} \max_{y \in \Delta^m} y^\top A x. \tag{ZSG}$$

Saddle points of (ZSG) are called *mixed Nash equilibria*, which always exist for zero-sum games due to von Neumann's minimax theorem [67].

Zero-sum games can be reformulated as LPs and vise-versa. The general idea follows from the fact that a linear function over the simplex attains its maximum on a vertex, i.e.,

$$\min_{x \in \Delta^n} \max_{y \in \Delta^m} y^\top A x = \min_{x \in \Delta^n} \max_{i \in [m]} e_i^\top A x,$$

where $e_i \in \mathbb{R}^m$ denotes the $i$-th unit vector in the standard orthonormal basis for $\mathbb{R}^m$. Therefore, upon introducing one additional variable we can equivalently solve the LP:

$$\min_{(\lambda_p, x) \in [-1,1] \times \Delta^n} \left\{ \lambda_p : Ax \leq \lambda_p \mathbf{1}_n \right\},$$

where $\mathbf{1}_n \in \mathbb{R}^n$ is the all-ones vector of length $n$. The dual is similarly formulated as

$$\max_{(\lambda_d, y) \in [-1,1] \times \Delta^m} \left\{ \lambda_d : A^\top y \geq \lambda_d \mathbf{1}_m \right\}.$$

Note that the optimal objective values of these problems are equivalent under strong duality, and so we call $\lambda^*$ the value of the game. We also have $r_p, r_d \leq 2$. These observations motivate the following corollary of Theorem D.3.

**Corollary D.1** (Zero-sum games). *Let $A \in [-1, 1]^{m \times n}$ be the payoff matrix of a zero-sum game. Suppose that the columns of $A$ and $c$ are stored in a quantum read-only memory. Choose $\varepsilon \in (0, 1)$. Then, with probability $2/3$, the quantum mirror descent method described in Theorem C.1 solves* (ZSG) *to additive error $\varepsilon$ using*

$$\mathcal{O}\left( \frac{m\sqrt{n}}{\varepsilon^2} \operatorname{polylog}(m, n, 1/\varepsilon) \right)$$

*queries and gates.*

