# OpenReview forum: "Fast Zeroth-Order Convex Optimization with Quantum Gradient Methods"
_NeurIPS.cc/2025/Conference — NeurIPS 2025 poster_

### Official Review · Reviewer_7YDD · 2025-06-07

**Clarity:** 3
**Significance:** 3
**Originality:** 3
**Rating:** 4
**Confidence:** 4

**Summary:**

This paper studies quantum algorithms for zeroth-order convex optimization using quantum gradient estimation with noisy function evaluation oracles and shows that by relying solely on noisy function value queries, the quantum algorithms match the first-order query complexities of classical gradient descent methods. The framework are then generalized to non-Euclidean settings via quantum mirror descent and its variants. Leveraging connections between SDP and eigenvalue optimization, the framework is applied to solve white-box problems, including SDPs, LPs, and zero-sum games, and achieves a better time complexity on the dependences of $r_d, r_p$ and $\varepsilon$.  For smooth objectives with Lipschitz gradients, quantum gradient descent and mirror prox achieve convergence rates matching classical first-order methods. A fast query complexity is also proven under the PL condition assumption.

**Questions:**

Suggestions:
1. In the white-box section, the authors' algorithms achieve improvements in terms of $r_d$, $r_p$, and precision $\varepsilon$, but exhibit slightly worse dimensionality dependence compared to other algorithms. Though such phenomena are common in recent works of quantum optimization, the authors are suggested to briefly explain the source of the additional dimensionality dependence.

2. In Theorem D.3 and Corollary D.1, the authors are advised to explicitly specify the form of the error rather than solely referring to it as "additive error".

3. The caption of the result table is suggested to include the meaning of symbols and explicitly highlight which factors demonstrate quantum advantage.

4. The statement about dimension independence may not be entirely rigorous. Under the error function value black-box setting, the quantum gradient estimation method still exhibits a logarithmic dependence on the dimension in its query complexity. Based on my knowledge, the quantum community only proved that in the error-free black-box setting, the bounded error quantum gradient estimation can achieve a query complexity of complete dimension independence.

5. If I understand right, the "domain" in Table 1 refers to a normed space, not the feasible domain. The authors are suggested to add a brief clarification to avoid potential confusion.

Questions:
1. Regarding SDP problems, the quantum algorithms in this work output a dual solution under additive error, is it possible to derive error bounds for primal solutions?

2. Selecting an appropriate mirror map is one of the crucial steps in implementing quantum mirror descent. Note that the authors provided several examples (e.g., for feasible domains like the probability simplex and Schatten-1 norm). For other feasible domains, are there general methodologies for choosing the mirror map and the Bregman divergence?

3. Regarding the result in Table 1, is there any classical or quantum lower bound?

**Ethical Concerns:**

["NO or VERY MINOR ethics concerns only"]

**Final Justification:**

This work give a few methods where quantum gradient estimation replaces classical first-order information, and analyzes the error requirement of quantum zeroth-order black-box for preserving the same rates as classical first-order method. This provides a theoretical foundation for future physical implementations. The white-box acceleration, however, is conditional (requiring m ≪ n), leaving considerable room for future investigation of applications. Overall, I maintain my original score and lean toward acceptance.

**Limitations:**

yes

**Paper Formatting Concerns:**

The hyperlinks in the manuscript appear not to be properly set, which cause some inconvenient for reading. Fortunately, the author has provided a complete version with correctly configured hyperlinks in the supplementary materials. The authors is suggested to submit the complete version directly upon submission for the next time, as appendices are not counted against the main text's page limit.

**Quality:**

3

**Strengths And Weaknesses:**

In previous work, quantum gradient estimation methods have demonstrated the capability for zeroth-order black-box approaches to match the performance of classical first-order black-box methods. However, when applied to concrete optimization problems, quantum advantages may fail to materialize—particularly in the presence of erroneous black-box settings. Therefore, rigorous analysis and ensuring correct parameter choices are both challenging and significant.

This work provides a rigorous proof that quantum gradient estimation with erroneous black-boxes, when employed as a gradient estimator subroutine in projected gradient descent and mirror descent, maintains the original convergence rate. This establishes a solid theoretical foundation for broader applications. Furthermore, the authors demonstrate quantum advantages in applications to SDPs, LPs, and zero-sum games, achieving superior complexity compared to prior approaches relying on the HHL algorithmic family. Consequently, this work holds substantial theoretical value and lays an important groundwork for future investigation to the application of quantum gradient estimation.

---

> ### Author Rebuttal · Authors · 2025-07-31
>
> Dear Reviewer 7YDD,
>
> Thank you for your constructive comments and feedback. We appreciate your positive comments on the rigor, significance, and theoretical value of our work. In the following, we address your suggestions and questions in detail.
>
> > White-box results exhibit slightly worse dimensionality dependence compared to other algorithms. Though such phenomena are common in recent works of quantum optimization, the authors are suggested to briefly explain the source of the additional dimensionality dependence.
>
> Thank you for the suggestion, and we will include a longer explanation in an updated manuscript.
> Indeed, our white-box results indicate a worse dimensionality dependence compared to other \textit{quantum} algorithms, yet, the comparison requires more subtlety since these other quantum algorithms also have worse dependence on the error and other problem-dependent parameters. As a consequence our white box results present a complimentary paradigm to these other quantum algorithms which are often based on quantum matrix multiplicative weight updates.  The source of our increased dimension dependence is largely due to the fact that our black-box results require highly accurate function evaluations, and so the subroutines we use to instantiate white-box versions of these black-box algorithms cannot tolerate $\text{poly} (1/\varepsilon)$ scaling (we need $\text{polylog} (1/\varepsilon)$ scaling). Though this results in a higher dimension dependence, our dependence on the scale-invariant error term $r_p r_d/ \varepsilon$ is significantly better. This is crucial, since both $r_p$ and $r_d$ can scale polynomially with $n$ and $m$, respectively. On the other hand, we also show that even for applications in which $r_p$ and $r_d$ are of size $\tilde{O} (1)$, there exist regimes in which our algorithms outperform the quantum algorithms that appear to have superior scaling with respect to $m$ and $n$.
>
> > Theorem D.3 and Corollary D.1, the authors are advised to explicitly specify the form of the error rather than solely referring to it as "additive error".
>
> We thank the reviewer for their suggestion to improve the clarity of our results. In an updated version of the manuscript, we will replace the closing sentence of Theorem D.3, which originally read  "The output is an $\varepsilon$-precise solution to the dual LP problem." with "The output is a classical description of a vector $y \in \mathbb{R}^{m}_{\geq 0}$ satisfying $A^{\top} y - c \leq 0$ and $b^{\top} y \leq \text{OPT} + \varepsilon$." We will also update the statement found in Corollary D.1 in a similar fashion.
>
> > The caption of the result table is suggested to include the meaning of symbols and explicitly highlight which factors demonstrate quantum advantage.
>
> Thank you for the suggestion. We will change the main text accordingly to improve the clarity of our results.
>
> > The statement about dimension independence may not be entirely rigorous. Under the error function value black-box setting, the quantum gradient estimation method still exhibits a logarithmic dependence on the dimension in its query complexity. Based on my knowledge, the quantum community only proved that in the error-free black-box setting, the bounded error quantum gradient estimation can achieve a query complexity of complete dimension independence.
>
> Thank you for the suggestion. Indeed the gradient estimates incur a logarithmic dependence, which is reflected in the theorems of the paper. We will make sure to say (near) dimension-dependence instead of dimension-dependence in updated versions of the manuscript.
>
> - If I understand right, the "domain" in Table 1 refers to a normed space, not the feasible domain. The authors are suggested to add a brief clarification to avoid potential confusion.
> > In an updated version of the manuscript, we will replace "domain'' with "ambient space.''
>
>
> > Regarding SDP problems, the quantum algorithms in this work output a dual solution under additive error, is it possible to derive error bounds for primal solutions?
>
> Under standard assumptions, strong duality holds, and so the output of our dual-only algorithm can be used to estimate the optimal objective value of the primal problem up to additive error $\varepsilon$. If one is interested in using this dual output to \textit{compute} an approximately optimal primal solution, the dual solution can be given as input for a simplex-spectraplex saddle point problem. Note however, that this still requires solving another optimization problem.
>
> > Selecting an appropriate mirror map is one of the crucial steps in implementing quantum mirror descent. Note that the authors provided several examples (e.g., for feasible domains like the probability simplex and Schatten-1 norm). For other feasible domains, are there general methodologies for choosing the mirror map and the Bregman divergence?
>
> The choice of mirror map is indeed crucial, as \textit{i)} the query complexity of the mirror descent algorithm depends polynomially on the strong convexity parameter of the mirror map; and \textit{ii)} we need to be able to efficiently compute the Bregman projection induced by the mirror map in order to update the solution in each iterate. Broadly, the mirror maps we discuss in the manuscript are the standard ones found in the literature, because \textit{i)} they are all 1- or 2-strongly convex with respect to their applicable domains and \textit{ii)} the Bregman projection associated to these maps admits a closed form expression. We also point out that optimization problems over more complicated problem domains can be reduced to the domains we consider. For example, one can reduce SDP problem whose feasible set is defined by the intersection of an affine space with the cone of positive semidefinite matrices, to mirror descent over the spectraplex through a well-known reformulation as a saddle point problem.
>
> > Regarding the result in Table 1, is there any classical or quantum lower bound?
>
> Thank you for the question. Indeed, classical lower bound exists: for the first-order methods (last column of Table 1), when the objective function is $L$-smooth, one can have quadratic improvements via Nesterov's accelerated gradient methods [Nes83]. Intuitively, these "accelerated" methods use not only the gradient information in the current iterate, but also the past information. In short, the classical first-order methods constitue the update of the form $x_{t+1} \in x_0 + \text{span}\{ \nabla f(x_0), \dots, \nabla f(x_t) \}$.
>
> However, we want to emphasize that the main message we wanted to convey in this work is that zeroth-order quantum gradient methods can match the classical first-order query complexity of gradient descent-type of algorithms. That is, nothing prevents us from analyzing zeroth-order quantum accelerated gradient methods in the similar manner that we presented our current work. As we replied to Reviewer fZUu, this constitutes a natural future direction.\\
> Quantumly, there exists lower bounds that illustrate there is no quantum advantage with first or higher order oracle [GKNS20, GKNS21]---this is exactly why we considered the zeroth-order setting, and showed exponential separation in terms of the dimension.
>
> Thank you again for your time and efforts in reviewing our paper. We hope the above clarification addresses any outstanding questions or concerns. Please do not hesitate to ask follow up questions during the discussion period.
>
> [GKNS20] A. Garg, R. Kothari, P. Netrapalli, \& S. Sherif, (2021). No quantum speedup over gradient descent for non-smooth convex optimization. arXiv preprint arXiv:2010.01801.
>
> [GKNS21] A. Garg, R. Kothari, P. Netrapalli, \& S. Sherif, (2021). Near-optimal lower bounds for convex optimization for all orders of smoothness. \textit{Advances in Neural Information Processing Systems}, 34, 29874-29884.}
>
> [Nes83] Y. E. Nesterov, 1983 "A method for solving the convex programming problem with convergence rate $O(1/k^2)$"

---

> > ### Comment · Reviewer_7YDD · 2025-08-03
> >
> > OK, no more questions from me. Thanks for the thorough responses and good luck!

---

> > > ### Author Response · Authors · 2025-08-06
> > >
> > > Dear Reviewer 7YDD,
> > >
> > > We appreciate your feedback and engagement throughout the discussion.
> > > If all concerns and questions are resolved, we kindly ask you to reflect it in the score.  Otherwise, please don’t hesitate to follow up with more questions.
> > > Thank you again for taking the time to carefully review our paper and provide constructive feedback.

---

### Official Review · Reviewer_eitt · 2025-06-28

**Clarity:** 3
**Significance:** 2
**Originality:** 2
**Rating:** 4
**Confidence:** 2

**Summary:**

This paper shows that quantum algorithms can solve zeroth-order convex optimization problems, using only noisy function evaluations, with query complexity matching that of classical first-order methods. By combining quantum gradient estimation with standard optimization routines like projected subgradient and mirror descent, the authors achieve exponential improvements in dimension over classical zeroth-order methods. They extend the framework to smooth functions, non-Euclidean settings, and structured problems like SDPs, LPs, and zero-sum games, demonstrating improved complexities in several regimes.

**Questions:**

In view of the "Weaknesses" section:

* Are there genuinely new technical challenges in adapting classical convergence guarantees to the quantum zeroth-order setting, beyond tracking bias and variance from oracle errors?
* Does any part of the analysis fundamentally rely on quantum-specific insights, or could the results be derived by combining known quantum gradient estimation techniques with standard optimization proofs?

It would be important for the authors to clarify this, as it directly impacts the novelty and technical contribution of the work..

**Ethical Concerns:**

["NO or VERY MINOR ethics concerns only"]

**Final Justification:**

In response to the reviewer comments, the authors have clarified their contribution.

**Limitations:**

No limitations

**Paper Formatting Concerns:**

No concerns

**Quality:**

3

**Strengths And Weaknesses:**

* Strengths:
    * The key positive result is that zeroth-order quantum methods match the rates of classical first-order methods, exhibiting an exponential separation from classical zeroth-order optimization.
    * The paper offers a unified and complete treatment across various settings—nonsmooth and smooth objectives, Euclidean and non-Euclidean geometries—using tools like mirror descent, dual averaging, and mirror prox.
    * The framework is broadly applicable and extends to important structured problems such as semidefinite programming, linear programming, and zero-sum games.
    * The paper is well written. It is mathematically precise, and the results are well contextualized within the broader literature on quantum optimization.
* Weakness
    * The core technical contribution seems limited to plugging known quantum gradient (or subgradient) estimators into standard first-order convex optimization routines. Once an oracle with sufficiently controlled bias and variance is assumed (or constructed), the convergence analyses for mirror descent, projected subgradient, or gradient descent follow in a largely black-box, modular fashion. The analysis mirrors classical proofs, accounting for estimation error.

---

> ### Author Rebuttal · Authors · 2025-07-31
>
> Thank you for your constructive comments and feedback. We are glad that you find our work to exhibit key positive results in a unified and complete manner across various settings. Below, we address your concerns in detail.
>
>
> > Are there genuinely new technical challenges in adapting classical convergence guarantees to the quantum zeroth-order setting, beyond tracking bias and variance from oracle errors?
>
> Thank you for the question. While the framework of our analysis appears to follow classical convergence analysis closely, there are some subtle but fundamental challenges posed by the guarantees of quantum gradient estimators. The primary difference from typical classical gradient estimators is that the quantum gradient estimators exhibit non-zero bias, resulting from the phase estimation, a subroutine used in the original quantum gradient estimation [Jor05]. Most classical convergence proofs are specific to unbiased estimators and thus extending them to handle bias is a non-trivial challenge. These extensions are more technically complicated in the case of arbitrary normed spaces. In fact, the gradient estimation algorithms of [Jor05, GAW19] would not in fact be sufficient for most of our results and we must use instead a modified version of the improved gradient estimation from [vACGN23], which utilizes suppressed-bias phase estimation, such that we can also control the variance (c.f., Corollary A.1 and Theorem A.2). As an illustration of this challenge, a recent paper [CSW25] (Theorem 47) considers the optimization of a smooth and strongly convex function with condition number $\kappa$, and achieves an oracle complexity of $(1/\varepsilon)^\kappa$, exactly due to the complication of existing bias in quantum gradient estimation.
>
> The quantum subgradient methods pose similar challenges, with the estimate obeying the subgradient guarantee at a point that is shifted from the input point, again with a biased error term.
>
> We note that quantum gradient estimation algorithms have been studied for nearly two decades at this point, and quantum algorithms for zeroth-order convex optimization (in the regime $d \ll 1/\epsilon$) are also nearly 5 years old. However, despite the naturalness of the question, an exponential speedup in $d$ for the regime $1/\epsilon \ll d$ has not been demonstrated. We believe this gap in the literature, which is first addressed by this work, illustrates the non-triviality of the technical challenge.
>
> > Does any part of the analysis fundamentally rely on quantum-specific insights, or could the results be derived by combining known quantum gradient estimation techniques with standard optimization proofs?
>
>
>
> Thank you for the question. If by ``quantum-specific insights", you mean an entirely new quantum subroutine that directly uses low level tools such as phase estimation or singular value transforms, then the answer is no. We use a minor modification of suppressed bias gradient estimation from [vACGN23] and the quantum subgradient methods from [CCLW20,vAGGdW20] as our building blocks. However, as discussed in the section above, the optimization proofs are not standard since care needs to be taken to incorporate the quantum estimators into the analysis.
>
> We note that this style of analysis, where known quantum subroutines must be carefully integrated into classical frameworks, is common in the field of quantum machine learning, including several recent papers at NeurIPS/ICML.
>
> Finally, we note that our whitebox results present a new paradigm for quantum algorithms for SDPs/LPs whose performance dominates all known algorithms in some parameter regimes. The fundamental building block for these results are the gradient methods in normed spaces, which is constructed for the first time in this work.
>
>
>
>
> We thank the reviewer for pushing us to be more precise about the technical significance of our contribution, and we hope this clarification addresses the concern raised. If so, we kindly request the reviewer to reflect it in the score. Thank you again for your time and efforts in reviewing our paper.
>
> [Jor05] S. P. Jordan, 2005 "Fast quantum algorithm for numerical gradient estimation."
>
> [GAW19] A. Gilyén, S. Arunachalam, N. Wiebe, 2019 "Optimizing quantum optimization algorithms via faster quantum gradient computation."
>
> [vACGN23] J. van Apeldoorn, A. Cornelissen, A. Gilyén, G. Nannicini, 2023 "Quantum tomography using state-preparation unitaries."
>
> [CSW25] A. B. Catli, S. Simon, N. Wiebe, 2025 "Exponentially better bounds for quantum optimization via dynamical simulation."
>
> [CCLW20] S. Chakrabarti, A. M. Childs, T. Li, X. Wu, 2020 "Quantum algorithms and lower bounds for convex optimization."
>
> [vAGdW20] J. van Apeldoorn, A. Gilyén, S. Gribling, R. de Wolf, 2020 "Convex optimization using quantum oracles."

---

> > ### Comment · Reviewer_eitt · 2025-08-05
> >
> > I would like to thank the authors for their detailed response. I have updated my score accordingly.

---

> > > ### Author Response · Authors · 2025-08-06
> > >
> > > Dear Reviewer eitt,
> > >
> > > We appreciate the active participation during the discussion period!
> > > Thank you again for taking the time to carefully review our paper and provide constructive feedback.

---

### Official Review · Reviewer_fZUu · 2025-07-02

**Clarity:** 3
**Significance:** 2
**Originality:** 3
**Rating:** 4
**Confidence:** 3

**Summary:**

This paper presents new quantum algorithms for zeroth-order convex optimization—that is, optimization using only noisy function value queries, without access to gradients. The main contributions are:

Exponential Quantum Speedup: The authors show that quantum algorithms can match the first-order query complexities of classical gradient-based methods, even though they only use zeroth-order (function value) oracles. This results in an exponential speedup in the dimension d compared to classical zeroth-order methods, whose complexity is typically linear in d

Dimension-Independent Query Complexity: The proposed quantum algorithms achieve dimension-independent query complexities (up to polylogarithmic factors) for both smooth and nonsmooth convex optimization problems, a significant improvement over classical zeroth-order approaches.

Generalization to Non-Euclidean Settings: By leveraging quantum (sub)gradient estimation, the authors extend their methods to non-Euclidean spaces using quantum mirror descent and its variants (including dual averaging and mirror prox), broadening the applicability of their techniques.

Applications to White-box Problems: The framework is applied to important structured optimization problems such as semidefinite programming (SDP), linear programming (LP), and zero-sum games. For these, the quantum algorithms achieve better or comparable complexities to the best known classical and quantum methods in certain parameter regimes.

Technical Advances: The paper develops robust quantum subgradient and gradient estimation procedures that work with noisy oracles, and provides convergence guarantees for quantum versions of projected subgradient, mirror descent, and mirror prox methods

Applications and Speedups:

SDP and LP: The quantum mirror descent framework yields SDP and LP solvers with gate complexities that provide quadratic speedups in certain regimes over classical matrix multiplicative weights methods and some previous quantum approaches.

Zero-sum Games: The algorithm achieves improved complexity for finding Nash equilibria in zero-sum games, outperforming prior classical and quantum algorithms in some parameter ranges

**Questions:**

Stated above.

**Ethical Concerns:**

["NO or VERY MINOR ethics concerns only"]

**Limitations:**

None.

**Paper Formatting Concerns:**

None.

**Quality:**

3

**Strengths And Weaknesses:**

Overall, i think this is a nice paper  making a good contribution to Quantum optimization.

The results are fairly theoretical (which i think is OK) and assume access to sufficiently precise quantum oracles; practical implementation awaits advances in quantum hardware. I guess it'd be nice if the authors discuss what happens if the oracles are a bit noisy.

The quantum algorithms match the rates of standard (non-accelerated) classical first-order methods, but do not (yet) achieve the accelerated rates of Nesterov-type methods. Again, it'd be good to have a discussion on this.

The precision requirements for the oracles are stringent; relaxing these is an open direction for future research

---

> ### Author Rebuttal · Authors · 2025-07-31
>
> Dear Reviewer fZUu,
>
> Thank you for your constructive comments and feedback. We are glad that you find our work to be a good contribution to the field of quantum optimization. Below, we address your primary concerns and questions in detail.
>
> > What happens if the oracles are a bit noisy?
>
> We already consider noisy oracles, as shown in Definition 1 and the noise tolerance of each algorithm is reflected in the corresponding theorem. If you mean what happens if oracles are noisier than what our theorems require, then in general the final sub-optimality guarantee will not hold.
>
> We note that this is also the situation for classical algorithms, and each gradient method has a maximum noise tolerance (which is at most $\max\left(\frac{\epsilon^2}{d},\frac{\epsilon}{d}\right)$ for any algorithm with polynomially many queries [RL16]. We also note that the classical algorithms with the highest robustness are generally slower than gradient methods. The situation is the same for quantum algorithms, and while we provide noise thresholds for all our algorithms for reasons of concreteness, developing the most robust algorithm is a somewhat orthogonal endeavor.
>
> Finally, we emphasize that in the classical setting, even with infinitely-accurate (i.e., noiseless) zeroth-order oracles, the query complexity remains linear in $d$ [JNR12], and therefore our methods listed in Table 1 exhibit an exponential query speedup in the whole noise regime for which the guarantees hold.
>
> > Is Nesterov-type acceleration possible?
>
> Thank you for the thoughtful comment. Indeed, Nesterov-type acceleration should be possible. The complication, though, is the well-known error accumulation on using erroneous oracle, for the accelerated methods [DGN14]. As a result, we expect the noise tolerance $\theta$ might be more stringent than the non-accelerated results we provided in the current work. Looking into the accelerated scenario definitely is an interesting future step.
>
> > Can you relax the precision requirements?
>
> The main message we wanted to convey in this work is that zeroth-order quantum gradient methods can match the classical first-order query complexity of gradient descent-type of algorithms. That being said, it is possible that the precision requirement can be relaxed to a certain degree, e.g., improving polynomial dependence on the $\varepsilon$ and $d$ for $\theta$ required in Theorem 2.1, for instance. This may necessitate further development in the underlying quantum gradient and subgradient computation subroutines.
>
>
> Thank you again for your time and efforts in reviewing our paper. We hope the above clarification addresses any outstanding questions or concerns. Please do not hesitate to ask follow up questions during the discussion period.
>
> [JNR12] K. G. Jamieson, R. D. Nowak, B. Recht, 2012 "Query Complexity of Derivative-Free Optimization".
>
> [RL16] A. Risteski, Y. Li, 2016 "Algorithms and matching lower bounds for approximately-convex optimization."
>
> [DGN14] O. Devolder, Y. Nesterov, 2014 "First-order methods of smooth convex optimization with inexact oracle."

---

> > ### Author Response · Authors · 2025-08-06
> >
> > Dear Reviewer fZUu,
> >
> > We thank you again for your constructive review.
> >
> > Given that the discussion period is ending soon, we kindly ask you to review our replies; if there are remaining concerns, please don’t hesitate to follow up with more questions.
> >
> > Thank you again for taking the time to carefully review our paper and provide constructive feedback.

---

### Official Review · Reviewer_Xxvf · 2025-07-04

**Clarity:** 3
**Significance:** 2
**Originality:** 2
**Rating:** 2
**Confidence:** 4

**Summary:**

This paper presents a unified algorithmic framework for zeroth-order convex optimization with certain quantum oracles to achieve dimension-independent optimization errors. By assuming the errors in the function values of the quantum oracle to be vanishingly small, the proposed achieved rate is comparable to the errors achievable under the classical first-order-oracle setting under stochastic unit-variance errors. The key technical idea in the proposed approach is to leverage a gradient estimator analyzed in [74] (presented as in Lemma 2.1) to adapt first order algorithms to the quantum zeroth order setting. This approach is extended beyond zeroth order convex and smooth optimization, and achievability analysis was provided form SDP, LP, and zero-sum games.

**Questions:**

While the topic investigated in the manuscript is clearly of significance and there might be technical merits of the proposed results. As mentioned in the earlier discussion on the weakness of the submitted draft, we recommend the authors to provide following discussions for the manuscript to be considered publishable.

1. The achieved rates of the quantum algorithm must be compared to the ones achieved by the best known classical algorithms under the same vanishing error condition, i.e., the noise variance decays with the same polynomial dependency of epsilon and d.

2. The comparison between classical and quantum algorithms needs to be based on the above fair comparison, and any claims in the abstract needs to be justified in the manuscript.

3. We recommend the authors to clearly mention in the abstract that the quantum oracle considered in this work under the pure state assumption is known to be strictly easier than its classical counterpart. There has been abundant works in recent years that demonstrated this distinction, which should also be mentioned.

4. Section 2 mentioned that one main benefit of first-order methods vs newton's approach is the convergence of global optimum irrespective of the starting point (line 103-107 on page 3), however we would point out that this global convergence is achievable with a modified newton's method with second-order smoothness condition.

**Ethical Concerns:**

["NO or VERY MINOR ethics concerns only"]

**Final Justification:**

After the discussion period, our main concerns remain, as elaborated in our responses to the authors. Especially, while we appreciate the authors' contribution, we recommend an accurate presentation on the contribution and scope of the paper, especially on the quantum vs classical comparison.

**Limitations:**

yes

**Quality:**

2

**Strengths And Weaknesses:**

Strengths:  Zeroth-order optimization is important in settings where gradients are unavailable or expensive to compute, and understanding potential speedup of quantum algorithms is an important topic that could have significant impact in Neurips society. Any significant result in this direction on improving the best known results and approaches could provide valuable theoretical insights. In terms of clarity, the technical sessions of this paper are well-written and accessible. The assumptions are clearly stated and proved, and the analysis is carefully structured.

 Weaknesses: A main issue of the submitted manuscript is the mismatch between the noise amplitude between the proposed quantum framework and the classical oracles, which results in an unfair advantage of the proposed scheme on achieving an optimization error that could have been achieved and outperformed by the classical oracles under the same noise condition, even under zeroth order access.

For instance, definition 1 assumes that the quantum oracle has a zeroth order measurement error of theta which is conventionally presumed to be constant. But through a careful investigation, the achievability results presented in this work in fact requires the measurement error theta to be polynomially smaller than the targeting estimation error epsilon (e.g., see the 2.1). Knowing that under such vanishing error condition, the classical zeroth order oracle can in fact achieve rates that are orderwise better than the rates in e.g. Table 1 that was used in the comparison. Particularly, the bound on the gradient estimation error this manuscript relies on (Lemma 2.1) is weaker than the guarantees of the classical zeroth order two point gradient estimator under the same regime of theta used in the main theorems. Therefore, if we apply the same analysis under the same error conditions, the rates achievable for classical algorithms would outperform the rates of quantum algorithms presented in this work, which is the opposite of what is claimed in the abstract.

Two other important but relatively minor issues: 1. the abstract claims an exponential separation between quantum and classical optimization but this is not clearly elaborated in the main text. 2. the quantum oracle considered in this work assumes a pure state access which is well known to be algorithmically easier than its classical counterpart. Therefore, it is not a generalization of the classical access. However, this distinction is not made clear in the abstract, which could potentially mislead the readers into thinking the proposed result is a quantum speedup.

---

> ### Author Rebuttal · Authors · 2025-07-31
>
> Dear reviewer Xxvf,
>
> Thank you for your detailed comments and feedback. We appreciate your positive comments regarding the significance of the results, and the clarity of the presentation. You raise important technical points in your review, however, we feel that there is some misunderstanding regarding the most important ones. Below, we address each of your questions and concerns in detail.
>
> > Definition 1 assumes that the quantum oracle has a zeroth order measurement error of theta which is conventionally presumed to be constant. But through a careful investigation, the achievability results presented in this work in fact requires the measurement error theta to be polynomially smaller than the targeting estimation error epsilon (e.g., see Thm 2.1). Knowing that under such vanishing error condition, the classical zeroth order oracle can in fact achieve rates that are orderwise better than the rates in e.g. Table 1 that was used in the comparison.
>
> Thank you for this thoughtful comment. It is indeed important to compare the classical and quantum settings under similar noise assumptions. However, as we clarify below, the speedup demonstrated in our paper persists under our noise assumptions.
> - Firstly, we note that the noise $\theta$ in the oracle cannot be chosen to be a constant independent of $d,\epsilon$ even in the classical case, since if the noise is asymptotically polynomially larger than $\max\left(\frac{\epsilon^2}{d},\frac{\epsilon}{d}\right)$, no classical algorithm can output an $\epsilon$-approximate minimum using $\mathrm{poly}(d,\epsilon)$ queries (see Theorem~3.1 in [RL16]). Therefore any polynomial query classical algorithms also tolerate noise that is falling polynomially with the target error and the dimension.
> - Furthermore, the last column in Table 1, we state the classical **first-order** query complexity, not zeroth-order. If we instead choose to write classical zeroth-order, every row in the classical column is multiplied by $d$; importantly, this holds even when the function evaluation oracle has perfect precision (i.e., completely noiseless) [JNR12]. Thus the quantum algorithm exhibits an exponential speedup in terms of $d$ **even if the classical oracles are completely noiseless**, and the following statement is not accurate.
> > Knowing that under such vanishing error condition, the classical zeroth order oracle can in fact achieve rates that are orderwise better than the rates in e.g. Table 1 that was used in the comparison.
>
> Finally, we are aware that the classical methods can be accelerated by using Nesterov's accelerated gradient methods, for instance in the smooth setting. However, this is an orthogonal discussion to our claim of exponential speedup in terms of the dimension $d$. In principle, nothing prevents us from considering accelerated zeroth-order quantum gradient methods. Indeed, as we reflected in our response to reviewer fZUu02, analyzing Nesterov's accelerated gradient method within our framework constitutes a natural future direction.
>
> > Particularly, the bound on the gradient estimation error this manuscript relies on (Lemma 2.1) is weaker than the guarantees of the classical zeroth order two point gradient estimator under the same regime of theta used in the main theorems. Therefore, if we apply the same analysis under the same error conditions, the rates achievable for classical algorithms would outperform the rates of quantum algorithms presented in this work, which is the opposite of what is claimed in the abstract.
>
> Thank you for this thoughtful comment. By the "classical two point gradient estimator," we believe you are referring to the work by [DJW15] (otherwise, please let us know which reference you have in mind). Their **lower bound** result clearly exhibits dimension dependence: at least $\Omega(d/\varepsilon^2)$ observations are necessary using two function evaluations, and at least $\Omega(d^2/\varepsilon^2)$ using a single function evaluation. Note that these results are in the **noiseless** function evaluation setting.
> To sum up, we are unsure how to interpret your comment "classical algorithms would outperform the rates of quantum algorithms presented in this work." We note that classically, even for (strongly convex) quadratics, a tight bound of $\Theta (d^2/ T)$ exists [Sha13]. Based on these observations, the claim of speedup in the abstract is indeed accurate.
>
> > The abstract claims an exponential separation between quantum and classical optimization but this is not clearly elaborated in the main text.
>
> We appreciate the opportunity to clarify this point. As we mention in the main text (e.g., line 74), we achieve exponential separations in terms of the dimension $d$ for convex optimization using zeroth-order oracle. This is evident in Table 1, where we state the classical first-order query complexity to highlight that they match our results on zeroth-order quantum gradient methods.
> In future versions, we will make an effort to highlight the separation even more clearly.
> We can change the last column of Table 1 to classical (0th order) query complexity, in which case every row is multiplied by the dimension $d$, to present the exponential separation in a more explicit way.
>
> > The quantum oracle considered in this work assumes a pure state access which is well known to be algorithmically easier than its classical counterpart. Therefore, it is not a generalization of the classical access. However, this distinction is not made clear in the abstract, which could potentially mislead the readers into thinking the proposed result is a quantum speedup.
>
> We appreciate this comment and agree that it is very important to be careful about the nature of quantum oracle access that is assumed. However, we note that the oracle access assumed in our manuscript does not return the function value encoded in the amplitudes of a pure state (which is indeed incomparable to a standard oracle), but rather simply allows the classical oracle to be queried in superposition. Given a classical circuit implementing the classical oracle, we can construct a quantum circuit for our oracle by replacing each gate with a reversible equivalent using Toffoli gates. Using such a construction, the gate complexity of the quantum oracle is within a constant factor of that of the classical oracle. A speedup that assumes this sort of oracle access is the standard assumption for almost all quantum speedups in query complexity, including well-known algorithms such as Simon's algorithm [Sim97], and most previous works on quantum speedups for convex optimization [CCLW20, vAGdW20, GKNS21, GAW19, Jor05].
> Therefore, the algorithms presented in this work **do in fact constitute a quantum query speedup in the usual sense**.
>
>
> > Section 2 mentioned that one main benefit of first-order methods vs newton's approach is the convergence of global optimum irrespective of the starting point (line 103-107 on page 3), however we would point out that this global convergence is achievable with a modified newton's method with second-order smoothness condition.
>
> Thank you for the suggestion. We will reflect this in the text in future versions.
>
> We thank the reviewer for very specific technical comments and giving us the opportunity to clarify some important points. Please do not hesitate to ask follow up questions during the discussion period, and if your concerns are adequately addressed, we kindly request you to reflect it in the score. Thank you again for your time and effort in reviewing our paper.
>
> [JNR12] K. G. Jamieson, R. D. Nowak, B. Recht, 2012 "Query Complexity of Derivative-Free Optimization".
>
> [DJW15] J. C. Duchi, M. I. Jordan, M. J. Wainwright, A. Wibisono, 2015 "Optimal rates for zero-order convex optimization: the power of two function evaluations".
>
> [Sha13] O. Shamir, 2013 "On the Complexity of Bandit and Derivative-Free Stochastic Convex Optimization".
>
> [RL16] A. Risteski, Y. Li, 2016 "Algorithms and matching lower bounds for approximately-convex optimization."
>
> [CCLW20] S. Chakrabarti, A. M. Childs, T. Li, X. Wu, 2020 "Quantum algorithms and lower bounds for convex optimization."
>
> [vAGdW20] J. van Apeldoorn, A. Gilyén, S. Gribling, R. de Wolf, 2020 "Convex optimization using quantum oracles."
>
> [GKNS21] A. Garg, R. Kothari, P. Netrapalli, S. Sherif, 2021 "No Quantum Speedup over Gradient Descent for Non-Smooth Convex Optimization."
>
> [GAW19] A. Gilyén, S. Arunachalam, N. Wiebe, 2019 "Optimizing quantum optimization algorithms via faster quantum gradient computation."
>
> [Jor05] S. P. Jordan, 2005 "Fast quantum algorithm for numerical gradient estimation."
>
> [Sim97] D. R. Simon, 1997 "On the Power of Quantum Computation."

---

> > ### Comment · Reviewer_Xxvf · 2025-08-06
> >
> > The authors made a clarification that partially addressed some of my earlier concerns. Specifically, the classical rates in Table 1 correspond to fixed algorithms such as gradient descent, rather than fundamental limits. This distinction explains why even the rates are stated under first-order, and possibly noiseless settings (as suggested by the authors), the reported rate remains exponentially worse compared to known results for zeroth-order methods with vanishing error. Since the authors expressed doubt regarding the accuracy of this point, we would like to clarify that our claim of “Knowing that under such vanishing error condition, the classical zeroth order oracle can in fact achieve rates that are orderwise better than the rates in e.g. Table 1 that was used in the comparison.” was precisely on the comparison to best known algorithms, which achieve rate improvements from $1/\varepsilon$ to $\log(1/\varepsilon)$
> >
> > However, our concern remains, and we would like the authors to clarify whether the proposed quantum algorithm or the general approach achieves exponential speedup over the best classical algorithms (under vanishing errors), not just over PSG and GD? Since the proposed method already deviates from exact GD through accessing a quantum oracle, restricting comparison only to GD is insufficient unless the improvement holds universally for all classical methods, in which case, the scope of comparison should be stated explicitly.
> > While we are still reviewing the rest of the authors’ responses, to ensure timely feedback, we also ask the authors to briefly clarify the following key technical point:
> > The authors cite Line 74 and Theorem 2.1 as demonstrating exponential improvement. However, given that the best-known classical rate the reviewer has in mind is better ($\log(1/\varepsilon)$) than what is stated in the theorem. Please specify which classical rate the quantum result is compared against, and confirm whether that classical rate is tight for the corresponding algorithm (e.g., PSG). Without this, the claim of exponential speedup remains ambiguous. Especially, note that the stated rate achieves at most a polynomial speed up compared to well-known results in stochastic optimization that are achievable even with unit variance noises. E.g., see the following paper.
> >
> > Tor Lattimore and Andras Gyorgy. “Improved regret for zeroth-order stochastic convex bandits”. CoLT, 2021.
> >
> > Finally, as a clarification for the authors about constant-bounded-error assumptions in zeroth-order optimization, the model we referred to is the standard stochastic derivative-free optimization framework, where the additive noise is random rather than fixed. In such cases, classical algorithms average multiple noisy queries per iteration to reduce the effective variance over time (e.g., see [Sha13] in the authors’ response and the reference cited above). While this paper considers a stronger fixed-noise model that we were aware of in the initial review, our stated question was not to consider the regime of increased variances, but the impact of sample reduction due to reduced variances. Based on the authors’ response, it appears that we can agree that an appropriate benchmark is the rates for the zeroth algorithms in the zero-noise setting.

---

> > > ### Author Response · Authors · 2025-08-06
> > >
> > > We thank the reviewer for their comments. Unfortunately, we feel that there remain some misunderstandings, that we address below in response to the reviewer’s primary question.
> > >
> > > > However, our concern remains, and we would like the authors to clarify whether the proposed quantum algorithm or the general approach achieves exponential speedup over the best classical algorithms (under vanishing errors), not just over PSG and GD?
> > >
> > > **Yes**, the exponential speedup is over all classical algorithms and **not** only against PSG and GD. Note that our claimed speedups are in terms of the **dimension $d$** and *not* the output error $1/\varepsilon$. We focus on speedups in $d$ rather than $1/\varepsilon$ because we are interested in the high-dimensional, low-precision regime where $d \gg 1/\varepsilon$ (mentioned on Page 3, Line 99).
> > >
> > > > ... on the comparison to best known algorithms, which achieve rate improvements from $1/\varepsilon$ to $\log(1/\varepsilon)$.
> > >
> > > For classical Lipschitz convex optimization, even in the **first-order** oracle setting, the query complexity is $\Theta(\min \\{ 1/\varepsilon^2, d \log(1/\varepsilon) \\})$, where the former rate is achieved by PSG [1], and the latter rate is achieved by cutting plane method [2]. In short, if you want to have $\log(1/\varepsilon)$ dependence on the error $\varepsilon$, you **must pay $d$ even with the first-order oracle, classically**. Naturally, in high-dimensional, low-precision regime, (sub)gradient-based methods dominate.
> > >
> > > No classical zeroth-order algorithm can **possibly** achieve a polylogarithmic dependence on $d$, as indicated by the classical lower bounds [3, 4, 5, 6, 7].
> > >
> > > We hope this clarifies the main concern. Please do not hesitate to ask follow up questions, if there are more.
> > >
> > > [1] A. Nemirovski, D. Yudin, 1983 "Problem Complexity and Method Efficiency in Optimization"
> > >
> > > [2] L. Khachiyan, S. Tarasov, I. Erlikh, 1988 "The method of inscribed ellipsoids"
> > >
> > > [3] A. Jambulapati, A. Sidford, K. Tian, 2024 "Closing the Computational-Query Depth Gap in Parallel Stochastic Convex Optimization"
> > >
> > > [4] J. Duchi, M. Jordan, M. Wainwright, and A. Wibisono, 2015 "Optimal rates for zero-order convex optimization: the power of two function evaluations"
> > >
> > > [5] M. Braverman, E. Hazan, M. Simchowitz, B. Woodworth, 2021 "The gradient complexity of linear regression"
> > >
> > > [6] O. Shamir, 2013 "On the Complexity of Bandit and Derivative-Free Stochastic Convex Optimization"
> > >
> > > [7] J. C. Duchi, M. I. Jordan, M. J. Wainwright, A. Wibisono, 2015 "Optimal rates for zero-order convex optimization: the power of two function evaluations"

---

> > > > ### Comment · Area_Chair_Kt8i · 2025-08-09
> > > >
> > > > Dear Reviewer Xxvf,
> > > >
> > > > Given that our author-reviewer discussion period is coming to an end, please take a close look at the further reply by authors, in particular if you agree with their clarification on results. Thanks.
> > > >
> > > > Best wishes,
> > > > AC

---

> > > > > ### Comment · Reviewer_Xxvf · 2025-08-09
> > > > >
> > > > > I have read both responses by the authors. Some of my earlier concerns remain, and some are partially resolved.
> > > > >
> > > > > Our first main concern, which was fully investigated in the discussion period, is about the appropriate comparison between classical and quantum complexities. Especially, the earlier review claimed that one can achieve better rates with classical algorithms than those presented in Table 1. While the authors framed this as a misunderstanding, this gap is justified by an inline assumption stated on page 3, which appears after the introduction and where the table is introduced. Therefore, our first suggestion on the manuscript is to follow the guidelines in the checklist to ensure the abstract and introduction accurately reflect the paper’s contributions and scope. If any claims rely on such non-standard assumptions, it has to be stated in advance.
> > > > >
> > > > > Further, while the authors justify improvement on sample complexity, we have agreed that the right point of comparison is the zero-error complexities for zero-th order oracles. The lower bounds the authors cited in the responses are still for the unit-variance additive noise cases, which are exactly what we agreed to be the unfair comparisons and do not apply to our discussion. Specifically, while we do not have time to read through all details in the entire list of references, the ones that are highlighted in the first response ([DJW15] [Sha13], or [4][6] in the second response) are both for cases with non-vanishing noises, therefore, the cited rates does not apply. As a simpler check, any complexity bound on the gradient estimator with both d and epsilon involved is not applicable for the zero-error case (if they are stated appropriately).
> > > > >
> > > > > Importantly, through the discussion, we realized the main technical block that leads to all achievability results in this paper is through the suppressed-bias gradient estimation in [vACGN23], so that with the stronger quantum oracle and with vanishing error, one can essentially simulate a classical first-order oracle with a zeroth-order quantum oracle by paying log factors in complexities. If this is true, which seems to be confirmed by the authors, the clarity of the main messages in this paper can be clearly improved by merging the achieved quantum complexity and first-oracle complexities (e.g., in Table 1), and instead providing the lower bound for the zeroth-order oracles in the zero-error case as suggested.
> > > > >
> > > > > Finally, while the pure-state quantum oracle considered in this paper is not a generalization of the classical ones, the proposed rates do not constitute a quantum speedup. The authors tried to defend against this point by considering the case where we have full classical circuit access and can use it to build the quantum oracle. However, in such a case, we would have full information on the objective function; hence, the sample complexity is zero; therefore, there would be no quantum speedup.

---

### Decision · Program_Chairs · 2025-09-17

**Decision:**

Accept (poster)

**Comment:**

This paper studied quantum algorithms for zeroth-order convex optimization. There are two main parameters: dimension d and precision eps, and previous quantum algorithm literature on this topic mainly studied the dependence in terms of d. This paper carefully studied quantum algorithms for zeroth-order convex optimization in terms of eps dependence. In particular, this paper proved that for various convex optimization algorithms, including Gradient Descent (GD), Projected Subgradient Method (PSG), Mirror Descent (MD), Dual Averaging (DA), and Mirror Prox (MP), the quantum algorithms achieve a similar convergence rate compared to that of classical first-order algorithms. If compared to classical zeroth-order algorithms with a linear factor in d, the proposed quantum algorithms achieve an exponential speedup in d. Technically, the results rely on a “suppressed-bias” version of Jordan's quantum gradient estimation algorithm + detailed analysis of corresponding classical algorithms. As applications, the proposed quantum algorithm provided new bounds for solving SDPs, LPs, and zero-sum games.

Among the reviews, there are three with positive scores (7YDD, eitt, fZUu) and one with negative score (Xxvf). There were adequate discussions between Xxvf and the authors with a few interactive rounds, but disagreements still exist. Given this, the AC carefully checked the manuscript as well as the arguments from both Xxvf and authors. The perspective from the AC is as follows:

- Xxvf is confused about the setting of the noise: It is noted that the paper's claim are all based on **adversarial** noises, i.e., a function value can be an arbitrary one within a certain error compared to its true evaluation (Definition 1). This model is more restricted than having stochastic noises. In this sense, all the proved bounds are stronger than having a precise evaluation oracle and hence present robustness.

- Xxvf is also confused about the oracle setting: Both classical and oracle here are set for black-box functions. Such black boxes are implemented by arithmetic operations - classically those are additions, multiplications, etc., and quantumly those are the same arithmetic operations in superposition (quantum adders, multipliers, etc.). There is no sampling in constructing the oracles.

- However, the authors did not make correct citations to classical lower bounds. The AC agrees with Xxvf's comment that the mentioned classical lower bound papers [JNR12], [DJW15], [Sha13] all considered **stochastic** convex optimization with $\Omega(1)$ variance, and it is to be emphasized that lower bounds on stochastic convex optimization and non-stochastic convex optimization are not the same problem. Nevertheless, I think [WS17] below is a correct bound to refer to as it proved a $\Omega(L^2 B^2/eps^2)$ classical first-order lower bound (L being the Lipschitz parameter and B being an upper bound on x's norm) that allows each query to give any subgradient and applies to any randomized algorithm. It would be even better to find a $\Omega(d \cdot L^2 B^2/eps^2)$ classical zeroth-order lower bound, but the AC cannot easily find such a result - would be great for authors to check more into corresponding literature.

In all, given that

- This paper studied quantum algorithms for zeroth-order convex optimization in terms of eps dependence, which an important setting, with further applications into widespread optimization problems such as SDPs, LPs, and zero-sum games, and

- Three reviewers voted for positive scores, and the review with negative score got confused with points clarified above,

the decision to accept this paper at NeurIPS 2025.

Nevertheless, the authors should make the following changes in the final version of the paper:

- Make correct citations to classical lower bounds (see the point above), especially the [WS17] paper. It would be helpful if the authors can find a paper with classical $\Omega(d)$ lower bound using zeroth-order oracle, or discuss about why it is believed that this holds for classical **non-stochastic** convex optimization.

- All promised changes in rebuttal.

- This was not raised by reviewers, but when the AC checked the paper, it is observed that the authors should expand "Limitations and future work" at the end of the paper about the applications studied in this paper, from the following two perspectives:

- - In rebuttal, the authors mentioned that this paper considered the high-dimensional, low-precision regime. However, it is noted that
that in such cases, especially when the precision is as low as a constant, the bound for SDP may be worse than the earlier quantum work [72], and the bound for LP may be worse than the earlier quantum works [11,29]. I suggest the authors to further discuss the limitations of their results compared to previous quantum algorithm works, in particular highlight more on which domains the proposed quantum algorithms are better.

- - Given that the applications to LP, SDP, and zero-sum games all have polynomial dependence in 1/eps, I suggest the authors to comment more on whether Nesterov-type acceleration is possible for these applications in particular given that the noise tolerance might be more stringent than the non-accelerated results.

**References**

[JNR12] K. G. Jamieson, R. D. Nowak, B. Recht, 2012 "Query Complexity of Derivative-Free Optimization".

[DJW15] J. C. Duchi, M. I. Jordan, M. J. Wainwright, A. Wibisono, 2015 "Optimal rates for zero-order convex optimization: the power of two function evaluations".

[Sha13] O. Shamir, 2013 "On the Complexity of Bandit and Derivative-Free Stochastic Convex Optimization".

[WS17] B. Woodworth and N. Srebro. "Lower bound for randomized first order convex optimization." arXiv preprint arXiv:1709.03594 (2017).